# Towards Completion of the “Periodic Table” of Di-2-Pyridyl Ketoxime

**DOI:** 10.3390/molecules30040791

**Published:** 2025-02-08

**Authors:** Christina Stamou, Christina D. Polyzou, Zoi G. Lada, Konstantis F. Konidaris, Spyros P. Perlepes

**Affiliations:** 1Department of Chemistry, University of Patras, 26504 Patras, Greece; xrstamou@gmail.com (C.S.); chpolyzou@upatras.gr (C.D.P.); 2Institute of Chemical Engineering Sciences, Foundation for Research and Technology-Hellas (FORTH/ICE-HT), Platani, P.O. Box 1414, 26504 Patras, Greece; 3Department of Chemistry, Materials Science and Chemical Engineering “Giulio Natta”, Politecnico di Milano, Via L. Mancinelli 7, 20131 Milan, Italy

**Keywords:** coordination chemistry, di-2-pyridyl ketoxime’s metal complexes, magnetic properties, reactivity, synthesis, structures

## Abstract

The oxime group is important in organic and inorganic chemistry. In most cases, this group is part of an organic molecule possessing one or more donor sites capable of forming bonds to metal ions. One family of such compounds is the group of 2-pyridyl (aldo)ketoximes. Metal complexes of 2-pyridyl oximes continue to attract the intense interest of many inorganic chemistry groups around the world for a variety of reasons, including their interesting structures, physical and biological properties, and applications. A unique member of 2-pyridyl ketoximes is di-2-pyridyl ketoxime (dpkoxH), which contains two 2-pyridyl groups and an oxime functionality that can be easily deprotonated giving the deprotonated ligand (dpkox^−^). The extra 2-pyridyl site confers a remarkable flexibility resulting in metal complexes with exciting structural and reactivity features. Our and other research groups have prepared and characterized many metal complexes of dpkoxH and dpkox^−^ over the past 30 years or so. This work is an attempt to build a “periodic table” of dpkoxH, which is near completion. The filled spaces of this “periodic table” contain metal ions whose dpkoxH/dpkox^−^ complexes have been structurally characterized. This work reviews comprehensively the to-date published coordination chemistry of dpkoxH with emphasis on the syntheses, reactivity, relationship to metallacrown chemistry, structures, and properties of the metal complexes; selected unpublished results from our group are also reported. The sixteen coordination modes adopted by dpkoxH and dpkox^−^ have provided access to monomeric and dimeric complexes, trinuclear, tetranuclear, pentanuclear, hexanuclear, heptanuclear, enneanuclear, and decanuclear clusters, as well as to a small number of 1D coordination polymers. With few exceptions ({M^II^Ln^III^_2_} and {Ni^II^_2_Mn^III^_2_}; M = Ni, Cu, Pd, and Ln = lanthanoid), most complexes are homometallic. The metals whose ions have yielded complexes with dpkoxH and dpkox^−^ are Cr, Mn, Fe, Co, Ni, Cu, Zn, Ru, Rh, Pd, Ag, Cd, Re, Os, Ir, Au, Hg, lanthanoids (mainly Pr and Nd), and U. Most metal complexes are homovalent, but some mixed-valence Mn, Fe, and Co compounds have been studied. Metal ion-assisted/promoted transformations of dpkoxH, i.e., reactivity patterns of the coordinated ligand, are also critically discussed. Some perspectives concerning the coordination chemistry of dpkoxH and research work for the future are outlined.

## 1. Introduction and Organization of This Review

The oxime group (R_1_R_2_C=NOH, vide infra) is important in organic and bioorganic chemistry [1]. Oximes can be categorized into two broad families based on the reactants for their synthesis; aldoximes are derived from aldehydes, while ketoximes are derived from ketones (Figure 1). The general formula of oximes is R_1_R_2_C=NOH, where R_1_ is an organic side, and R_2_ is an H atom (for aldoximes) or an organic group (for ketoximes). Depending on the relative locations of the higher priority group and the –OH group, oximes exist in two forms: *syn* and *anti*. Both exist for aldoximes and ketoximes, with the exception of aromatic aldoximes (Figure 2). Equally important is the role of the oxime functionality in coordination and supramolecular chemistry.

In most cases, the oxime group is part of an organic molecule that possesses one or more sites containing lone pair(s) of electrons, i.e., other donor sites. One family of such compounds is the group of 2-pyridyl (aldo)ketoximes (Figure 3, left), where R is a non-donor group, e.g., H, Me, Ph, etc. Another similar family of 2-pyridyl oximes consists of molecules in which the R group is a potential donor site, e.g., NH_2_ (pyridine-2-amidoxime). In the special case where R is a second 2-pyridyl group, the resulting molecule is di-2-pyridyl ketoxime (Figure 3, right), abbreviated as dpkoxH; the H in the abbreviation denotes the acidic hydrogen atom of the oxime group. The IUPAC name of this compound is di-2-pyridin-2-yl-methanone oxime. The dpkoxH compound is an exciting ligand for a variety of reasons (vide infra). Our and other research teams have been studying intensely the coordination chemistry of dpkoxH, and many metal complexes have been prepared and studied. Over the past 25 years or so, we have been trying to create a “periodic table” of dpkoxH, by synthesizing complexes with as many metal and metalloid ions as possible. The spaces in this “periodic table” contain ions, whose complexes with dpkoxH or its anionic derivative (dpkox^−^) have been structurally characterized; few exceptions (in which the metal complexes have been fully characterized by physical and spectroscopic methods) are also mentioned. Thus, this comprehensive review describes the metal chemistry of dpkoxH and dpkox^−^. Some of the complexes reported to-date are from our group, and this justifies the relatively high number of self-citations.

The emphasis of this work is on the synthetic and structural aspects of the metal complexes; in a few cases, a brief note on their spectroscopic and physical properties will be given. We intend to avoid long preparative descriptions and we shall thus present the syntheses of the complexes with the aid of balanced chemical equations written in their molecular (and not ionic) format. Such equations are correct only if we assume that there is only one metal-containing product in the reaction system; in some cases, this is not the case because equilibria may exist in the solution, a fact supported by the low yields of the preparations. However (as one of the referees points out), caution is needed in the messages conveyed through the use of balanced equations. Most structures have been generated from the corresponding CIFs, which has have been deposited with the Cambridge Crystallographic Data Center (CCDC); some structures are illustrated in a schematic way (ChemDraw). For the description of the coordination modes of dpkoxH and dpkox^−^ (and other ligands), the well-established “Harris notation” [2] (and its alternative using numerical subscripts) is used. The ligation modes are also represented in a schematic way (ChemDraw) and, for clarity reasons, bold lines will be used for the coordination bonds. The oxidation states of the metals in their complexes are indicated in several ways in their formulae in order to be unambiguous to the readers. Oxidation states are often not written for metals in which these are common, e.g., Ni and Zn. For mixed-valence homometallic complexes, the metals in the formulae are separated according to their oxidation states; for mixed-valence compounds containing three metal ions, the symbol of the metal appears once and the oxidation states (in Latin) are written as superscripts in a way that makes clear the oxidation levels of the metals involved. In the structural discussions, the focus is on the molecular structures with sporadic descriptions of the supramolecular features of the complexes.

This review is divided into sections. Section 2 provides the reader with basic information concerning the organic, coordination, and supramolecular chemistry of the simple oxime group, together with the importance of this group and its complexes in various branches of Chemistry. Section 3 gives an overview of the chemistry of the metal complexes of simple 2-pyridyl (aldo)ketoximes with a non-donor substituent on the oxime carbon (Figure 3, left). Since the coordination chemistry of dpkox^−^ is closely related to the evolution of metallacrowns [3], Section 4 will give some basic information about this unique class of compounds and the reader can thus easily understand some aspects of the following Section 5. Section 2, Section 3 and Section 4 are introductory of the article. Section 5 and Section 6 contain the central information of the present review. Section 5 describes the so-far published coordination chemistry of di-2-pyridyl ketoxime; we have chosen to arrange our discussion according to the nuclearity of the homo- or heterometallic products. Another way would be arranging the complexes based on the elements as they appear in the standard periodic table. Since most complexes are dinuclear and polynuclear, we prefer the former way of presentation. Section 6 describes briefly some unpublished results on the topic from our group. Section 7 is a brief summary of the material mentioned in Section 5 and Section 6, also providing some perspectives and a prognosis for the future. 

A review dealing exclusively with the coordination chemistry of dpkoxH has never appeared. An older review [4] from our group concerning the coordination chemistry of pyridyl oximes (not confined to those possessing a second 2-pyridyl group) contains a chapter with metal complexes of dpkoxH and/or dpkox^−^ published before 2005. 

## 2. The Oxime Group: A Versatile Tool in Chemistry

The real history of oximes begins in 1885 when Tschugaeff used dimethylglyoxime for the gravimetric determination of Ni^2+^ in the form of the highly insoluble, red bis(dimethylglyoximato)nickel(II) [5]. This reaction still exists in the undergraduate chemistry laboratory curriculum (qualitative and quantitative analysis); the complex is also prepared in inorganic chemistry laboratories to prove its square planar structure by a magnetic susceptibility measurement (the complex is diamagnetic).

The synthesis of oximes is a rather easy process, as it involves the direct reaction of aldehydes and ketones with hydroxylamine. Other methods [1,4] include nitrosation at a carbon bearing an active hydrogen, addition of NOCl to olefins, oxidation of primary aliphatic amines, reduction of aliphatic nitro compounds, and addition of Grignard reagents to the conjugate bases of nitro compounds. Oximes exhibit a rich reactivity [1,6], including the following: (i) their utilization as precursors for iminylradicals (through N-O bond fragmentation), which can be used for the synthesis of amino alcohols, amines, and N-containing heterocyclic compounds; (ii) their easy transformation into oxime radicals (known as iminoxyl radicals) which are valuable intermediates in many organic syntheses, e.g., formation of isoxazoline derivatives; and (iii) their ability to release NO which is a unique endogenous regulatory agent, participating in many physiological processes across some organ systems. 

The oxime group is also present in several natural products and compounds with significant biological activity. Typical examples are pralidoxime, trimedoxime, and obidoxime (Figure 4) which are efficient inhibitors (and hence antidotes) of the highly toxic organophosphates [6]. Oxime ethers (compounds with alkyl or aryl substituents on the O atom) are also important in medicinal chemistry. For example, fluvoxamine (Figure 4) is an antidepressant that belongs to the selective serotonin reuptake inhibitor family [6]. In addition, derivatives of oximes are important in modern agriculture as agents which protect crops. A characteristic example [6] is kresoxim-methyl (Figure 4), a broad-spectrum fungicide, that controls and treats fungal problems on crops.

Complexes containing the oxime group, either neutral or deprotonated (oximato complexes), have played important roles in several aspects of coordination [7] and bioinorganic [8,9] chemistry, homogeneous catalysis [10], and molecular magnetism [11,12]. The oxime and oximate groups can bind a metal ion in a number of modes, shown in Figure 5. These modes can lead to a variety of mononuclear, dinuclear, polynuclear (coordination clusters), and polymeric (coordination polymers) complexes. The dinuclear, polynuclear, and polymeric complexes can be either homo- or heterometallic. 

The reactivity of the coordinated oxime group is also a research topic under intense research activity [4,13,14]. The general manners of the nucleophilic or electrophilic additions to the polarized C=N bond are illustrated in Figure 6. Around 15–20 years ago, most of the reported metal-involving oxime reactions were metal-mediated, whereas nowadays the focus is on metal-catalyzed transformations. There is also an interest in the rather rare metal-mediated processes for the synthesis of free oximes. 

Due to space limitation, we briefly comment on the uncommon metal-mediated synthesis of free oximes, without C-, O-, and N-functionalization. The basic routes are shown in Figure 7. The reactions include (between others) the formation of the oxime via the treatment of carbonyl compounds with H_2_NOH in the presence of a metal ion (**A**→**B**→**C**) and the involvement of the intermediate formation of **D** with its subsequent reaction with NO^+^ to give **E** followed by isomerization of this nitroso compound to oxime **C** [14]. 

The main goal of supramolecular chemistry is to control the packing of the molecules in the crystal via intermolecular interactions. The synthons that are involved in the self-assembly of homomeric or heteromeric aggregates are crucial for the prediction and construction of supramolecular assemblies. H bonds, which are often strong and directional, are ideal tools for these processes. The most often used synthons in supramolecular chemistry and crystal engineering are the carboxylic, amide, and alcohol groups. In contrast, the oxime group has received less attention. This group is capable of forming three types of classical H bonds (Figure 8); oxime groups can act as H-bond donors (via the –O-H) and as H-bond acceptors (through both –C=N and –O-H). The most common H-bonding arrangements for the simple oxime group are illustrated in Figure 9 and involve dimeric (**I**) and catemeric (**II**, **III**) forms. Less common trimeric and tetrameric forms have also been observed [1,6].

**Figure 7 molecules-30-00791-f007:**
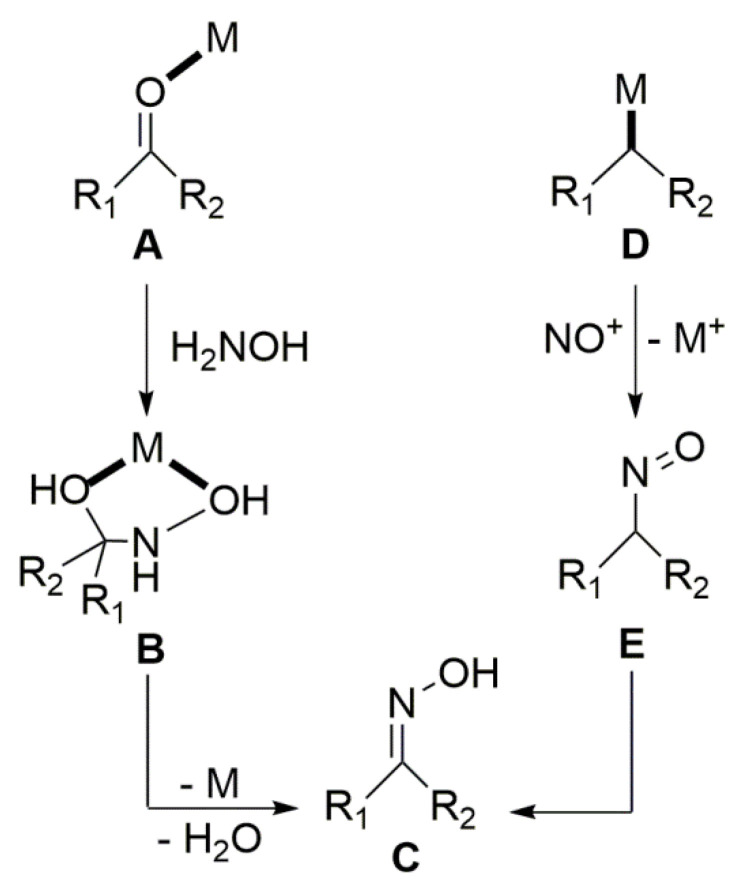
Two routes for the metal-mediated synthesis of oximes (without C-, N-, and O-functionalization). This drawing was adapted from Ref. [14]. M = metal center.

**Figure 8 molecules-30-00791-f008:**
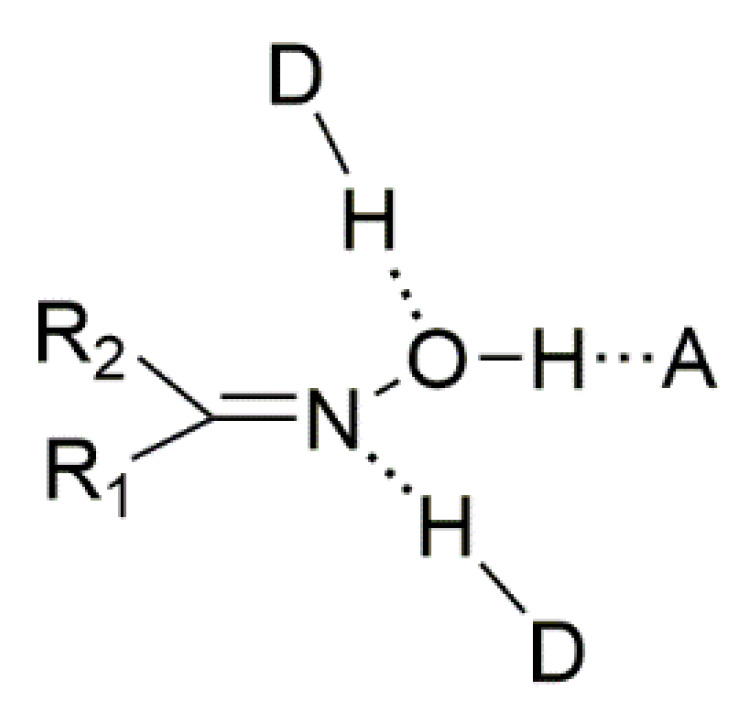
The three types of classical H bonds formed by the oxime group. H-bond acceptors can be the oxime N and O atoms, while the H-bond donor can be the O atom. A = acceptor, D = donor.

**Figure 9 molecules-30-00791-f009:**
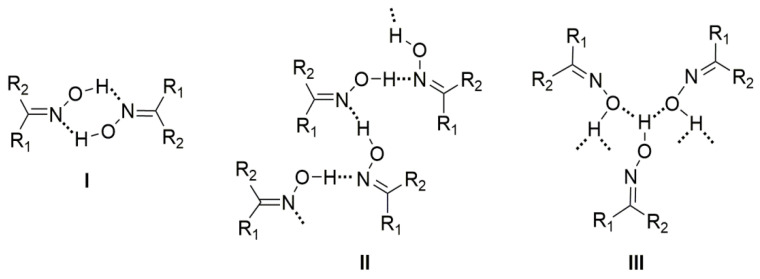
H-bonded dimer (**I**) and catemers formed by O-H∙∙∙N (**II**) and O-H∙∙∙O (**III**) H bonds.

## 3. Metal Complexes of Simple 2-Pyridyl Aldo(Keto)ximes

As mentioned in Section 1, the oxime group is most often a part of organic molecules bearing one or more other donor sites. When one donor site is the 2-pyridyl group and the other group to which the carbon atom is attached contains no donor atoms, the resulting family of molecules (Figure 3, left; R = H, Me, Ph, …) contains three atoms potentially capable of forming coordination bonds. The neutral molecules, and especially their deprotonated versions, are versatile ligands in coordination chemistry, as proven by the variety of their to-date established coordination modes illustrated in Figure 10. 

The 2-pyridyl oxime ligands of Figure 3 (left) have been writing their own “history” in inorganic chemistry. These ligands have been used to achieve several goals, including the following: (i) the modeling of solvent extraction of toxic Cd(II) from aqueous environments using 2-pyridyl ketoxime extractants [15]; (ii) the synthesis of 3d/4f dinuclear and polynuclear complexes [16]; (iii) the “switching on” of single-molecule magnetism (SMM) properties [17]; (iv) the isolation of single-chain magnets (SCMs) [18]; and (v) the linking of smaller clusters to supramolecular assemblies by means of coordination bonds [19]. We briefly mention an example of two of these research objectives.

The 1:3 reaction between the triply oxido-bridged, antiferromagnetically coupled triangular Mn(III) complexes [Mn_3_O(O_2_CR)_6_(py)_3_](ClO_4_) (**F**; R = Me, Et, Ph; py = pyridine) and methyl 2-pyridyl ketoxime (mepaoH; R = Me in Figure 3, left) in MeCN/MeOH led to complexes [Mn_3_O(O_2_CR)_3_(mepao)_3_](ClO_4_) (**G**) in high yields (>75%), Equation (1). The 1:3 ratio was selected to allow for the incorporation of one mepao^−^ ligand onto each edge of the {Mn^III^_3_(μ-O)}^7+^ core. Thus, one carboxylato ligand on each edge and one py molecule at each Mn^III^ atom are replaced by one 2.111 mepao^−^ group (Figure 10). This substitution gives *S* = 6 spin ground states for the products **G** which are single-molecule magnets (SMMs); in contrast, the reactants possess a low-spin ground state and non-SMM behavior [17,20]. Thus, the mepao^−^ ligand is responsible for “switching on” exciting magnetic properties.
[Mn^III^_3_O(O_2_CR)_6_(py)_3_](ClO_4_)+ 3 mepaoH $\overset{MeCN/MeOH}{\to }$
[Mn^III^_3_O(O_2_CR)_3_(mepao)_3_](ClO_4_)+ 3 RCO_2_H + 3 py(1)**F**
**G**


The 2-pyridyl ketoximes with long aliphatic R chains, e.g., 1-(2-pyridyl)-trideca-1-one oxime (2PC12) [R = (CH_2_)_11_CH_3_ in Figure 3, left], are efficient agents for the liquid–liquid extraction of toxic Cd(II) from chloride-containing aqueous media. Cadmium(II) can be effectively extracted using chloroform or hydrocarbon solutions of 2PC12 as an organic phase. Solution studies have indicated that the neutral octahedral species [CdCl_2_(extractant)_2_] is formed during the extraction process, allowing the transfer of the complexed toxic metal ion into the organic phase, from which it is stripped with aqueous NH_3_ [21]. The preparation and structural characterization of complex [CdCl_2_(phpaoH)_2_] (**H**) [15], where phpaoH is phenyl 2-pyridyl ketoxime (R = Ph in Figure 3, left), proves that neutral octahedral complexes [CdCl_2_(extractant)_2_] are capable of existence and that **H** is a model for the species formed during the extraction.

## 4. Metallacrowns: Short Notes

Since the coordination chemistry of dpkoxH is related to the evolution of metallacrowns, in this section, we give some information concerning the latter. Metallacrowns (MCs) belong to a broad class of metallamacrocycles, which also include metallacryptands, metallacoronates, metallacalixarenes, metal molecular wheels, rings and polygons, and metal molecular machines [3,22]. MCs have been employed as anion, cation and molecular recognition agents, catalysts, sensors, and building blocks for coordination polymers; in addition, they often have interesting structures. MCs are the inorganic analogs of the macrocyclic crown ethers first reported by Pederson (1987 Nobel prize) in 1967. The MC-crown ether analogy is illustrated in Figure 11 for the particular case of 12-membered rings with iron(III). The typical definition of an MC is a repeat unit of –[M-N-O]_n_- in a cyclic arrangement; in this arrangement, the ring metal ions and nitrogen atoms replace the methylene carbon atoms of a crown ether. A perusal of Figure 11 reveals that the 12-crown-4 (12-C-4) structure is similar to that of the 12-MC-4 one. Both molecules can give four O atoms as donors to an ion that can exist in the central cavity. Structural studies show that both molecules have similar sizes in the central cavity, despite the difference between the C-O and C-C bond lengths and those involving M-O or M-N bonds (M = a first-row transition metal ion). The difference in bond lengths is mitigated by the different bond angles which are ca. 109° for sp^3^ carbon, but ca. 90° for square planar or octahedral metal ions. The MCs have a higher affinity for encapsulation of M^II^ and M^III^ first-row transition metal ions because the deprotonated O atoms of MCs are better donors to intermediate-valent ions than are the neutral O atoms of the organic crown ethers. The MCs become robust when N and O atoms from deprotonated hydroxamic acids or/and oximate groups are utilized for bonding [22]. 

As in crown ethers, MCs are named taking into account the ring size and the number of O donors. For example, 12-MC-4, where MC represents metallacrown, is a 12-membered ring consisting of four repeating –[M-N-O-]- units with four donating O atoms. The nomenclature [3] involves the bound central metal ion, the ligand that creates the ring, and any bound or unbound ions. A typical nomenclature scheme is {MX[ring size-MC_M,Z(L)_-ring oxygens]}Y. In this scheme, M is the bound central metal (if any) including its oxidation state, X is any bound anion, M’ is the ring metal with its oxidation state, Z is the third heteroatom (usually N), L is the organic ligand used, and Y is any unbound anion. Sometimes, there can be unbound cations, and these are placed before the bound central metal. Examples of MC types include 9-MC-3, 10-MC-5, 12-MC-3, 12-MC-4, 12-MC-6, 14-MC-7, 15-MC-3, 15-MC-5, 15-MC-6, 16-MC4, 16-MC-8, 20-MC-10, 22-MC-11, 24-MC-6, 24-MC-8, 24-MC-12, 30-MC-10, 32-MC-8, 36-MC-12, 40-MC-10, 48-MC-24, 60-MC-20, etc. [3].

## 5. The Published Coordination Chemistry of Di-2-Pyridyl Ketoxime

### 5.1. Di-2-Pyridyl Ketoxime: An Exciting Ligand

Di-2-pyridyl ketoxime (dpkoxH; Figure 3, right) is a very interesting ligand, as it will be shown in the next parts, and occupies a special position among the 2-pyridyl ketoximes. Not only is the R unit a donor site, but also this group is a second 2-pyridyl group. A unique aspect of dpkoxH (and its anionic derivative) is its extraordinary coordination flexibility and versatility which have resulted in interesting metal complexes from the structures and properties viewpoints (vide infra). As has already been mentioned above, the deprotonated ligand is relevant to the chemistry of MCs. Another interesting aspect of dpkoxH is its activation by 3d-metal ions resulting in complexes that contain a different ligand. A characteristic example is the reaction system [V^III^Cl_3_(THF)_3_] (THF = tetrahydrofurane) which, depending on the reaction conditions, provided access [23] to complexes [V^IV^OCl_2_(dpi)(THF)] (L), [V^IV^OCl_2_(adpe)] (M), [V^IV^OCl_2_(adpm)] (N), and [V^IV^OCl_2_(adpm)] (O), where dpi, adpe, and adpm are the neutral molecules shown in Figure 13. The vanadyl complexes are all octahedral. The dpi molecule behaves as an N(pyridyl), N(imino)-bidentate chelating ligand, whereas the other ligands are tridentate chelating employing the three N atoms (M, N), and the two 2-pyridyl N atoms and the ether O atom in the case of O. Thus, complexes N and O have the same formula but differ in the coordination mode of the ligand adpm.

The to-date crystallographically confirmed ligation modes of dpkoxH and dpkox^−^ are illustrated in Figure 14. The structural formulae of the non-trivial ligands that are present in the complexes are shown in Figure 15. 

### 5.2. Mononuclear Complexes

The to-date structurally characterized mononuclear (monometallic) complexes of dpkoxH or/and dpkox^−^ (**1**–**17**) are listed in Table 1. The molecular structures of some of them are presented in Figures 16, 17, and 19–25 and described in Refs. [24,25,26,27,28,29,30,31,32,33,34,35,36,37], while docking representations of complex **11** and diflH are shown in Figure 18. 

We first give some general remarks about the mononuclear compounds: (i) Most complexes contain the neutral ligand (dpkoxH); exceptions are the Au(III) complex [Au^III^Cl(dpkox)_2_] (**23**) and complex [Cu^II^Cl(dpkox)(dpkoxH)] (**8**), the latter containing both the neutral and deprotonated ligands. There is no specific effect of the two different forms of the ligand on the structural features of the complexes. The anion (dpkox^−^) behaves as a 1.0110 ligand, while the neutral molecule (dpkoxH) behaves either as a 1.0110 or 1.0101 ligand. As we shall see later, both forms can act as bridging ligands. (ii) Many complexes are neutral; complexes **6**, **18**, **19**, **21,** and **22** are cationic with +1 charge. (iii) With the exception of **18**–**22** which are organometallic-coordination hybrids, the rest are purely coordination complexes; and (iv) the Ni(II) [**2**–**7**] and four Zn(II) [**14**–**17**] complexes contain monoanionic non-steroidal anti-inflammatory carboxylate drugs (mef^−^, tolf^−^, difl^−^, mclf^−^ fluf^−^, mcpa^−^, indo^−^; for their structural formulae; see Figure 15), which give remarkable biological properties to these compounds (vide infra). We now discuss some important synthetic, structural, and biological features of selected mononuclear complexes. 

**Figure 16 molecules-30-00791-f016:**
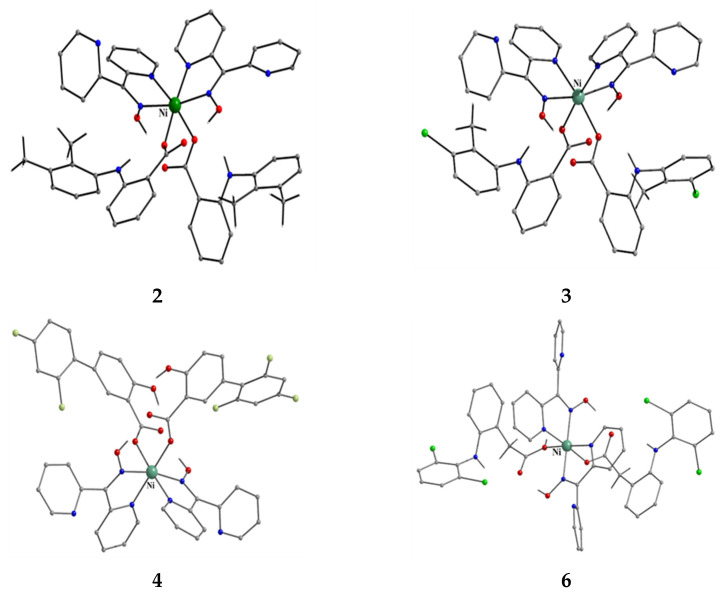
The structures of the molecules [Ni(mef)_2_(dpkoxH)_2_], [Ni(tolf)_2_(dpkoxH)_2_], [Ni(dilf)_2_(dpkoxH)_2_], and the cation [Ni(dicl)(diclH)(dpkoxH)_2_]^+^ that are present in the crystals of **2**, **3**, **4,** and **6**, respectively.

**Figure 17 molecules-30-00791-f017:**
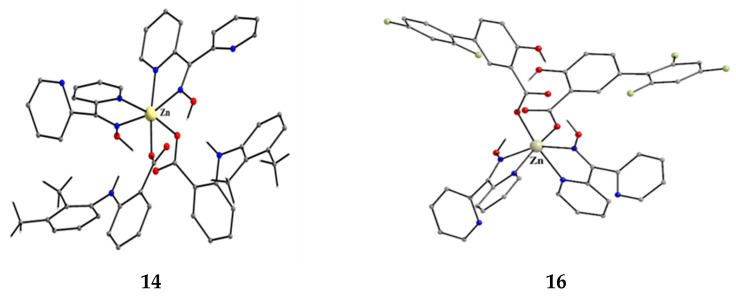
The structures of the molecules [Zn(mef)_2_(dpkoxH)_2_] and [Zn(difl)_2_(dpkoxH)_2_] that are present in the crystals of **14** and **16**, respectively.

Complexes **2**–**7** [25,26,27,28,29] and **14**–**17** [36,37,38,39] are easily prepared in moderate-to-good yields by the reaction of a metal “salt” (usually chloride), dpkoxH, the neutral ancillary NSAID ligand, and a base in a 1:2:2:2 molar ratio; the commonly used solvent was MeOH. As a typical example, the preparation of **3** is given in Equation (2).
${\mathrm{NiCl}}_{2}\cdot 6H_{2}O+2\;\mathrm{tolfH}+2\;\mathrm{dpkoxH}+2\;\mathrm{KOH}\;\overset{\mathrm{MeOH}}{\to }\;$[Ni(tolf)_2_(dpkoxH)_2_]+ 2 KCl + 8 H_2_O(2)
**3**


Complexes **2**–**7** and **14**–**17** (Figure 16 and Figure 17) are neutral. The octahedral coordination sphere of the metal ion consists of two carboxylato O atoms from two monodentate deprotonated NSAIDs and two 1.0110 dpkoxH ligands (Figure 14). The O atoms and the pyridyl N atoms are in *cis* position and the oxime nitrogens in *trans*; thus, each O atom is trans to a pyridyl N atom. Complex **6** is cationic because one of NSAID ligands is neutral (diclH; Figure 16); the positive charge is counterbalanced by a non-coordinated difl^−^ anion. The molecular structure of this complex [29] is similar to that of **2**–**7** and **14**–**17**, with the carboxylate/carboxylic (via the O atom of the double carbon-oxygen bond) O atoms and the pyridyl N atoms in *cis* positions, and the oxime nitrogens in *trans*. In most molecular structures, there are intramolecular H bonds with the oxime O and the secondary amine N atoms of the NSAID as donors, and the carboxylato (“free” and coordinated) O atoms as acceptors. 

**Figure 18 molecules-30-00791-f018:**
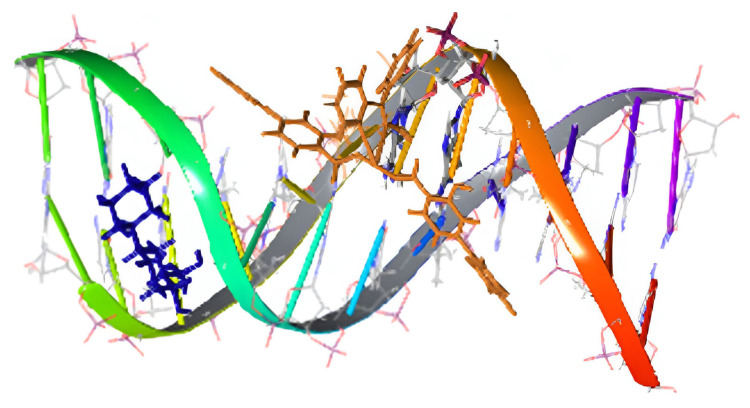
Docking representations of [Zn(difl)_2_(dpkoxH)_2_] (**16**) in orange and free diflH in blue interacting with calf-thymus DNA. Reproduced from Ref. [38]. Copyright 2017 Elsevier.

The presence of NSAIDs in **2**–**7** and **14**–**17** provides the complexes with remarkable biological properties, including albumin binding, interactions with calf-thymus DNA, and antioxidant and anticholinergic activities. In most cases, the complexes are more active than the free NSAIDs, suggesting their potential application as metallodrugs. In silico studies have also been performed. For example, docking calculations for calf-thymus DNA have indicated that **16** has a high affinity in accordance with the large experimentally calculated DNA-binding constant (K_b_ = 5.85 × 10^5^ M^−1^) [38]. The differences in binding position for **16** and free diflH are presented in Figure 18 and it is clear that the two compounds interact with different bases. The complex interacts with H bonds and π-π stacking interactions; in contrast, diflH interacts only with H bonds. For the complex, this justifies the interaction mode revealed from experiments (in vitro UV-Vis spectroscopy and viscosity measurement, and indirectly with fluorescence emission) that could be either intercalation (π-π interactions in-between bases) or groove-binding via H-bonding [38].

Compound **8** [31] was prepared by the reaction shown in Equation (3) with a yield of ca. 70%. Its molecular structure (Figure 19) is unique because it seems to contain simultaneously the neutral (dpkoxH) and deprotonated (dpkox^−^) ligands, both coordinated with the 1.0110 mode; a chlorido ligand completes the coordination number five at Cu^II^. A notable feature is the presence of a strong intramolecular H bond between the two oxime/oximate O atoms (O∙∙∙O = 2.442 Å, O∙∙∙H∙∙∙O = 169.1°). The H atom is located at practically the middle of the O atom’s distance, preventing a clear assignment of the exact position of this atom and consequently a precise attribution of the negative charge on one of the two ligands. This strong H bond rationalizes the high thermodynamic stability of the complex. Alternatively, the (dpkox∙∙∙H∙∙∙dpkox)^−^ system can be considered as one tetradentate N_4_-chelating ligand (Figure 20). The metal coordination geometry is well described as distorted square pyramidal with the chlorido ligand occupying the apical position.

CuCl_2_∙2H_2_O + 2 dpkoxH + Et_3_N
$\overset{{\mathrm{CH}}_{2}{\mathrm{Cl}}_{2}}{\to }$
[Cu^II^Cl(dpkox)(dpkoxH)]
 + (Et_3_NH)Cl + 2 H_2_O

(3)

**8**


Complexes **11**–**13** were prepared by the 1:1 reactions between the corresponding zinc halides and dpkoxH in alcohols in yields >60%. Use of excess of dpkoxH gives again the 1:1 complexes and not the anticipated [ZnX_2_(dpkoxH)_2_] (X = Cl, Br) ones. The Zn^II^ atom is in a distorted tetrahedral arrangement (Figure 21) with one 1.0101 dpkoxH ligand and two terminal halido groups. The smallest coordination angles are those involving the two 2-pyridyl N atoms (~92°) and the largest ones are those involving the two X^−^ ligands (~115°). The six-membered chelating rings are not planar. The complexes described in Refs. [33,34,35] are polymorphs. 

As already mentioned, compounds **18**–**22** are organometallic hybrids and were prepared by the reactions shown in Equations (4)–(7). The yields were in the range of 65–82%.
[(Ph)_2_Ru^III^_2_Cl_4_] + 2 dpkoxH$+2\;{\mathrm{NH}}_{4}{\mathrm{PF}}_{6}\;\overset{\mathrm{dry}\;\mathrm{MeOH}}{\to }$2 [(Ph)Ru^III^Cl(dpkoxH)](PF_6_)+ 2 NH_4_Cl   (4)**P**
**18**

[(Cp*)_2_Rh^III^_2_Cl_4_] + 2 dpkoxH+ 2 NH_4_PF_6_ $\overset{\mathrm{dry}\;\mathrm{MeOH}}{\to }$
2 [(Cp*)Rh^III^Cl(dpkoxH)](PF_6_)+ 2 NH_4_Cl(5)**Q**
**19**

[Re^I^(CO)_5_Cl]+ dpkoxH $\overset{\mathrm{toluene}}{\to }$
[Re^I^(CO)_3_Cl(dpkoxH)]+ 2 CO                                                        (6)**R**
**20**

[(Cp*)_2_Ir^III^_2_Cl_4_]+ 2 dpkoxH + 2 NH_4_PF_6_ $\overset{\mathrm{dry}\;\mathrm{MeOH}}{\to }$
2 [(Cp*)Ir^III^Cl(dpkoxH)](PF_6_)+ 2 NH_4_Cl    (7)**S**
**21** (major isomer)


**22** (minor isomer)


**Figure 21 molecules-30-00791-f021:**
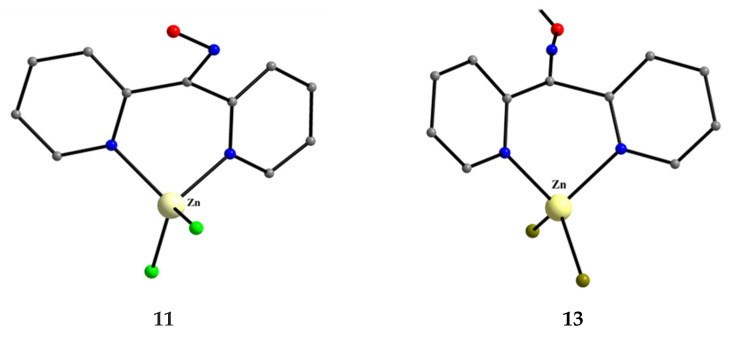
The molecular structures of [ZnCl_2_(dpkoxH)_2_] (**11**) and [ZnBr_2_(dpkoxH)_2_] (**13**).

**Figure 22 molecules-30-00791-f022:**
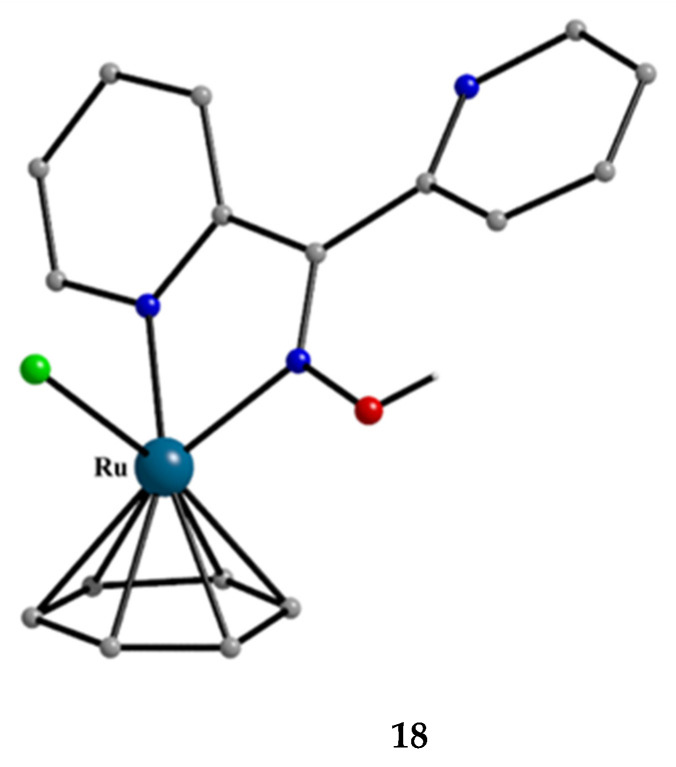
The molecular structure of the cation [(Ph)Ru^III^Cl(dpkoxH)]^+^ that is present in the hexafluorophosphate salt **18**.

**Figure 23 molecules-30-00791-f023:**
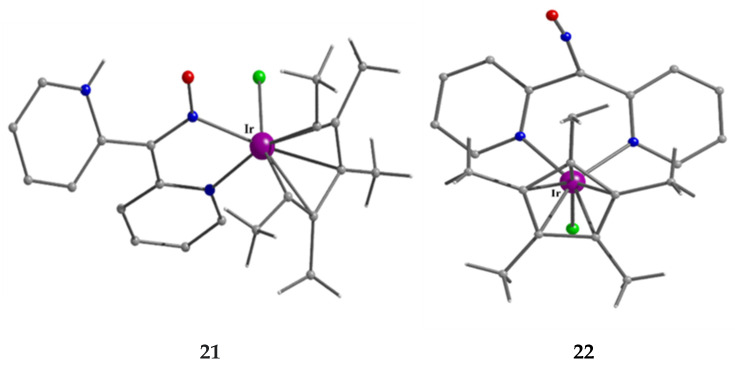
The molecular structures of the isomeric cations that are present in complexes **21** (major isomer) and **22** (minor isomer); see text for discussion.

**Figure 24 molecules-30-00791-f024:**
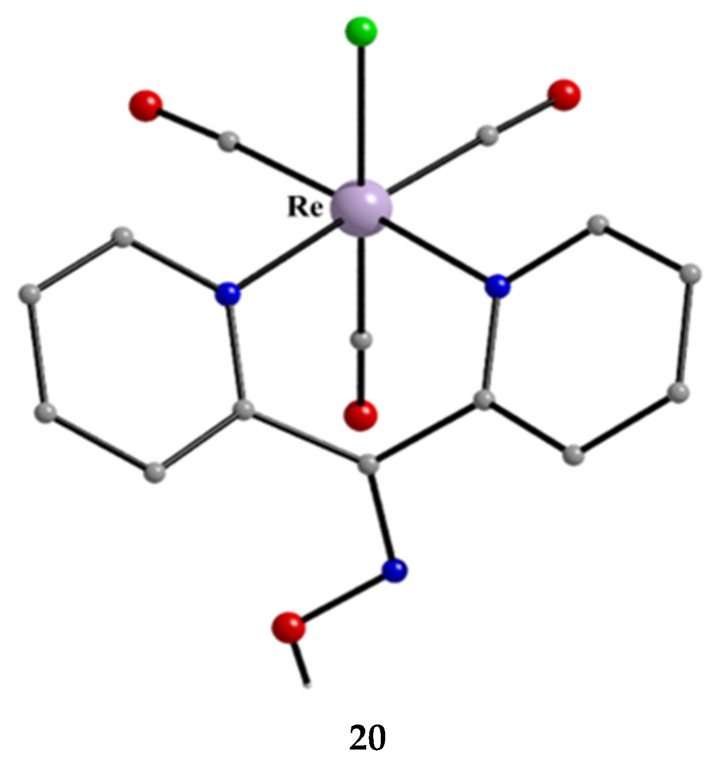
The molecular structure of [Re^I^(CO)_3_Cl(dpkoxH)] (**20**).

The synthetic chemistry of the [(Cp*)_2_Ir^III^_2_Cl_4_] (**S**)/dpkoxH reaction system in dry MeOH is interesting. The reaction gives a mixture of two products, as evidenced by ^1^H NMR spectroscopy [40]. The products could not be prepared separately in pure form, and so they were isolated manually. Single-crystal X-ray crystallography revealed that the major coordination isomer was **21** in which the dpkoxH molecule is coordinated to Ir^III^ through one of the 2-pyridyl N atoms and the oxime nitrogen (1.0110; Figure 14), whereas in the case of the minor isomer **22,** the coordination is through the two 2-pyridyl nitrogens (1.0101; Figure 14).

The molecular structures of the cations of **18**, **21,** and **22** are illustrated in Figure 22 and Figure 23, respectively. Complex **19** is structurally similar to **22**, and it is the only product from the reaction. The molecular structure of **20** is presented in Figure 24.

The structures of **18**, **19**, **21,** and **22** [39] display a characteristic three-legged “piano stool” arrangement around the central metal center; the coordination sites are occupied by two N atoms from the chelating dpkoxH (1.0110 in **18** and **21**; 1.0101 in **19** and **22**), one chlorido group and the Ph/Cp* ring in a η^6^ (**18**)/η^5^ (**19**, **21**, **22**) manner. In the distorted octahedral complex **20**, the three carbonyl groups are in *fac* positions and are orthogonal with an average C-Re^I^-C angle of 88.5°. The distortion from the ideal octahedral geometry in this complex arises from the constraints associated with the binding of the 1.0101 dpkoxH ligand, the N(pyridyl)-Re^I^-N(oxime) bite angle being ca. 80° [41]. 

It should be mentioned at this point that Ru complexes with O,N-containing ligands currently attract [43] intense interest because they are very good molecular catalysts for water oxidation (WOCs).
K[Au^III^Cl_4_] + 2 dpkoxH$\overset{H_{2}O}{\to }$[Au^III^Cl(dpkox)_2_]+ 2 HCl + KCl(8)**T**
**23**


Complex **23** was prepared in H_2_O at room temperature (yield not reported), Equation (8). The Au^III^ center has a coordination number of five with four N atoms from two 1.0110 dpkoxH molecules; the four nitrogen donor atoms are nearly coplanar. Five-coordinated gold(III) complexes are rather rare. A chlorido group occupies the apical position in the square pyramidal geometry (Figure 25) [42].

### 5.3. Dinuclear Complexes

The to-date structurally characterized dinuclear (dimetallic) complexes of dpkoxH and dpkox^−^ (**24**–**35**) are listed in Table 2. The molecular structures of selected complexes are presented in Figure 26, Figure 27, Figure 28, Figure 29, Figure 30, Figure 31, Figure 32, Figure 33, Figure 34 and Figure 35 and fully described in Refs. [31,34,35,44,45,46,47,48,49,50,51,52,53]. All the dinuclear complexes contain divalent or monovalent metals. The bridging ligands vary from neutral dpkoxH (**25**, **26**, **32, 34, 35**) to dpkox^−^ (**24**, **27**, **31**, **33**), demonstrating the flexibility of di-2-pyridyl ketoxime in both neutral and deprotonated forms; in two cases, the metal ions are bridged by bromido (**30**) and thiocyanido (**35**) groups. Complex **28** contains both the neutral and deprotonated ligands.

The isolation of **24** was rather surprising. The reaction of [Cr^III,III,III^_3_O(piv)_6_(H_2_O)](piv) (**U**), where piv^−^ is the Me_3_CCO_2_^−^ ligand (Figure 15), with three equivalents of dpkoxH in MeCN under aerobic and refluxing conditions for 12 h gave a dark brown solution from which brownish red crystals were isolated in low yield (~25%) upon layering the reaction solution with Et_2_O. Thus, an unusual reduction of chromium(III) to chromium(II) *in air* had taken place [44,45]. After many experiments, we came to the conclusion that the reducing agent is an amount of the dpkoxH ligand itself, and thus, the repeatedly low yields justify our belief that unidentified oxidation product(s) remain in solution. Metal ion-assisted oxidations of oximes have been reported [13,14], but reduction reactions of Cr(III) starting materials with ligands acting as reductants, and simultaneously appearing in the Cr(II) products, are extremely rare. Complex **24** can also be prepared by the reaction of [Cr^0^(CO)_6_] (**V**) with an excess of dpkoxH under refluxing aerobic conditions in MeCN/H_2_O in satisfactory yields, higher than 50%, based on the total available chromium [45], Equation (9).
2 [Cr^0^(CO)_6_] + 4 dpkoxH+ O_2_ $\overset{\mathrm{MeCN}/H_{2}O,T}{\to }$
[Cr^II,II^_2_(dpkox)_4_] + 12 CO+ 2 H_2_O(9)**V**
**24**


Crystals of **24** can be kept for days in the mother liquor. When isolated and exposed to air, they are oxidized to a green-brown powder containing Cr(III) as revealed by EPR spectroscopy; the room-temperature spectrum of the green-brown powder displays an isotropic signal, whose intensity increases with storage time, with *g* ≈ 2.1 at X-band frequency, characteristic of a Cr(III) content in the sample. The molecular structure of **24** is shown in Figure 26. The Cr^II^ atoms are doubly bridged by two 2.1110 dpkox^−^ ligands. A chelating 1.0110 dpkox^−^ group completes a five-coordinate geometry at each metal center. The coordination polyhedron about each Cr^II^ atom is described as a distorted trigonal bipyramid, with the axial sites occupied by an oximato nitrogen of a chelating dpkox^−^ group and an oximato nitrogen of a bridging dpkox^−^ ligand. The Cr∙∙∙Cr’ distance is long (~3.48 Å) and this precludes any thoughts of the existence of Cr^II^-Cr^II^ bond. The 3d^4^ Cr^II^ atoms are high-spin as deduced from the Cr^II^-O (~2.13 Å) and Cr^II^-N (1.96–2.05 Å) bond lengths and a room-temperature magnetic susceptibility measurement performed almost immediately (~3 min) after the isolation of **24** from the mother liquor. The 1.0110 ligands are in *syn* arrangement. Each dimer is stabilized by an intramolecular π-π stacking interaction between the terminal 1.0110 ligands, the interaction involving the coordinated 2-pyridyl rings. This interaction seems to be the driving force for the isolation of the *syn* terminal dpkox^−^ groups, and not the isomer with the anti-arrangement (or a mixture of the two isomers). The uncoordinated N and O atoms are acceptors of H bonds with the lattice H_2_O molecules (that exist in the crystal structure) as donors.

**Table 2 molecules-30-00791-t002:** Dinuclear metal complexes of dpkoxH and dpkox^−^.

Complex	Coordination Mode of dpkoxH/dpkox^− c^	Ref.
[Cr^II,II^_2_(dpkox)_4_] (**24**)	1.0110, 2.1110	[44,45]
[Mn^II,II^_2_(O_2_CCF_3_)_2_(hfac)_2_(dpkoxH)_2_] (**25**) ^a^	2.0111	[24]
[Ni_2_(PhPO_3_)_2_(dpkoxH)_4_] (**26**)	1.0110	[46]
[Cu^II,II^_2_(dpkox)_4_] (**27**)	1.0110, 2.1110	[47,48]
[Cu^II,II^_2_(dpkox)_2_(dpkoxH)_2_](ClO_4_)_2_ (**28**)	1.0110, 2.1110	[49]
[Cu^II,II^_2_Cl_4_(H_2_O)_2_(dpkoxH_2_)_2_]Cl_2_ (**29**) ^b^	1.011	[50]
[Cu^II,II^_2_Br_4_(dpkoxH_2_)_2_][Cu^II^Br_4_] (**30**)	1.0110	[34,35]
[Cu^II,II^_2_(hfac)_2_(dpkox)_2_] (**31**) ^a^	2.1110	[31]
[Cu^I,I^_2_Cl_2_(dpkoxH)_2_] (**32**)	2.0111	[51]
[Ru^I,I^_2_(CO)_4_(dpkox)_2_] (**33**)	2.1110	[52]
[Ag^I,I^_2_(NO_3_)_2_(dpkoxH)_2_] (**34**)	2.0111	[49,50]
[Hg^II,II^_2_(SCN)_4_(dpkoxH)_2_] (**35**)	1.0110	[53]

^a^ hfac is the hexafluoroacetylacetonato(−1) ligand. ^b^ dpkoxH_2_^+^ is protonated (at one of its 2-pyridyl nitrogen atoms) di-2-pyridyl ketoxime. ^c^ Using the Harris notation [2]. Lattice solvent molecules have not been incorporated in the formulae of the complexes.

Complex **25** was not prepared using a trifluoroacetate-containing Mn(II) starting material. Treatment of Mn(hfac)_2_∙3H_2_O (**W**) with one equivalent of dpkoxH in CH_2_Cl_2_ led to **25** in 70% yield [24]. The CF_3_CO_2_^−^ ligand arises from the transformation of hfac^−^, Equation (10). β-diketonates often undergo the retro-Claisen condensation reactions to give a ketone and a carboxylate under strongly basic conditions; in the present case, dpkoxH could function as the base to assist the decomposition. In the dinuclear complex (Figure 27), the two Mn^II^ centers are bridged by two 2.0111 dpkoxH ligands. Each metal ion is further coordinated by a chelating (1.11) hfac^−^ ligand, and a terminal (1.10) CF_3_CO_2_^−^ group, completing a distorted octahedral geometry at each Mn^II^. The chromophores are *fac*-{Mn^II^O_3_N_3_}. The long Mn∙∙∙Mn distance (5.603 Å) reflects the polyatomic nature of the bridges.
F_3_CCOCH_2_COCF_3_ $\overset{\mathrm{base}}{\to }$ CF_3_CO_2_^−^ + CH_3_COCF_3_(10)

**Figure 27 molecules-30-00791-f027:**
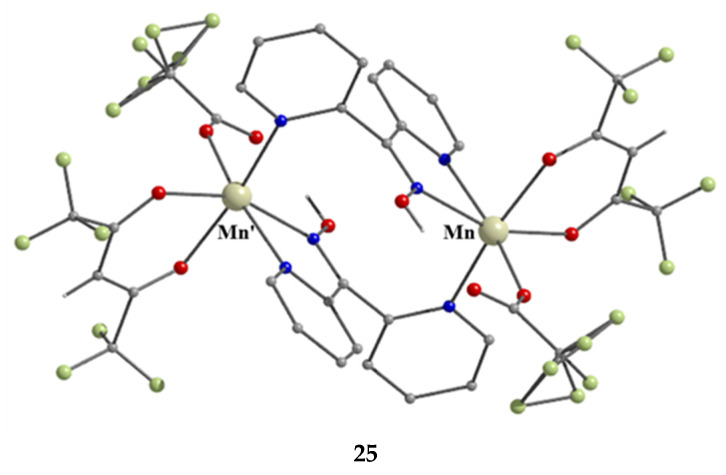
The structure of the centrosymmetric molecule [Mn^II,II^_2_(O_2_CF_3_)_2_(hfac)_2_(dpkoxH)_2_] that is present in the crystal of **25**.

The Cu(II)/dpkoxH reaction system is extremely fertile (Table 2 and vide infra). Complex **27** was prepared [47,48] through the reaction represented by Equation (11). Its molecular structure is shown in Figure 28. The complex is isostructural with **24** (Figure 26).
2 Cu^II^(ClO_4_)_2_∙6H_2_O + 8 dpkoxH + 4 NaOH $\overset{\mathrm{MeOH}/\mathrm{DMF}}{\to }$
2 [Cu^II,II^_2_(dpkox)_4_]+ 4 NaClO_4_ + 16 H_2_O(11)
**27**


**Figure 28 molecules-30-00791-f028:**
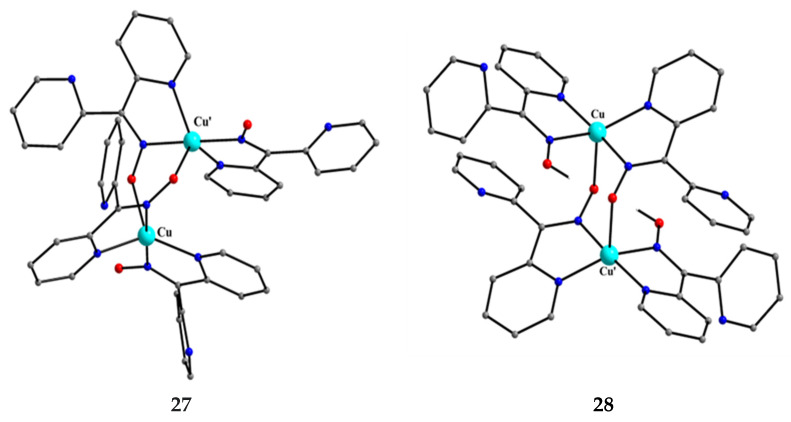
The structure of the molecule [Cu^II,II^_2_(dpkox)_4_] and the cation [Cu^II,II^_2_(dpkox)_2_(dpkoxH)_2_]^2+^ that are present in complexes **27** and **28**, respectively.

The 1:2 reaction between Cu^II^(ClO_4_)_2_∙6H_2_O and dpkoxH in alcohols at room temperature, in the absence of an external base, gave a green solution from which were subsequently isolated dark green crystals of the cationic complex **28** in typical yields of 35–40%, Equation (12) [49]. This complex seems to be an intermediate of the reaction that leads to the neutral dimer **27** [47,48]. In accordance with this, **28** reacts with two equivalents of LiOH∙H_2_O in MeOH/DMF to give **27** in yields ~50%, Equation (13). Complex **28** can also be isolated by stoichiometric acidification of **27** with aqueous HClO_4_ 1 N in MeOH, but in yields lower than 25%, Equation (14). In the centrosymmetric cation (Figure 28), the two Cu^II^ centers are doubly bridged by the *syn*, *anti* oximato groups of the two dpkox^−^ ligands. A chelating (1.0110) neutral dpkoxH molecule completes five-coordination at each metal ion. The geometry of copper(II) is distorted square pyramidal, the apical position being occupied by an oximato O atom.
2 Cu^II^(ClO_4_)_2_∙6H_2_O + 4 dpkoxH$\overset{\mathrm{ROH}}{\to }$[Cu^II,II^_2_(dpkox)_2_(dpkoxH)_2_](ClO_4_)_2_+ 2 HClO_4_ + 12 H_2_O           (12)

**28**

[Cu^II,II^_2_(dpkox)_2_(dpkoxH)_2_](ClO_4_)_2_$+2\;\mathrm{LiOH}\cdot H_{2}O\;\overset{MeOH/DMF}{\to }$2 [Cu^II,II^_2_(dpkox)_4_]+ 2 LiClO_4_ + 4 H_2_O(13)**28**
**27**

[Cu^II,II^_2_(dpkox)_4_]+ 2 HClO_4_$\overset{\mathrm{MeOH}/H_{2}O}{\to }$Cu^II,II^_2_(dpkox)_2_(dpkoxH)_2_](ClO_4_)_2_                                             (14)**27**

**28**

Although **27** and **28** have chemically similar {Cu^II,II^_2_(μ-ON)_2_}^2+^ cores, the topology of the oximato bridges and the coordination geometry of the metal ions are different. This difference has a dramatic influence on the magnetic properties of the complexes due to the different types of interactions between the Cu^II^ 3d orbitals and the p orbitals of the oximato bridge. The metal ions in **27** are strongly antiferromagnetically coupled, and the compound is almost diamagnetic at room temperature! In contrast, the Cu^II^∙∙∙Cu^II^ exchange interaction in **28** is ferromagnetic with an S = 1 ground state [49].

**Figure 29 molecules-30-00791-f029:**
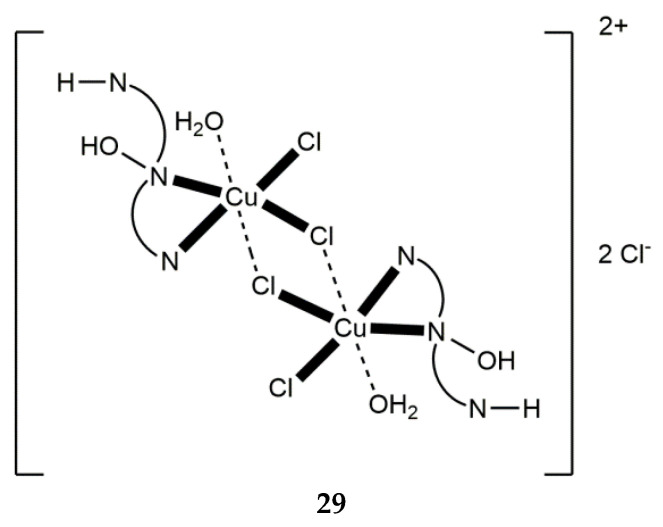
Schematic illustration of the structure of the cationic complex [Cu^II,II^_2_Cl_4_(H_2_O)_2_(dpkoxH_2_)_2_]Cl_2_ (**29**). The symbol with the three nitrogen atoms, the hydroxyl group, and the proton is a short representation of the cationic ligand dpkoxH_2_^+^ in which the non-coordinated 2-pyridyl N atom is protonated. The dashed bold lines indicate weak coordination bonds.

Change of the perchlorate anions of the copper(II) “salt” with chlorides and bromides has a remarkable impact on the chemical and structural identity of the products. The dinuclear complex **29** (Figure 29) was prepared by the reaction shown in Equation (15). The low pH (~2) has as a result the protonation of one 2-pyridyl N atom of each dpkoxH (thus becoming a pyridinium ring), and therefore a cation dpkoxH_2_^+^ is formed which is coordinated in a chelating fashion through the “free” 2-pyridyl N and the oxime N atoms (1.011 in Figure 14). The Cu^II^ centers are doubly bridged by two centrosymmetrically bonded chlorido ligands, while a weakly coordinated H_2_O molecule completes a Jahn–Teller distorted octahedral geometry at Cu^II^ (the coordination bonds of the Jahn–Teller positions are drawn with dashed lines in Figure 29) [50].
$2\;{\mathrm{Cu}}^{\mathrm{II}}{\mathrm{Cl}}_{2}\cdot 2H_{2}O+2\;\mathrm{dpkoxH}+2\;\mathrm{HCl}\;\overset{H_{2}O}{\to }$[Cu^II,II^_2_Cl_4_(H_2_O)_2_(dpkoxH_2_)_2_]Cl_2_+ 2 H_2_O(15)
**29**


An analogous reaction with that described in Equation (15), but using CuBr_2_ instead of CuCl_2_∙2H_2_O, gave a crystalline product that could be characterized. The CuBr_2_/dpkoxH/HBr reaction system gave complex [Cu^II,II^_2_Br_2_(dpkoxH_2_)_2_][Cu^II^Br_4_] (**30**) [34α]; the solvent and the yield were not reported. The reaction is represented by Equation (16). The complex is cationic, and the positive charge is counterbalanced by a distorted tetrahedral [Cu^II^Br_4_]^2−^ anion. In the cation, the two Cu^II^ atoms are doubly bridged by two asymmetrically bonded bromido groups (thus resembling the bridging unit of **29**). A terminal bromido group and a 1.011 dpkoxH_2_^+^ cationic ligand complete five-coordination at each metal ion (Figure 30). The aqua ligands of **29** are missing in **30**, presumably due to steric effects. Variable-temperature magnetic susceptibility data and X-band powder EPR spectra suggest a negligible Cu^II^∙∙∙Cu^II^ exchange interaction in **30** (like in **29**).
3 Cu^II^Br_2_ + 2 dpkoxH + 2 HBr →[Cu^II,II^_2_Br_2_(dpkoxH_2_)_2_][Cu^II^Br_4_](16)
**30**

**Figure 30 molecules-30-00791-f030:**
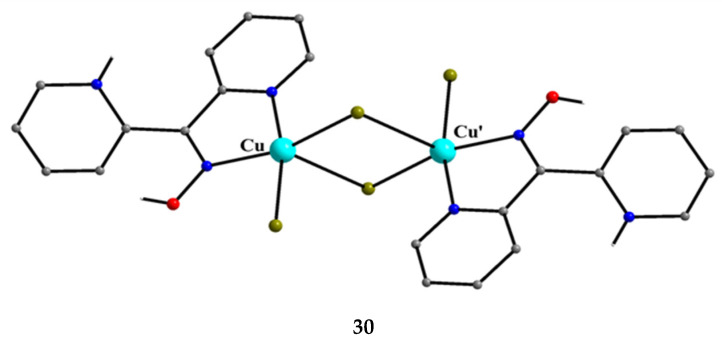
The structure of the cation [Cu^II,II^_2_Br_4_(dpkoxH_2_)_2_]^2+^ that is present in complex **30**.

The incorporation of the chelating ligand hfac^−^ in and omission of HCl from the Cu^II^Cl_2_∙2H_2_O/dpkoxH reaction system kept the nuclearity of the product to two, but changed the bridging unit. The 1:1:2 reaction between CuCl_2_∙2H_2_O, dpkoxH, and Na(hfac) in CH_2_Cl_2_ gave a slurry, which was filtered to remove the insoluble NaCl. The addition of Et_2_O/n-hexane into the dark green solution gave **31** a good yield (~60%), Equation (17). This complex can also be prepared by the treatment of **8** (Table 1) with one equivalent of Na(hfac), Equation (18). The Cu^II^ atoms in the centrosymmetric complex **31** (Figure 31) are doubly bridged by the diatomic oximato groups of 2.1110 dpkox^−^ ligands; the bridging CuNOCu’ unit is not planar. A bidentate chelating (1.11) hfac^−^ ligand completes five-coordination at Cu/Cu’. The geometry of each metal ion is almost perfect square pyramidal, with the apical position occupied by a hfac^−^ O atom. The Cu^II^∙∙∙Cu^II^ distance is 3.77 Å [31]. The crystal structure is stabilized by intermolecular Van der Waals F∙∙∙F contacts (2.92 Å) which link neighboring dinuclear molecules into 1D double chains; these interactions create channels in which lattice CH_2_Cl_2_ molecules reside. Variable-temperature magnetic data are indicative of a very strong intramolecular antiferromagnetic exchange interaction with a resulting S = 0 ground state, which is well-isolated from the S = 1 excited state. This magnetic feature seems to be typical for Cu(II) complexes with double oximato bridges, which usually exhibit nearly complete spin coupling even at 20 °C.
$2\;{\mathrm{Cu}}^{\mathrm{II}}{\mathrm{Cl}}_{2}\cdot 2H_{2}O+2\;\mathrm{dpkoxH}+4\;\mathrm{Na}(\mathrm{hfac})\;\overset{CH_{2}Cl_{2}}{\to }$[Cu^II,II^_2_(hfac)_2_(dpkox)_2_]+ 2 hfacH + 4 NaCl + 4 H_2_O(17)
**31**
2 [Cu^II^Cl(dpkox)(dpkoxH)] + 2 Na(hfac)$\;\overset{CH_{2}Cl_{2}}{\to }$
[Cu^II,II^_2_(hfac)_2_(dpkox)_2_]+ 2 dpkoxH + 2 NaCl        (18)
**31**


**Figure 31 molecules-30-00791-f031:**
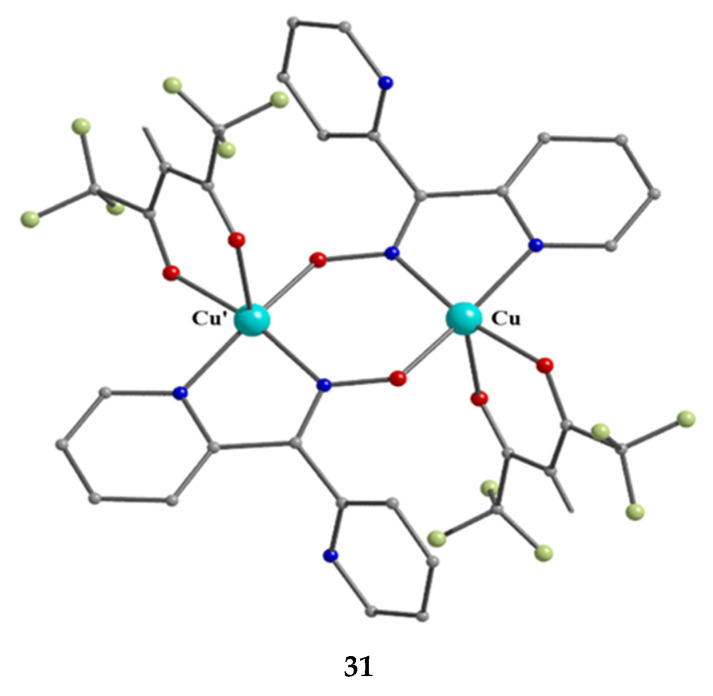
The molecular structure of [Cu^II,II^_2_(hfac)_2_(dpkox)_2_] (**31**).

The copper/chloride/di-2-pyridyl ketoxime chemistry is not only confined to Cu(II). The 1:2:excess CuCl_2_∙2H_2_O/dpkoxH/NaCl reaction system in EtOH/H_2_O, in the presence of L(+) ascorbic acid (a reducing agent), under gentle heating, gave complex [Cu^I,I^_2_Cl_2_(dpkoxH)_2_] (**32**) in low yield (ca. 20%). The molecular structure of the centrosymmetric diamagnetic complex is shown in Figure 32. The two Cu^I^ atoms are doubly bridged by two 2.0111 dpkoxH ligands. One terminal chlorido group completes a distorted tetrahedral geometry at each metal ion. The Cu^I^∙∙∙Cu^I^ separation is 4.83 Å.

**Figure 32 molecules-30-00791-f032:**
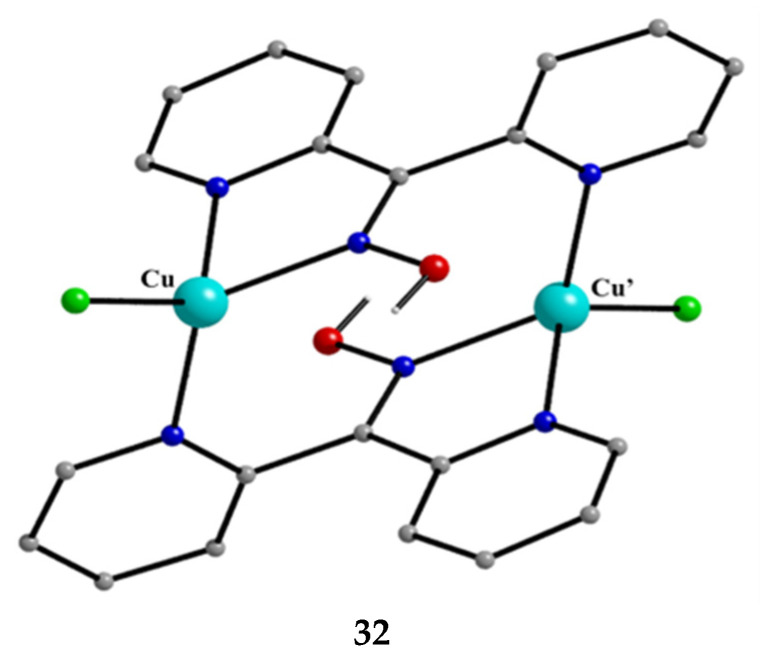
The molecular structure of [Cu^I,I^_2_Cl_2_(dpkoxH)_2_] (**32**).

From the 2nd- and 3rd-row transition metals, only Ru, Ag, and Hg have been reported to form dinuclear complexes with dpkoxH and dpkox^−^. Complex **33** [52] was isolated as the major reaction product (in a mixture with **50**, vide infra) using [Ru^0^_3_(CO)_12_] (**X**) as the starting material, Equation (19). The ^13^C{^1^H} NMR spectrum of the complex shows only two signals attributed to terminal carbonyl ligands. The molecule contains two 2.1110 dpkox^−^ ligands spanning the same edge of the dimetallic core in a head-to-tail arrangement (Figure 33). The Ru1-Ru2 distance is 2.620 Å, as expected for a single metal–metal bond, in accordance with the 34-electron count of the molecule.
2 [Ru^0^_3_(CO)_12_]$+6\;\mathrm{dpkoxH}+3/2\;O_{2}\;\overset{\mathrm{THF}}{\to }$3 [Ru^I,I^_2_(CO)_4_(dpkox)_2_]+ 12 CO + 3 H_2_O(19)**X**
**33**


**Figure 33 molecules-30-00791-f033:**
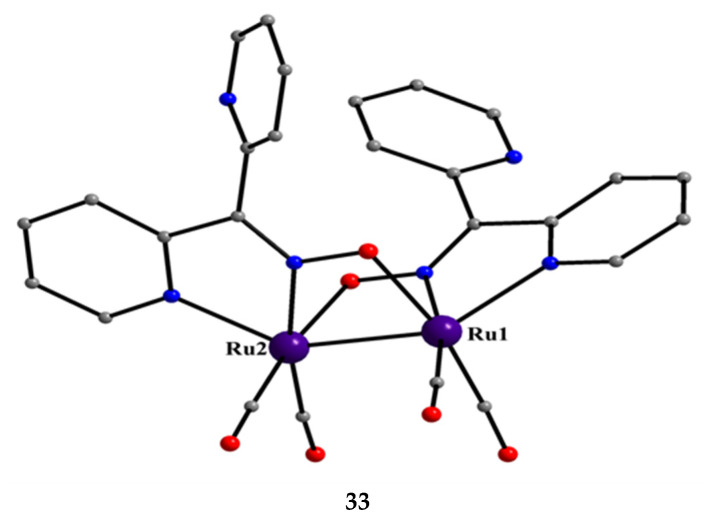
The molecular structure of [Ru^I,I^_2_(CO)_4_(dpkox)_2_] (**33**).

Complex **34** was prepared by the 1:1 reaction of AgNO_3_ and dpkoxH in warm water (~60 °C) at neutral pH. The two Ag^I^ atoms in the centrosymmetric complex are bridged by two 2.0111 dpkoxH molecules, with two monodentate nitrato groups completing a distorted tetrahedral coordination geometry of the metal centers (Figure 34).

**Figure 34 molecules-30-00791-f034:**
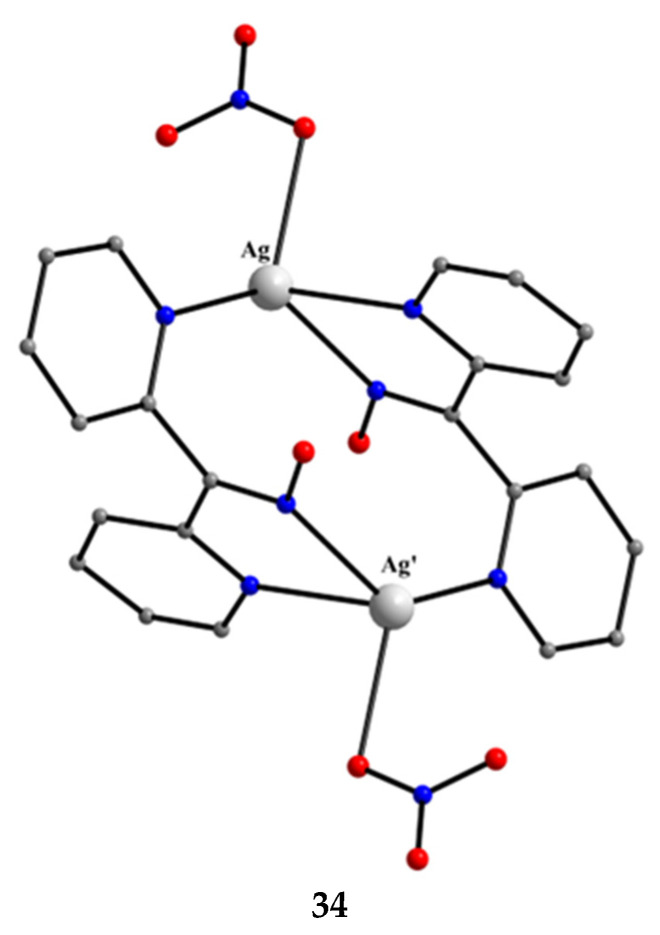
The molecular structure of [Ag^I,I^_2_(NO_3_)_2_(dpkoxH)_2_] (**34**).

The reaction of Hg(SCN)_2_ with an excess of dpkoxH (1:2) in Me_2_CO gave complex **35** in typical yields of 30–35% [53]. The Hg^II^ centers in the centrosymmetric complex are bridged by a pair of *syn*, *syn*-2.11 SCN^−^ groups. The Hg^II^ atoms are each chelated by an 1.0110 dpkoxH ligand and a terminal S-bonded thiocyanido ion (Figure 35). The metal coordination geometry is intermediate between square pyramidal and trigonal bipyramidal. The crystal lattice of the complex is built through H bonds and S∙∙∙S contacts. The dinuclear molecules are connected through intermolecular H bonds with the oxime O atom as donor and the N atom of the uncoordinated 2-pyridyl ring as acceptor; these H bonds create a 2D lattice. The 2D sheets are further linked through intermolecular S∙∙∙S interactions (S∙∙∙S = 3.83 Å) generating a 3D architecture. 

**Figure 35 molecules-30-00791-f035:**
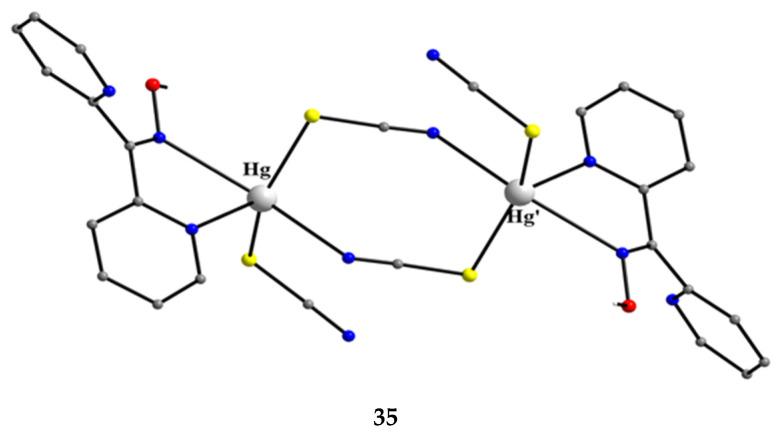
The molecular structure of [Hg^I,I^_2_(SCN)_4_(dpkoxH)_2_] (**35**).

### 5.4. Trinuclear Complexes

The to-date structurally characterized homotrinuclear and heterotrinuclear metal complexes of dpkoxH and dpkox^−^, **36**–**52** and **53**–**63**, are listed in Table 3 and Table 4, respectively. The molecular structures of some complexes are presented in Figures 36, 38–45, 47, and 48, while physical/spectroscopic data for a few of them are shown in Figures 37 and 49–53.

**Figure 36 molecules-30-00791-f036:**
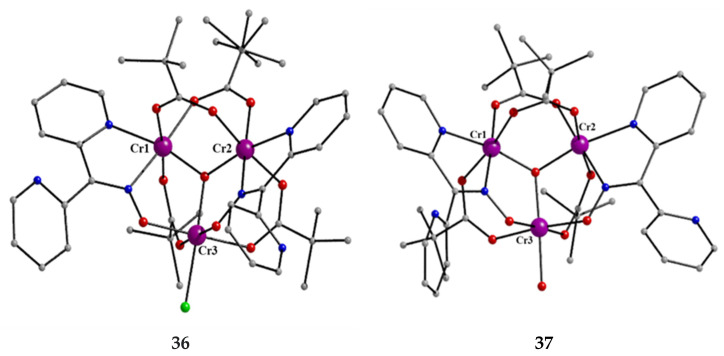
The structures of the cation [Cr^III,III,III^_3_O(piv)_4_(H_2_O)(dpkox)_2_]^+^ and the molecule [Cr^III,III,III^_3_O Cl(piv)_4_(dpkox)_2_] that are present in the crystals of **36** and **37**, respectively.

The Cr(III) complexes **36** and **37** were prepared by the reactions shown in Equations (20) and (21), respectively [45]; yields were 34% (**36**) and 72% (**37**). Complex **36** could also be isolated using **37** as the starting material, Equation (22). The synthesis of **37** can be achieved *only* by solvothermal techniques, which sometimes favor the formation of metastable compounds that are difficult or even impossible to obtain by convenient coordination chemistry techniques, i.e., solution chemistry under atmospheric pressures and at temperatures limited to the boiling points of the solvents. The presence of the coordinated chlorido group in **37** is intriguing and can be attributed to the decomposition of the CH_2_Cl_2_ solvent used, Equation (23). This transformation is favored by a combination of the presence of Cr^III^ and the high-pressure/high-temperature conditions during the solvothermal reactions [45].
[Cr^III,III,III^_3_O(piv)_6_(H_2_O)_3_](piv) + 2 dpkoxH + NaNO_3_ $\overset{{\mathrm{Me}}_{2}\mathrm{CO},\;T}{\to }$(20)                           **Y**                                   [Cr^III,III,III^_3_O(piv)_4_(H_2_O)(dpkox)_2_](NO_3_) + Na(piv) + 2 pivH + 2 H_2_O                                                                                                  **36**
2 [Cr^III,III,III^_3_O(piv)_6_(H_2_O)_3_](piv) + 4 dpkoxH + 2 CH_2_Cl_2_ 
$\;\overset{{\mathrm{CH}}_{2}{\mathrm{Cl}}_{2}/{\mathrm{Me}}_{2}\mathrm{CO},\;T,P}{\to }$
(21)                                **Y**                                                                2 [Cr^III,III,III^_3_OCl(piv)_4_(dpkox)_2_] + 6 pivH + Cl-CH = CH-Cl + 6 H_2_O                                                                                                                                                         **37**
[Cr^III,III,III^_3_OCl(piv)_4_(dpkox)_2_] + NaNO_3_ + H_2_O $\overset{Me_{2}CO/H_{2}O}{\to }\;$ [Cr^III,III,III^_3_O(piv)_4_(H_2_O)(dpkox)_2_](NO_3_) + NaCl(22)                     **37**                                                                                                       **36**
2 CH_2_Cl_2_ + 2 piv^−^ → Cl-CH = CH-Cl + 2 pivH + 2 Cl^−^(23)

The core of the cation [Cr^III,III,III^_3_O(piv)_4_(H_2_O)(dpkox)_2_]^+^ in complex **36** consists of a near-equilateral Cr^III,III,III^_3_ triangle capped by a central μ_3_-oxido (μ_3_-O^2−^) group. The μ_3_-O^2−^ ion is ~0.20 Å above the plane of the three metal ions and occupies the common vertex of the coordination octahedra around them. Each of the Cr1∙∙∙Cr3 and Cr2∙∙∙Cr3 edges (Figure 36) is further bridged by one *syn*, *syn*-2.11 piv^−^ ligand and one oximato group of a 2.1110 dpkox^−^ ligand, while the Cr1∙∙∙Cr2 edge is further bridged by two *syn*, *syn*-piv^−^ groups. A terminal aquo ligand completes a distorted octahedral geometry around Cr3. There are two crystallographically independent clusters in the asymmetric unit with almost identical structural characteristics. The resulting cationic units of **36** are counterbalanced by NO_3_^−^ anions, strongly H-bonded to the coordinated aquo group. The molecular structure of **37** is very similar to that of the cation of **36**, the only difference being the presence of a chlorido group in the former instead of the aquo group in the latter [45]. Thus, **37** consists of neutral molecules (Figure 36).

**Figure 37 molecules-30-00791-f037:**
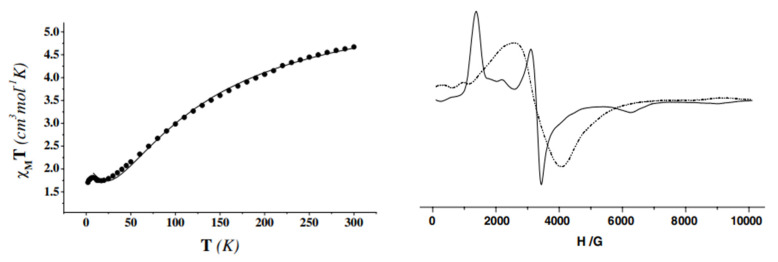
(**Left**) χ_M_T vs. T plot for a powdered sample of **37** in a 3 KG field (χ_M_ is the molar magnetic susceptibility and T is the absolute temperature). The solid line is the fit of the experimental data to the appropriate 2-*J* model [45]. (**Right**) X-band EPR spectra of solid **37** recorded at 295 K (dashed line) and 20 K (solid line). Reproduced from Ref. [45]. Copyright 2007 Elsevier.

Variable-temperature magnetic susceptibility data (Figure 37, left), and EPR data at 295 and 20 K (Figure 37, right) of solid **37** reveal an antiferromagnetically coupled system with an *S* = 3/2 ground state.

**Table 3 molecules-30-00791-t003:** Homotrinuclear (homotrimetallic) complexes of dpkoxH and dpkox^−^.

Complex ^a,b^	Coordination Mode of dpkoxH/dpkox^− c^	Ref.
[Cr^III,III,III^_3_O(piv)_4_(H_2_O)(dpkox)_2_](NO_3_) (**36**)	2.1110	[45]
[Cr^III,III,III^_3_OCl(piv)_4_(dpkox)_2_] (**37**)	2.1110	[45]
[Mn^II,IV,II^_3_(OMe)_2_Cl_2_(dpkox)_4_] (**38**)	2.1110	[54]
[Mn^II,IV,II^_3_(OMe)_2_(SCN)_2_(dpkox)_4_] (**39**)	2.1110	[54,55]
[Mn^II,IV,II^_3_(OMe)_2_(NCO)_2_(dpkox)_4_] (**40**)	2.1110	[54]
[Mn^II,IV,II^_3_(ed)Cl_2_(dpkox)_4_] (**41**)	2.1110	[56]
[Mn^II,IV,II^_3_(pd)Cl_2_(dpkox)_4_] (**42**)	2.1110	[56]
[Mn^II,IV,II^_3_(perH_2_)Cl_2_(dpkox)_4_] (**43**)	2.1110	[56]
[Fe^II,III,II^_3_(dpkox)_6_](ClO_4_) (**44**)	2.1110	[57]
[Ni_3_(shi)_2_(dpkoxH)_2_(py)_2_] (**45**)	1.0110	[58]
[Ni_3_(N_3_)_4_(Medpt)_2_(dpkox)_2_] (**46**)	2.1110	[59,60]
[Cu^II,II,II^_3_(OH)(O_2_CPh)_2_(dpkox)_3_] (**47**)	2.1110	[61]
[Cu^II,II,II^_3_(OH)Cl(^t^BuPO_3_H)(dpkox)_3_] (**48**)	2.1110	[62]
[Cu^II,II,II^_3_(OH)Br(^t^BuPO_3_H)(dpkox)_3_] (**49**)	2.1110	[62]
[Ru_3_(CO)_8_(dpkox)_2_] (**50**)	2.1110	[52]
[Os_3_(H)(CO)_9_(dpkox)] (**51**)	2.1110	[52]
[Os_3_(CO)_8_(dpkox)_2_] (**52**)	2.110	[52]

^a^ Abbreviations of the non-common ancillary ligands: piv = pivalato(−1), ed = the dianion of ethanediol, pd = the dianion of propanediol, perH_2_ = the dianion of pentaerythritol, shi = the triply deprotonated salicylhydroxamic acid (**K** in Figure 12), Medpt = N-methyldipropylenetriamine, py = pyridine, ^t^BuPO_3_H = the singly deprotonated tert-butylphosphoric acid. ^b^ The structural formulae of these ligands (with the exception of shi^3−^) are illustrated in Figure 15. ^c^ Using the Harris notation [2]. Lattice solvent molecules have not been incorporated in the formulae of the complexes.

Manganese forms two families of linear trinuclear mixed-valence clusters. A common feature of the compounds is that the central metal is Mn^IV^ and the terminal ones are Mn^II^. Complexes **38**–**40** were prepared from the reactions represented by Equations (24) and (25), where X = SCN, OCN [54,55]. The yields were not reported.
$3\;{\mathrm{Mn}}^{\mathrm{II}}{\mathrm{Cl}}_{2}\cdot 4H_{2}O+4\;\mathrm{dpkoxH}+4\;\mathrm{NaOH}+2\;\mathrm{MeOH}+½\;O_{2}\;\overset{MeOH,\;O_{2}}{\to }$[Mn^II,IV,II^_3_(OMe)_2_Cl_2_(dpkox)_4_]+ 4 NaCl + 17 H_2_O(24)
**38**
3 Mn^II^Cl_2_∙4H_2_O + 4 dpkoxH + 4 NaOH + 2 NaX + 2 MeOH + ½ O_2_$\overset{MeOH,\;O_{2}}{\to }$
(25)
[Mn^II,IV,II^_3_(OMe)_2_X_2_(dpkoxH)_4_]+ 6 NaCl + 17 H_2_O
**39**, **40**


The molecular structures of **38** and **40** are shown in Figure 38; the structure of **39** is similar. In the centrosymmetric molecules, the central Mn^IV^ atom is triply bridged to each Mn^II^ by a 2.2 methoxido group, and the diatomic oximato groups of two 2.1110 dpkox^−^ ligands; the O atoms of the latter are bonded to Mn^IV^. Two 2-pyridyl and two oximato N atoms (from two dpkox^−^) and a terminal chlorido (**38**), isothiocyanido (**39**), and isocyanato (**40**) groups complete a distorted trigonal prismatic coordination geometry around each Mn^II^, the distortion depending on the bound inorganic anion [54,55]. The Mn^IV^∙∙∙Mn^II^ magnetic exchange interactions are rather weakly ferromagnetic, and the spin ground state is *S* = 13/2 for the three compounds. XANES spectroscopy for **38** and **39** was used to clearly prove that the valence isomer {Mn^II^Mn^IV^Mn^II^} is present in the complexes and not the most commonly observed {Mn^III^Mn^II^Mn^III^} one [55].

**Figure 38 molecules-30-00791-f038:**
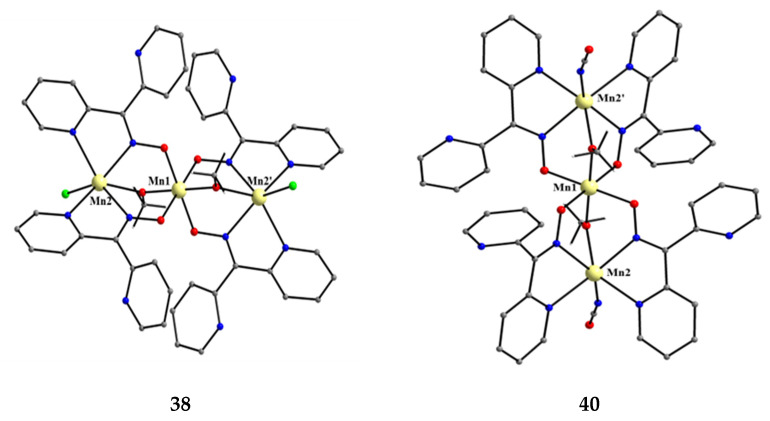
The molecular structure of [Mn^II,IV,II^_3_(OMe)_2_Cl_2_(dpkox)_4_] (**38**) and [Mn^II,IV,II^_3_(OMe)_2_(NCO)_2_(dpkox)_4_] (**40**).

Complexes **41**–**43** were prepared from the reactions represented by Equations (26) and (27), where LH_2_ = edH_2_ (ethanediol), pdH_2_ (propanediol) and perH_4_ = pentaerythritol (for the structural formulae of their anionic forms; see Figure 15). The yields were in the range of 55–65% [56].
3 Mn^II^Cl_2_∙4H_2_O + 4 dpkoxH + LH_2_ + 4 Et_3_N + ½ O_2_ $\overset{MeCN,\;T}{\to }$
[Mn^II,IV,II^_3_(L)Cl_2_(dpkox)_4_]+ 4 (Et_3_NH)Cl + 13 H_2_O         (26)
**41**, **42**

3 Mn^II^Cl_2_∙4H_2_O + 4 dpkoxH + perH_4_ + 4 Et_3_N + ½ O_2_ $\overset{MeCN}{\to }$
[Mn^II,IV,II^_3_(perH_2_)Cl_2_(dpkox)_4_]+ 4 (Et_3_NH)Cl + 13 H_2_O(27)
**43**


**Figure 39 molecules-30-00791-f039:**
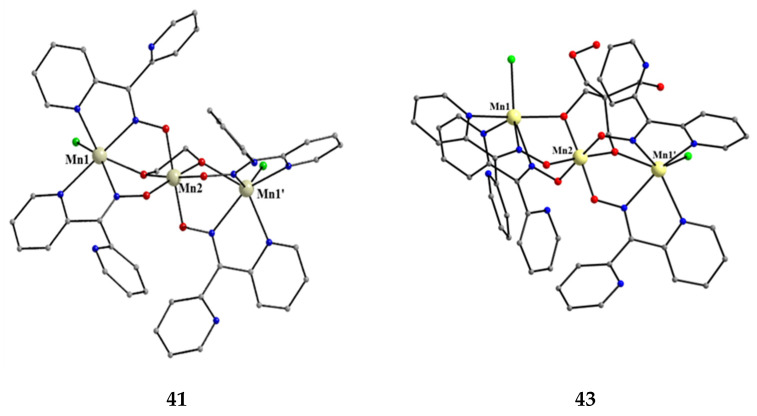
The molecular structures of [Mn^II,IV,II^_3_(ed)Cl_2_(dpkox)_4_] (**41**) and [Mn^II,IV,II^_3_(perH_2_)Cl_2_ (dpkox)_4_] (**43**); ed is the dianion of ethanediol and perH_2_ is the dianion of pentaerythritol (Figure 15).

The molecular structures of **41** and **43** are shown in Figure 39; the structure of **42** is similar, the only difference being the presence of the ancillary pd^2−^ ligand. Complexes **41**–**43** are structurally similar to **38** (Figure 38), but non-centrosymmetric. The linking between the central Mn^IV^ atom to each terminal, trigonal prismatic Mn^II^ atom occurs through a deprotonated alkoxido O atom of the 3.22 ed^2−^ and pd^2−^ ligands (**41**, **42**) or through a deprotonated O atom of the 3.2200 perH_2_^2−^ groups (**43**). Like **38**–**40**, complexes **41**–**43** are ferromagnetically coupled with an *S* = 13/2 ground state [56].

Iron forms an interesting, mixed-valence trinuclear complex based on dpkox^−^, Equation (28). The reported yield is low (~10%) [57].
3 Fe^II^(ClO_4_)_2_∙6H_2_O + 6 dpkoxH + ¼ O_2_ $\overset{\mathrm{MeOH}/H_{2}O}{\to }$
[Fe^II,III,II^_3_(dpkox)_6_](ClO_4_)+ 5 HClO_4_ + 18.5 H_2_O(28)
**44**


The molecular structure of the centrosymmetric cation that is present in **44** is shown in Figure 40. The central metal ion (Fe1) is bridged to each terminal ion (Fe2, Fe2′) through the oximato groups of three 2.1110 dpkox^−^ ligands in such a way that the six oximato O atoms are bonded to the central metal (Fe1). Three 2-pyridyl N atoms complete an octahedral N_6_ environment at each terminal Fe ion; the central Fe ion has a distorted trigonal prismatic geometry. Bond distances and magnetic data indicate that the terminal metals are low-spin Fe^II^ (and hence diamagnetic) and the central metal is high-spin Fe^III^ (*S* = 5/2) [57]; many Fe(II) complexes with {Fe^II^N_6_} chromophores are low-spin [iron(II) is a 3d^6^ system].

Nickel(II) forms two mixed-ligand trinuclear complexes which contain dpkoxH (**45**) or dpkox^−^ (**46**) as one type of ligands. Compound **45** was prepared in high yield (~70%) by the reaction represented in Equation (29); shi is the trianion of salicylhydroxamic acid (**K** in Figure 12) and py is pyridine. 

**Figure 40 molecules-30-00791-f040:**
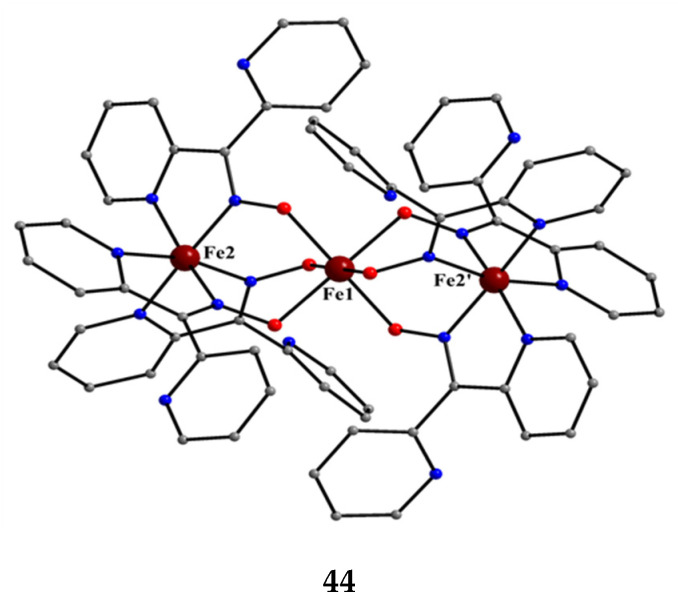
The structure of the cation [Fe^II,III,II^_3_(dpkox)_6_]^+^ that is present in complex **44**.




$3\;{\mathrm{NiCl}}_{2}\cdot 6H_{2}O+2\;{\mathrm{shiH}}_{3}\;+2\;\mathrm{dpkoxH}+6\;\mathrm{NaOH}+2\;\mathrm{py}\;\overset{MeOH/py}{\to }$

[Ni_3_(shi)_2_(dpkoxH)_2_(py)_2_]+ 6 NaCl + 24 H_2_O(29)

**45**




The molecule (Figure 41) is triangular. The crystallographically equivalent ions Ni2/Ni2’ have a square planar geometry and each is bonded to a 1.0110 dpkoxH ligand, while Ni1 is octahedral. The two py groups are coordinated to the octahedral Ni^II^ center. The linking between the three metal centers is achieved through two 2.1_1_1_1_1_2_1_2_ shi^3−^ ligands (Figure 42). 

**Figure 41 molecules-30-00791-f041:**
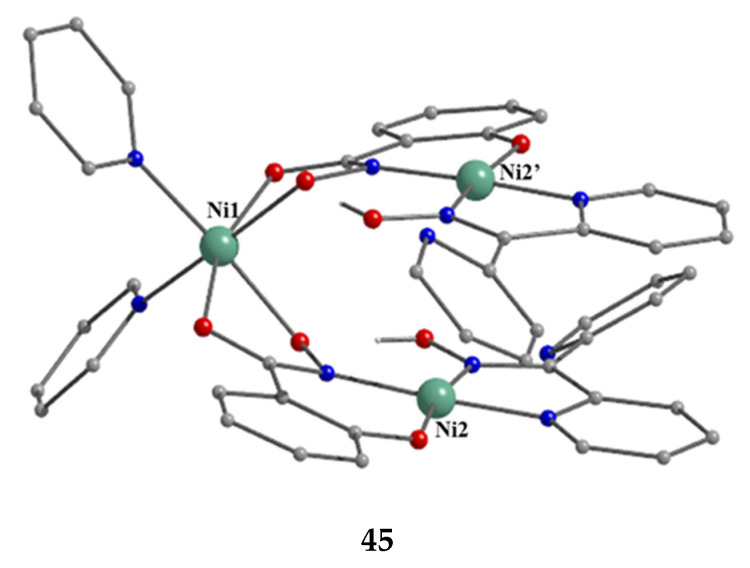
The molecular structure of [Ni_3_(shi)_2_(dpkoxH)_2_(py)_2_] (**45**).

**Figure 42 molecules-30-00791-f042:**
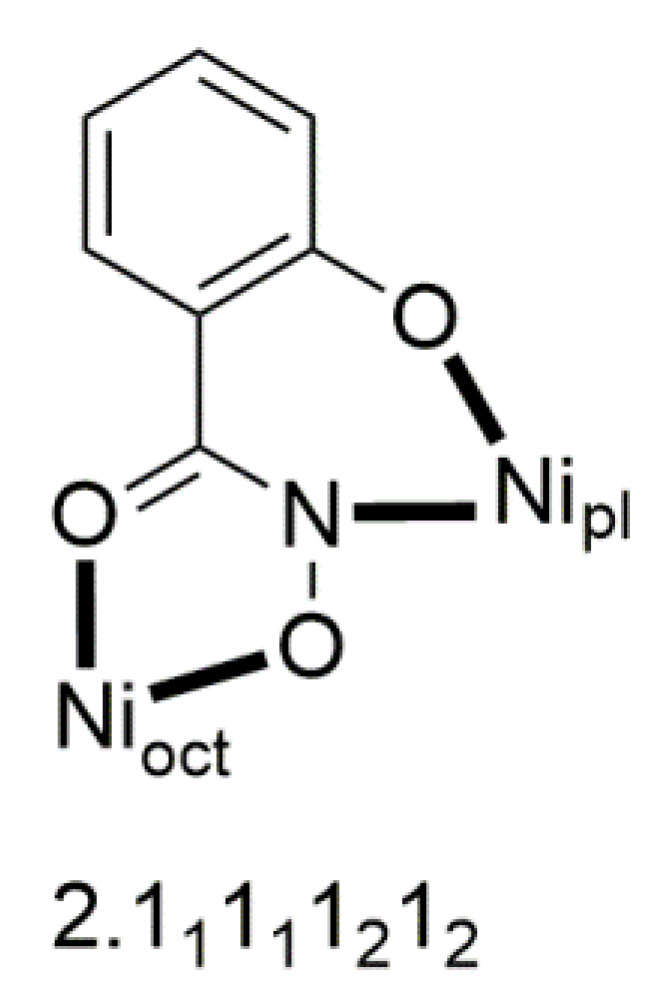
The coordination mode of shi^3−^ in complex [Ni_3_(shi)_2_(dpkoxH)_2_(py)_2_] (**45**) and the Harris notation that describes this ligation. The subscript 1 refers to the octahedral metal ion (Ni1 in Figure 41) and the subscript 2 to the square planar metal ion (Ni2 in Figure 41); pl = planar, oct = octahedral.

In a project aiming at the study of anion coordination by metallomacrocycles, the group of Escuer prepared the trinuclear complex [Ni_3_(N_3_)_4_(Medpt)_2_(dpkox)_2_] (**46**) in good yield from the reaction described in Equation (30), where Medpt is N-methyldipropylenetriamine (Figure 15) [59,60].
3 [Ni_2_(N_3_)_4_(Medpt)_2_] + 4 dpkoxH$+4\;{\mathrm{Et}}_{3}N\;\overset{MeOH}{\to }$2 [Ni_3_(N_3_)_4_(Medpt)_2_(dpkox)_2_] + 4 (Et_3_NH)(N_3_)+ 2 Medpt(30)**Z**
**46**


The molecule (Figure 43) is triangular. The “central” Ni2 ion is bridged to each of Ni1 and Ni3 through one end-on (2.200) azido ligand and one diatomic oximato group from a 2.1110 dpkox^−^ ligand, in such a way that Ni2 has a {Ni^II^(N_azido_)_2_(N_oximato_)_2_(N_2-pyridyl_)_2_} coordination sphere. A tridentate chelating, *mer*-coordinated Medpt ligand and a terminal (1.100) azido group complete an octahedral {Ni^II^ON_5_} coordination environment at each of Ni1 and Ni3 [53,54]. The Ni2∙∙∙Ni1 and Ni2∙∙∙Ni3 exchange interactions are ferromagnetic, promoted by the double oximato/end-on azido bridging units. 

**Figure 43 molecules-30-00791-f043:**
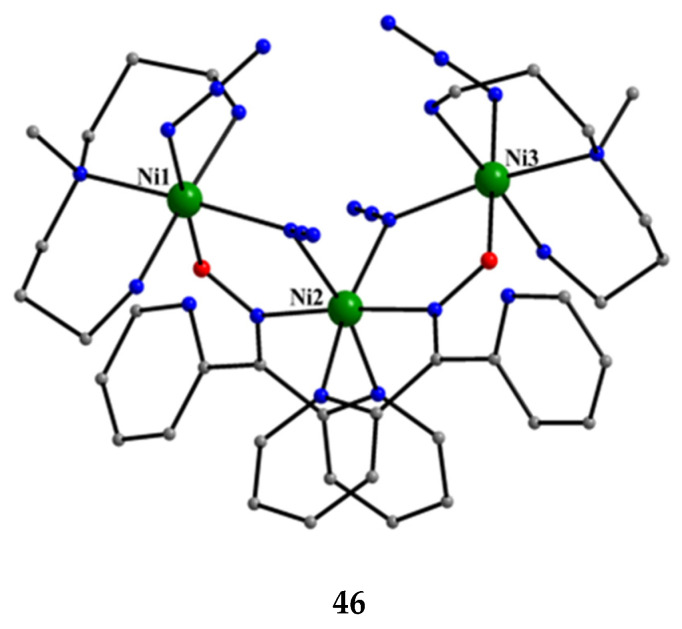
The molecular structure of complex [Ni_3_(N_3_)_4_(Medpt)_2_(dpkox)_2_] (**46**).

Copper(II) forms a small family of hydroxido-bridged triangular complexes with interesting magnetic properties [61,62]; their preparation is represented by Equations (31) and (32), where X = Cl, Br. The ancillary ligands are carboxylates and tert-butylphosphonate(−1); the structural formula of the latter is illustrated in Figure 15. The yields were in the range of 60–70%.
$3\;{\mathrm{Cu}}^{\mathrm{II}}(O_{2}\mathrm{CPh}{)}_{2}\cdot 2H_{2}O+3\;\mathrm{dpkoxH}\;\overset{\mathrm{MeCN}/H_{2}O\;\left(10:1\right),\;T}{\to }$[Cu^II,II,II^_3_(OH)(O_2_CPh)_2_(dpkox)_3_] + 4 PhCO_2_H+ 5 H_2_O(31)
**47**
Cu^II^(OMe)_2_ + 3 dpkoxH + ^t^BuPO_3_H_2_ + NaX + H_2_O $\overset{\mathrm{MeOH}}{\to }$
[Cu^II,II,II^_3_(OH)X(^t^BuPO_3_H) (dpkox)_3_] + NaOMe+ 5 MeOH(32)
**48**, **49**


**Figure 44 molecules-30-00791-f044:**
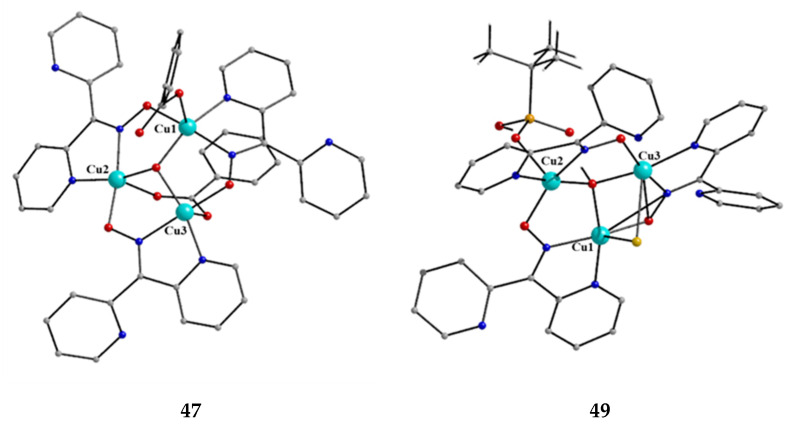
The structures of the molecules [Cu^II,II,II^(OH)(O_2_CPh)_2_(dpkox)_3_] and [Cu^II,II,II^_3_(OH)Br (^t^BuPO_3_H)(dpkox)_3_] that are present in the crystals of **47** and **49**, respectively.

The molecular structure of **47** (Figure 44, left) consists of a near-equilateral copper(II) triangle capped by the oxygen atom of the 3.3 hydroxido (μ_3_-OH^−^) ion. Each edge is bridged by the oximato group of a 2.1110 dpkox^−^ ligand. An edge of the triangle (Cu2∙∙∙Cu3 in Figure 44) is additionally bridged by a 2.11 benzoato ligand, while a monodentate (1.10) PhCO_2_^−^ completes five-coordination at Cu1. The μ_3_-OH^−^ oxygen atom is ~0.6 Å above the plane defined by the three metal ions, which have a distorted square pyramidal geometry; the apical positions are occupied by the three coordinated carboxylato O atoms [61]. The molecular structure of the acetato analog of **47**, [Cu^II,II,II^_3_(OH)(O_2_CMe)_2_(dpkox)_3_] (**47a**; this complex has not been incorporated in Table 3), has a complete similar molecular structure but a different supramolecular motif; the latter can be rationalized in terms of centrosymmetric pairs of trinuclear molecules held together by weak Cu^II^∙∙∙N_unbound 2-pyridyl_ interactions at distances of ~2.8 Å. These interactions generate dimers of trimers, no longer connected to each other.

The molecular structures of **48** and **49** (Figure 44, right) are almost identical [62]. They are similar to the structures of **47** and **47a**, the only differences being the replacement of the bidentate bridging carboxylato group of the PhCO_2_^−^/MeCO_2_^−^ ligands in **47** and **47a** by a bridging 2.2 halido group in **48** and **49**, and the presence of a monodentate ^t^BuPO_3_H^−^ (1.100) ligand in **48** and **49** instead of the monodentate PhCO_2_^−^/MeCO_2_^−^ ligand that is present in the carboxylato complexes. 

Compounds **47**, **47a**, **48,** and **49** can be alternatively described as rare examples of inverse 9-metallacrown-3 complexes [3]. Using metallacrown nomenclature, the formulae of the complexes are {(OH)[inv9-MC_Cu(II)N(dpkox)_-3](O_2_CR)_2_} (R = Ph, Me) and {(OH)[inv9-MC_Cu(II)N(dpkox)_-3](X)(^t^BuPO_3_H)} (R = Cl, Br).

Variable-temperature magnetic susceptibility studies for the complexes and the powder X-band EPR spectrum of **47a** reveal an antiferromagnetically coupled system, also showing intramolecular antisymmetric exchange. 

Interesting dpkox^−^ -based trinuclear complexes were obtained with the 2nd- and 3rd-row transition metals of group 8 (**50**–**52**) [52]. The 1:2 reaction of [Ru_3_(CO)_12_] (**X**) and dpkoxH in refluxing THF gave a mixture of **33** (major product, Table 2) and **50**. Assuming that the complex contains two Ru^I^ and one Ru^0^ atom, we can write Equation (33). The triangular monohydrido complex **51** was prepared by the reaction of [Os_3_(CO)_10_(MeCN)_2_] (**AA**) with dpkoxH in THF at room temperature (yield: 42%), Equation (34). Complex **51** can be used as starting material in the preparation of **52**, Equation (35), which was isolated in low yield (~20%). Equations (34) and (35) were written assuming that two Os atoms have a formal oxidation +I and one has 0.
[Ru_3_(CO)_12_]$+2\;\mathrm{dpkoxH}+½\;O_{2}\;\overset{\mathrm{THF},\;T}{\to }$[Ru_3_(CO)_8_(dpkox)_2_]+ 4 CO + H_2_O       (33)**X**
**50**

[Os_3_(CO)_10_(MeCN)_2_]$+\mathrm{dpkoxH}\;\overset{THF}{\to }$[Os_3_(H)(CO)_9_(dpkox)]+ CO + 2 MeCN(34)**AA**
**51**

[Os_3_(H)(CO)_9_(dpkox)]$+\mathrm{dpkoxH}\;\overset{\mathrm{toluene},\;T}{\to }$[Os_3_(CO)_8_(dpkox)_2_]+ CO + H_2_     (35)**51**
**52**


The structure of **50** (Figure 45, left) consists of triangular molecules. In addition to eight terminal CO groups, the molecule contains two 2.1110 dpkox^−^ ligands which span the same edge of the trimetallic unit (Ru1∙∙∙Ru2), in a head-to-tail arrangement through both the oximato O and N atoms. Each dpkox^−^ ligand is also attached through a 2-pyridyl N atom to one of the metal atoms of the bridged edge, in such a manner that the complex has a non-crystallographic two-fold axis. This symmetry was also indicated by its ^13^C{^1^H} NMR spectrum in d_6_-acetone which shows only four carbonyl signals. The length of the bridged edge (Ru1∙∙∙Ru2 = 3.539 Å) indicates the absence of a metal-metal bond, as expected for a 50-electron trinuclear cluster. The Ru1∙∙∙Ru3 (2.814 Å) and Ru2∙∙∙Ru3 (2.817 Å) bond lengths are indicative of metal-metal bonding [52].

**Figure 45 molecules-30-00791-f045:**
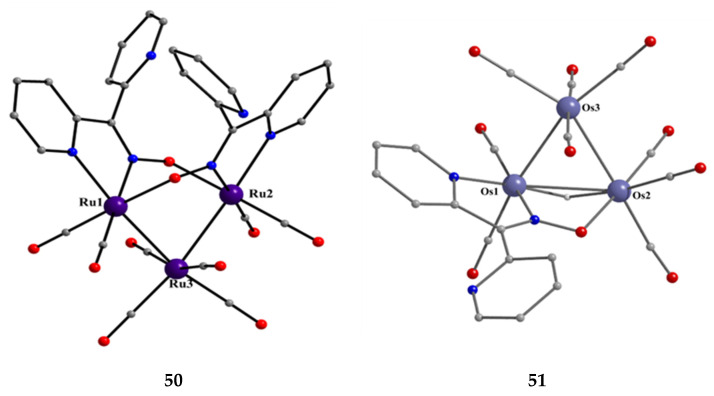
The structures of the molecules [Ru_3_(CO)_8_(dpkox)_2_] and [Os_3_(H)(CO)_9_(dpkox)] that are present in the crystals of **50** and **51**, respectively.

Complex **50** is structurally similar with complexes [Ru_3_(CO)_8_(L-L)_2_ [63], where L-L are various pyridine-alkoxide ligands; those complexes were found to be highly active toward oxidation of a wide range of primary and secondary alcohols to corresponding aldehyde and ketones in the presence of N-methylmorpholine-N-oxide as oxidant.

The structure of **51** consists of a nearly equilateral triangle of Os centers in which two metal atoms (Os1 and Os2) are attached to a 2.1110 dpkox^−^ ligand. A hydrido (H^−^) ion spans the same Os-Os edge as the diatomic NO oximato fragment. The shell of the complex is completed by nine terminal CO groups, and this indicates that the species is a closed-shell 48-electron cluster [52]. The molecular structure of **52** is similar to that of **50** [52]. Complexes **50**–**52** display low activity as DNA cleavage agents, requiring high complex concentrations, long incubation times, and the use of UV light as a trigger. 

The deprotonated di-2-pyridyl ketoxime has been successfully used for the synthesis of linear {M^II^Ln^III^_2_} clusters (M = Ni, Pd, Cu; Ln = lanthanoid) [64,65,66,67,68], some of which exhibit interesting magnetic properties. In the last 20 years or so, there has been an intense research activity in the chemistry of 3d/4f-metal coordination clusters. The reason is that such complexes display fascinating properties (magnetic, optical, catalytic, …) and often a combination of properties arising from the simultaneous presence of two completely different metal ions [16]. The synthesis of 3d/4f-metal clusters is not an easy task. Simple reactions of the 3d- and 4f-metal starting materials often give pure 3d- or 4f-metal compounds depending on the donor atoms of the ligand. Based on the “hard and soft acids and bases” (HSAB) model, an often-used strategy is the “metal complexes as ligands” or “metalloligand” approach. In most cases, the metalloligands are mononuclear divalent 3d-metal ion complexes (the 3d-metal ion is an “intermediate” or even “soft” acid) with uncoordinated (free) O-sites, which can easily further react with the oxophilic (“hard” acids) Ln^III^ ions providing access to mixed 3d/4f-metal species. Simple 2-pyridyl oximes (Figure 3, left) are ideal platforms for the synthesis of such complexes [16]. When deprotonated, these anionic ligands possess the 2-pyridyl N atom in a position that offers the possibility of formation of a stable five-membered chelating ring, also involving the oximato N atom, with the divalent 3d metal. Thus, the resulting complexes are efficient metalloligands, which can further react with the 4f-metal ion through their deprotonated oximato O atoms. The presence of an extra 2-pyridyl ring in dpkox^−^ could, in principle, enable the formation of a second chelating ring (6-membered this time) with the Ln^III^ ion involving the oximato O atom and the second 2-pyridyl N atom. Based on the HSAB model, the expected coordination mode of dpkox^−^ is the 2.1_2_1_1_1_1_1_2_ one illustrated in Figure 14, where subscript 1 refers to M^II^ and subscript 2 to Ln^III^. Of course, the possibility of bridging the oximato O atom to a second Ln^III^ center (3.2_2_1_1_1_1_1_2_ ligation mode) cannot be ruled out.

Ishida and co-workers prepared {NiLn_2_} and {MLn_2_} clusters ((M = Pd^II^, Cu^II^) [64,65,66,67,68], with a few of them exhibiting exciting magnetic properties. Since the complexes have similar molecular structures, we list some (but not all) in Table 4. For their preparation, the metalloligands **BB**, **CC,** and **DD**, shown in Figure 46, were designed and prepared. The preparation of the clusters is represented by Equations (36)–(38); the yields were moderate to good. The ligand hfac^−^ is the ancillary hexafluoroacetylacetonato(−1) group, py is pyridine, and phen is the bidentate chelating ligand 1,10-phenathroline.
[Ni(dpkox)_2_(phen)]$+2\;[\mathrm{Ln}(\mathrm{hfac}{)}_{3}(H_{2}O{)}_{2}]\;\overset{{\mathrm{Et}}_{2}O/{\mathrm{CH}}_{2}{\mathrm{Cl}}_{2}}{\to }$[NiLn_2_(hfac)_6_(dpkox)_2_(phen)]+ 4 H_2_O(36)**BB**
**53**–**57**

   [Ni(dpkox)_2_(py)_2_]$+2\;[\mathrm{Ln}(\mathrm{hfac}{)}_{3}(H_{2}O{)}_{2}]\;\overset{Et_{2}O/CH_{2}Cl_{2}}{\to }$[NiLn_2_(hfac)_6_(dpkox)_2_(py)_2_]+ 4 H_2_O  (37)**CC**
**57**–**60**

              [M(dpkox)_2_]$+2\;[\mathrm{Ln}(\mathrm{hfac}{)}_{3}(H_{2}O{)}_{2}]\;\overset{n-heptane}{\to }$[MLn_2_(hfac)_6_(dpkox)_2_]+ 4 H_2_O              (38)            **DD**

**61**–**63**


**Figure 46 molecules-30-00791-f046:**
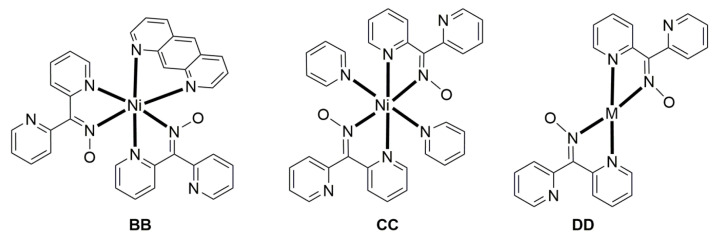
The metalloligands used for the synthesis of {NiLn_2_} and {M^II^Ln_2_} (M = Pd, Cu) complexes based on dpkox^−^.

**Table 4 molecules-30-00791-t004:** Heterotrinuclear (heterotrimetallic) complexes of dpkoxH and dpkox^−^.

Complex ^a^	Coordination Mode of dpkoxH/dpkox^− d^	Ref.
[NiTb_2_(hfac)_6_(dpkox)_2_(phen)] (**53**) ^b^	2.1_2_1_1_1_1_1_2_	[64]
[NiDy_2_(hfac)_6_(dpkox)_2_(phen)] (**54**) ^b^	2.1_2_1_1_1_1_1_2_	[64]
[NiHo_2_(hfac)_6_(dpkox)_2_(phen)] (**55**) ^b^	2.1_2_1_1_1_1_1_2_	[64]
[NiEr_2_(hfac)_6_(dpkox)_2_(phen)] (**56**) ^b^	2.1_2_1_1_1_1_1_2_	[64]
[NiGd_2_(hfac)_6_(dpkox)_2_(py)_2_] (**57**) ^c^	2.1_2_1_1_1_1_1_2_	[65]
[NiTb_2_(hfac)_6_(dpkox)_2_(py)_2_] (**58**) ^c^	2.1_2_1_1_1_1_1_2_	[65]
[NiDy_2_(hfac)_6_(dpkox)_2_(py)_2_] (**59**) ^c^	2.1_2_1_1_1_1_1_2_	[64]
[NiHo_2_(hfac)_6_(dpkox)_2_(py)_2_] (**60**) ^c^	2.1_2_1_1_1_1_1_2_	[64,65]
[Pd^II^Dy_2_(hfac)_6_(dpkox)_2_] (**61**)	2.1_2_1_1_1_1_1_2_	[66]
[Cu^II^Dy_2_(hfac)_6_(dpkox)_2_] (**62**)	2.1_2_1_1_1_1_1_2_	[67]
[Cu^II^Gd_2_(hfac)_6_(dpkox)_2_] (**63**)	2.1_2_1_1_1_1_1_2_	[68]

^a^ hfac is the hexafluoroacetylacetonato(−1) ligand. ^b^ phen = 1,10-phenanthroline. ^c^ py = pyridine. ^d^ Using the Harris notation; subscript 2 refers to the lanthanoid ion, and subscript 1 to the transition metal ion; see Figure 14.

**Figure 47 molecules-30-00791-f047:**
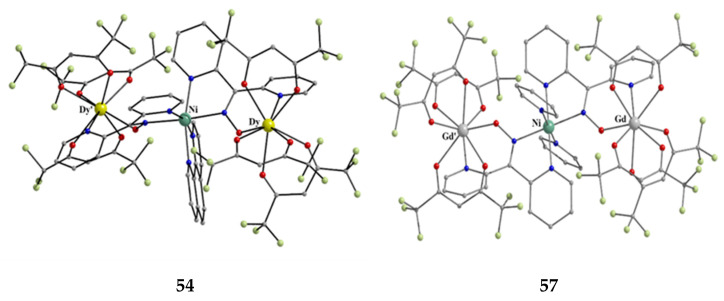
The molecular structures of [NiDy_2_(hfac)_6_(dpkox)_2_(phen)] (**54**) and [NiGd_2_(hfac)_6_(dpkox)_2_(py)_2_] (**57**).

**Figure 48 molecules-30-00791-f048:**
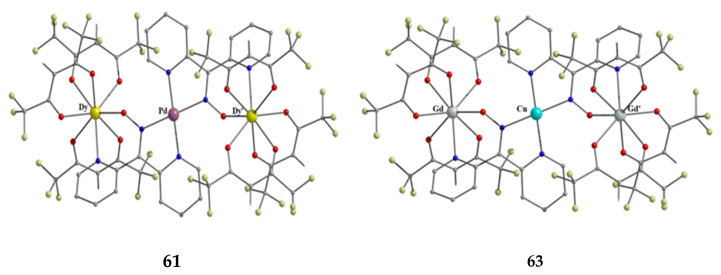
The molecular structures of [Pd^II^Dy_2_(hfac)_6_(dpkox)_2_] (**61**) and [Cu^II^Gd_2_(hfac)_6_(dpkox)_2_] (**63**).

The molecular structures of **54**, **57**, **61,** and **63** are shown in Figure 47 and Figure 48. Complexes **53**–**56** have completely similar molecular structures [64]. The precursor **BB** has been incorporated in the center of the molecule; the only difference is that the coordination mode of dpkox^−^ has changed from 1.0110 in the former to 2.1_2_1_1_1_1_1_2_ (Figure 14) in the latter, where the subscript 2 refers to Ln^III^ and 1 to Ni^II^. The molecules have a two-fold crystallographic axis that passes from the center of phen and Ni^II^. Due to the molecular symmetry, the three metal ions are arranged in a V-type manner, but actually the Ln^III^∙∙∙Ni^II^∙∙∙Ln^III^ array is close to linear (~177°). Three chelating hfac^−^ groups, and one O and one N atoms from the same dpkox^−^ complete eight-coordination at each 4f-metal ion. Thus, the coordination spheres are {Ni^II^N_6_} and {Ln^III^O_7_N}. 

The molecular structures of **57**–**60** [64,65] are also similar. The molecules have a crystallographically imposed inversion center at Ni^II^ and thus the topology is strictly linear. Two py ligands are *trans* in the {Ni^II^N_6_} coordination sphere. The coordination mode of dpkox^−^ ligands is identical to that in **53**–**56** and the peripheral ligation around the Ln^III^ centers is the same, i.e., three chelating hfac^−^ groups.

The molecular structures of the {MLn_2_} complexes (M = Cu^II^, Pd^II^) [66,67,68] are similar to those of **57**–**60**, the only difference being the absence of the two py molecules from the central, square planar transition metal ions. 

Some of the complexes listed in Table 4 (and a few similar ones that are not listed) exhibit interesting magnetic properties. Selected features are illustrated in Figure 49, Figure 50, Figure 51, Figure 52 and Figure 53. Among others: (a) Complexes **54** and **58** exhibit a temperature and frequency dependence of the out-of-phase molar magnetic alternating current (ac) susceptibility, obtained at a 5 G (ac) field and zero direct current (dc) field (Figure 49), suggesting that they are SMMs [64]. (b) High-Frequency EPR (HF-EPR) studies made possible the determination of the Ni^II^∙∙∙Ln^III^ exchange coupling in **57**–**60** and its chemical trend (Figure 50). In contrast to the antiferromagnetic {NiDy_2_} complex **59**, ferromagnetic couplings were precisely determined for **57**, **58,** and **60** [58]. (c) Magnetization studies at 0.4 K for **61** (Figure 51) indicate that this complex is SMM [66]; in addition, the diamagnetism of Pd^II^ was proven indispensable to clarify the contribution of the Ln^III^∙∙∙Ln^III^ exchange coupling in the magnetism of the isomorphous complex **62**. (d) Complex **62** is SMM (Figure 52) [67]. (e) HF-EPR spectra (Figure 53) and magnetization studies led to the conclusion that **63**, and its isomorphous Tb(III) and Ho(III) analogs [68] are characterized by ferromagnetic Cu^II^∙∙∙Ln^III^ exchange interactions. 

**Figure 49 molecules-30-00791-f049:**
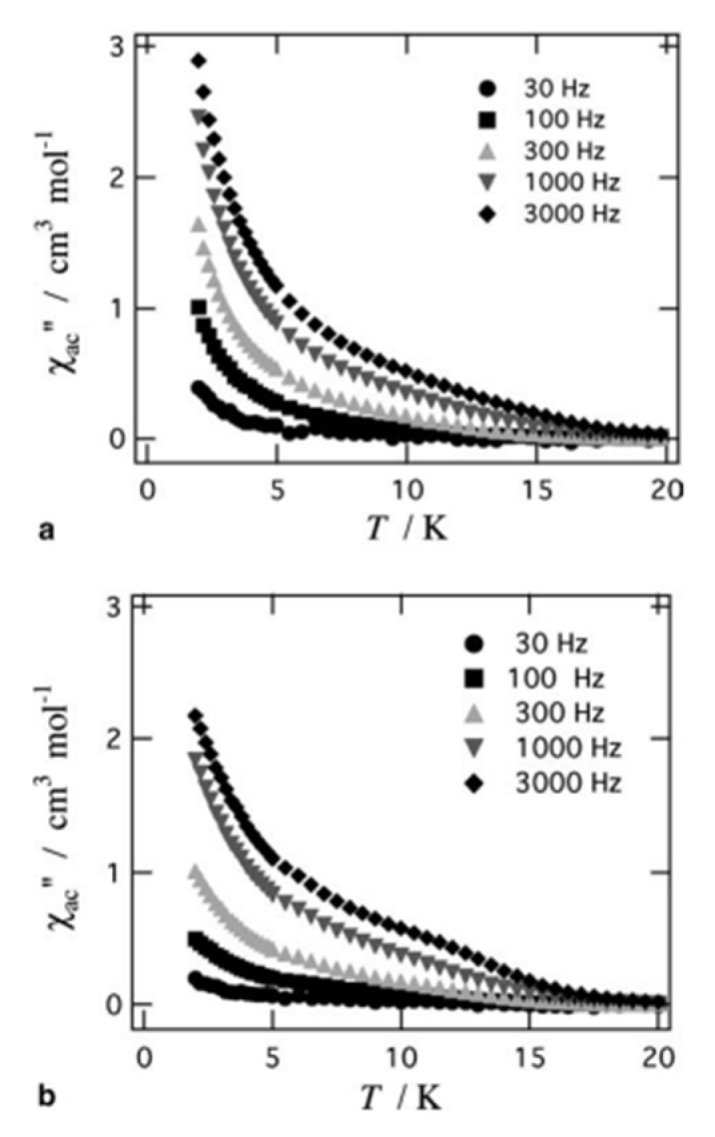
Temperature and frequency dependence of the out-of-phase molar magnetic alternating current (ac) susceptibility, χ^ΙΙ^_ac_, for complexes [NiDy_2_(hfac)_6_(dpkox)_2_(phen)] (**54**) (**a**) and [NiDy_2_(hfac)_6_(dpkox)_2_(py)_2_] (**59**) (**b**). Reproduced from Ref. [64]. Copyright 2005 Elsevier.

**Figure 50 molecules-30-00791-f050:**
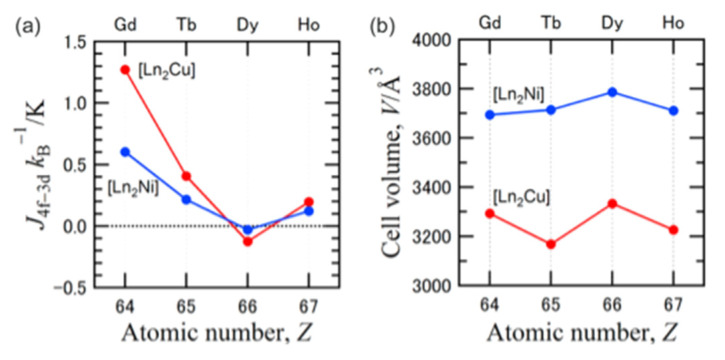
(**a**) Plot of the 3d-4f exchange parameters (*J*) in the {NiLn_2_} clusters **57**–**60** and the {Cu^II^Ln_2_} complexes **62**, **63**, [Cu^II^Tb_2_(hfac)_6_(dpkox)_2_] (not listed in Table 4) and [Cu^II^Ho_2_(hfac)_6_(dpkox)_2_] (also not listed in Table 4) as a function of the atomic number, *Z*. (**b**) Plot of the cell volume in the above-mentioned complexes as a function of *Z*. The {Cu^II^Tb_2_} and {Cu^II^Ho_2_} clusters are isomorphous with **62** and **63**. Reproduced from Ref. [65]. Copyright 2013 American Chemical Society.

**Figure 51 molecules-30-00791-f051:**
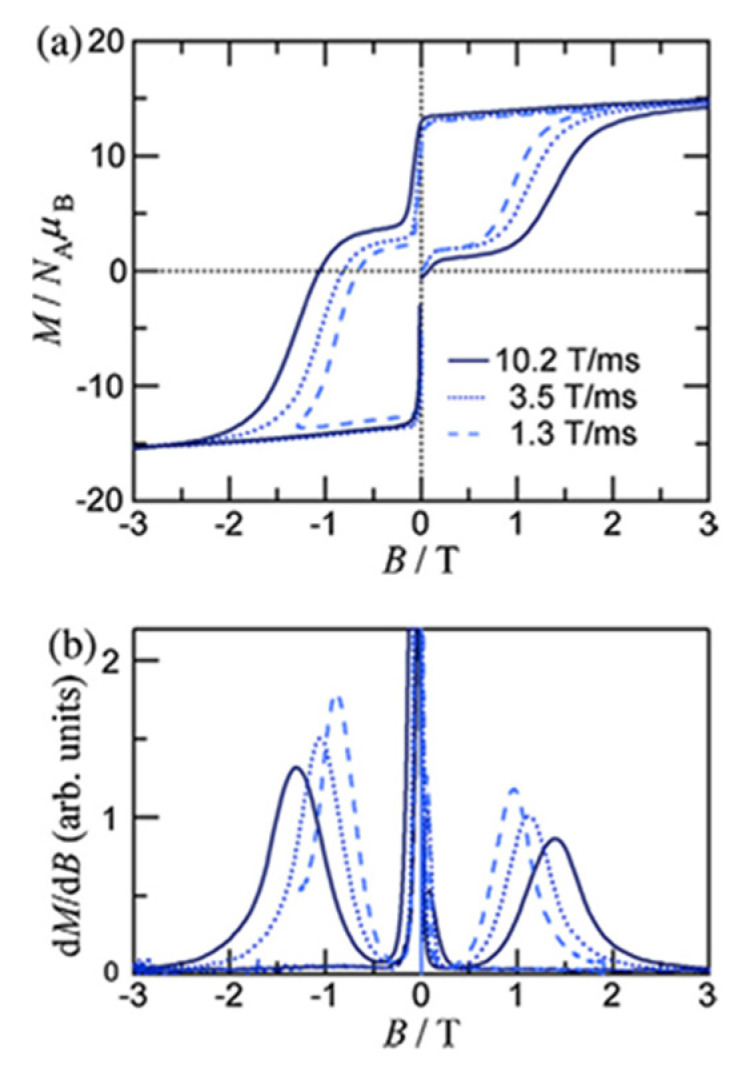
(**a**) Magnetization (*M*) curves and (**b**) their derivatives for complex [PdDy_2_(hfac)_6_(dpkox)_2_] (**61**) measured at 0.4 K using a pulse-field magnetometer. Reproduced from Ref. [66]. Copyright 2011 Elsevier.

**Figure 52 molecules-30-00791-f052:**
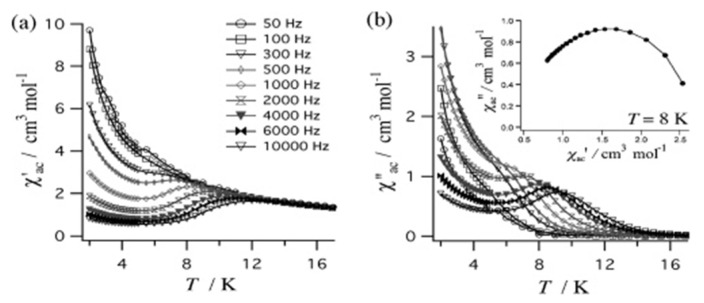
Frequency and temperature dependence of the ac molar magnetic susceptibility for [Cu^II^Dy_2_(hfac)_6_(dpkox)_2_] (**62**). (**a**) χ’_ac_ is in-phase part; (**b**) χ’’_ac_ is the out-of-phase part. The inset shows the Cole-Cole plot at 8 K. Reproduced from Ref. [67]. Copyright 2006 American Chemical Society.

**Figure 53 molecules-30-00791-f053:**
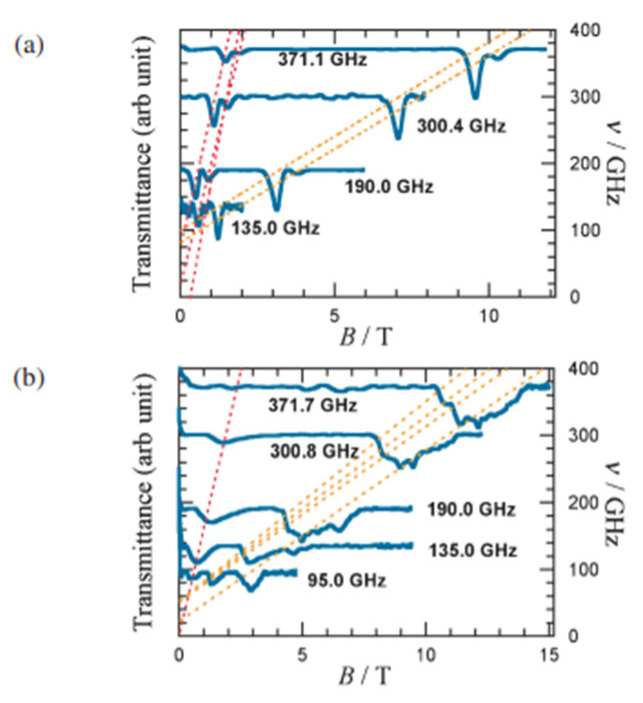
Selected High-Frequency EPR (HF-EPR) spectra at 4.2 K of complexes [Cu^II^Tb_2_(hfac)_6_(dpkox)_2_] (**a**) and [Cu^II^Ho_2_(hfac)_6_(dpkox)_2_] (**b**), not listed in Table 4; the two clusters are isomorphous with **62** and **63**. The spectra are offset in a linear scale of the frequency. Dotted lines are drawn from linear fitting in the frequency vs. field plot. Reproduced from Ref. [68]. Copyright 2010 The Chemical Society of Japan.

### 5.5. Tetranuclear Clusters

The largest family of dpkoxH/dpkox^−^-based clusters consists of tetranuclear compounds (**64**–**90**, Table 5). All these clusters contain deprotonated di-2-pyridyl ketoxime, which, in most cases, favors high nuclearity.

The interesting mixed-valence cluster **64** [69] was obtained by the reaction shown in Equation (39) in good yield (~60%); 3,4 D^−^ is the 3,4-dichlorophenoxyacetate(−1) anion.
4 Mn^II^Cl_2_∙4H_2_O + 4 Na(dpkox) + 4 Na(3,4-D) + ½ O_2_ $\overset{\mathrm{MeOH}}{\to }$
[Mn^II,II,II^_3_Mn^IV^O(3,4-D)_4_(dpkox)_4_] + 8 NaCl+ 16 H_2_O(39)
**64**


**Figure 54 molecules-30-00791-f054:**
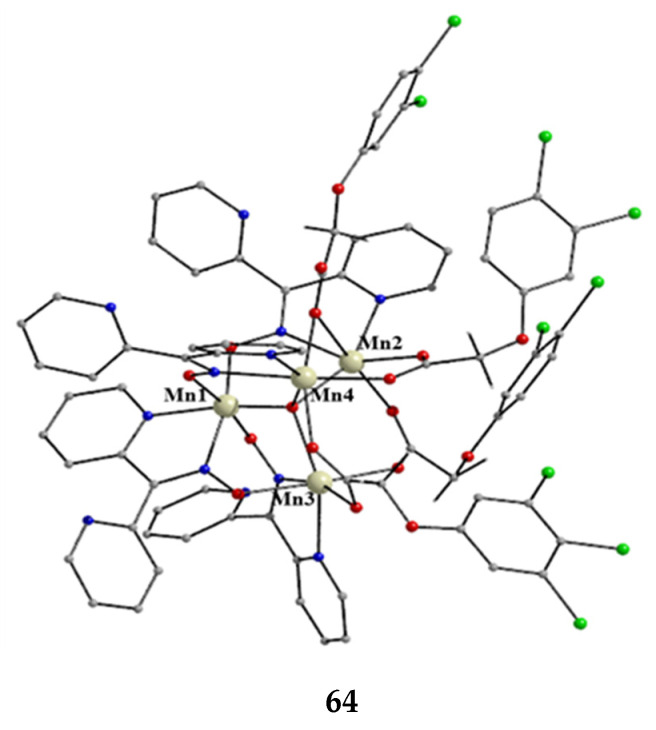
The molecular structure of [Mn^II,II,II^_3_Mn^IV^O(3,4-D)_4_(dpkox)_4_] (**64**). Only the H atoms of the –CH_2_- groups are shown. The three chlorine atoms in one of the 3,4-D^−^ ligands are a consequence of a crystallographic disorder issue.

The central core of the cluster is {Mn_4_(μ_4_-O)}^8+^ in which the octahedral Mn ions form a distorted tetrahedron centered on the oxido group [69]. The four oximato (=NO^−^) groups of three 2.1_1_1_2_1_2_0 and one 2.1_2_1_1_1_1_0 dpkox^−^ ligands link the Mn^IV^ atom (Mn1) with the Mn^II^ atoms; the latter are connected by three 2.110 and one 2.100 3,4-D^−^ groups. In the Harris notation that is used to describe the dpkox^−^ ligation modes, the subscript 1 refers to Mn^IV^ and 2 to Mn^II^. The coordination spheres are thus {Mn^IV^(O_oximato)3_O_oxido_N_oximato_N_2-pyridyl_}, {Mn^II^(O_carboxylato)3_O_oxido_N_oximato_N_2-pyridyl_} (for Mn2 and Mn4) and {Mn^II^(O_carboxylato)2_O_oximato_O_oxido_N_oximato_N_2-pyridyl_} (for Mn3). The structure of the complex is shown in Figure 54. Magnetically, there are both ferromagnetic and antiferromagnetic exchange interactions within the molecule propagated through Mn^IV^∙∙∙Mn^II^ and Mn^II^∙∙∙Mn^II^ pathways, respectively. Magnetization data at 2.5 and 4.5 K in the field range 0–6.5 T support an *S* = 6 ground state for the complex with *g* = 2.0 and a small zero-field splitting *D* = 0.025 cm^−1^ [69].

Complexes **65** [70] and **66** [71] have molecular structures similar to the structure of **64** [69]. The only difference is the nature of the carboxylato ligands; these are the 2,4,5-trichlorophenoxyacetate(−1) [Figure 15] in **65** and 2,3-dichlorophenoxyacetate(−1) [Figure 15] in **66**. The magnetic properties of the latter clearly indicate an *S* = 6 ground state (like **64**) [71]. Spectroscopic titration studies with calf thymus DNA suggest binding of **65** to the DNA helix, with a binding constant K_b_ equal to 1.1x10^−4^ M^−1^ [70]. Competitive binding studies with ethidium bromide (EthBr) showed that the interaction between DNA and **65** releases EthBr from its DNA compound, indicating that the Mn(II,II,II,IV) compound binds to DNA via intercalation mode. Additionally, DNA electrophoretic mobility experiments reveal that the complex, at low concentration, is obviously capable of binding to pDNA causing its cleavage at physiological pH and room temperature.

The simultaneous incorporation of dpkox^−^, X^−,^ and RCO_2_^−^ ligands in manganese complexes give mixed-valence tetranuclear clusters (**67**–**69**) in variable yields with interesting structures [24,72], where R = Me, Ph, X^−^ = Cl^−^, Br^−,^ and (py)_2_C(O)_2_^2−^ is the dianion of the *gem*-diol form of di-2-pyridyl ketone, (py)_2_CO; see Figure 15. The (py)_2_C(O)_2_^2−^ ligand is the product of the metal ion-assisted/promoted transformation of an amount of dpkoxH, Equations (40)–(42). For a better understanding of the simplified balanced Equations (40) and (43)–(48), dpkoxH is abbreviated as (py)_2_CNOH, where py stands for the 2-pyridyl group and 
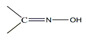
 is the oxime group. Note that the *gem*-diol form, (py)_2_C(OH)_2_, and its anions (py)_2_C(OH)(O)^−^ and (py)_2_C(O)_2_^2−^, do not exist free but only attached to metal ions, i.e., as ligands.
        (py)_2_CNOH + H_2_O $\overset{\mathrm{metal}\;\mathrm{ion}}{\to }$ (py)_2_CO + H_2_NOH(40)        (py)_2_CO + H_2_O $\overset{\mathrm{metal}\;\mathrm{ion}}{\to }$ (py)_2_C(OH)_2_(41)(py)_2_C(OH)_2_ $\overset{base}{\to }$ (py)_2_C(OH)(O)^−^/(py)_2_C(O)_2_^2−^ + 1H^+^/2H^+^(42)

Complexes **67**–**69** were prepared by several methods outlined in Equations (43)–(49).
6 Mn^II^(O_2_CPh)_2_∙2H_2_O + 5 Mn^II^X_2_ + (Bu^n^_4_N)Mn^VII^O_4_ + 12 (py)_2_CNOH + ¼ O_2_ $\overset{\mathrm{EtOH}/\mathrm{MeCN}}{\to }$ (Bu^n^_4_N)(O_2_CPh) +       (43)[Mn^II,II^_2_Mn^III,III^_2_X_2_(O_2_CPh)_2_{(py)_2_CNO}_2_{(py)_2_C(O)_2_}_2_] + 3 PhCO_2_H + 2 (H_3_NOH)(O_2_CPh) + 4 (H_3_NOH)X + 4.5 H_2_O                                         **67**, **69**

Mn^II,III,III^_3_O(O_2_CPh)_6_(py)_2_(H_2_O)] + Mn^II^X_2_ + 4 (py)_2_CNOH + 3 H_2_O $\overset{\mathrm{MeCN}}{\to }$ 2 PhCO_2_H + 2 py +      (44)                                 **EE**                              [Mn^II,II^_2_Mn^III,III^_2_X_2_(O_2_CPh)_2_{(py)_2_CNO}_2_{(py)_2_C(O)_2_}_2_] + 2 (H_3_NOH)(O_2_CPh)                                                                                                                                        **67**, **69**


(Bu^n^_4_N)[Mn^III^_4_O_2_(O_2_CPh)_9_(H_2_O)] + 4 Mn^II^X_2_ + 8 (py)_2_CNOH + 5 H_2_O $\overset{MeCN/CH_{2}Cl_{2}}{\to }$       (45)                             **FF**
                      2 [Mn^II,II^_2_Mn^III,III^_2_X_2_(O_2_CPh)_2_{(py)_2_CNO}_2_{(py)_2_C(O)_2_}_2_] + 4 (H_3_NOH)X + (Bu^n^_4_N)(O_2_CPh) + 4PhCO_2_H                                                   **67**, **69**

2 Mn^II^(O_2_CMe)_2_∙4H_2_O + 2 Mn^II^Br_2_ + 4 (py)_2_CNOH + ½ O_2_ $\overset{\mathrm{MeCN}}{\to }$ [Mn^II,II^_2_Mn^III,III^_2_Br_2_(O_2_CMe)_2_{(py)_2_CNO}_2_{(py)_2_C(O)_2_}_2_](46)                                                                                                                                                              **68**+ 2 MeCO_2_H + 2 (H_3_NOH)Br + 5 H_2_O

2 [Mn^III^_3_O(O_2_CMe)_6_(py)_3_](ClO_4_) + 6 Mn^II^Br_2_ + 12 (py)_2_CNOH + 10 H_2_O $\overset{\mathrm{MeCN}}{\to }$
        (47)                            **GG**
        3 [Mn^II,II^_2_Mn^III,III^_2_Br_2_(O_2_CMe)_2_{(py)_2_CNO}_2_{(py)_2_C(O)_2_}_2_] + 6 MeCO_2_H + 6 (H_3_NOH)Br + 2 (pyH)(ClO_4_) + 4 py                                                    **68**


[Mn^II,III,III^_3_O(O_2_CMe)_6_(py)_2_] + Mn^II^Br_2_ + 4 (py)_2_CNOH + 3 H_2_O $\overset{\mathrm{MeCN}}{\to }$
                                    (48)                              **HH**
[Mn^II,II^_2_Mn^III,III^_2_Br_2_(O_2_CMe)_2_{(py)_2_CNO}_2_{(py)_2_C(O)_2_}_2_] + 2 MeCO_2_H + 2 (H_3_NOH)(O_2_CMe) + 2 py                                                **68**


The 1:1 reaction between Mn^II^(O_2_CMe)_2_∙4H_2_O and (py)_2_CNOH (dpkoxH) in MeCN resulted in an orange solution which upon standing undisturbed turned dark brown (due to the oxidation of Mn^II^ under aerobic conditions); slow evaporation of the solution at room temperature gave dark brown crystals of [Mn^II,II^_2_Mn^III,III^_2_(NO_3_)_2_{(py)_2_CNO}_2_{(py)_2_C(O)_2_}_2_] (**70**) in good yield (~70%). Single-crystal X-ray structural solution surprisingly revealed the presence of nitrato groups in the complex, although there were no nitrates in the reactants. We proposed a detailed mechanism for the generation of NO_3_^−^s [24] which is based on the oxidation of H_2_NOH, produced from the metal ion-assisted hydrolysis of (py)_2_CNOH to (py)_2_C(O)_2_^2−^, to form nitrates. A simplified reaction for this experimental observation is shown in Equation (49). The characterization of **70** proves the non-critical role of Cl^−^ or Br^−^ in the (py)_2_CNOH → (py)_2_C(O)_2_^2−^ transformation. We proposed [24] that the reason why no NO_3_^−^ ligands were not coordinated in **67**–**69** is the presence of Cl^−^ or Br^−^ ions which are ligated to Mn^II^ ions, blocking the available sites for nitrato coordination.
Mn^II^(O_2_CMe)_2_∙4H_2_O + 4 (py)_2_CNOH + 3 O_2_ $\overset{MeCN}{\to }$
[Mn^II,II^_2_Mn^III,III^_2_(NO_3_)_2_{(py)_2_CNO}_2_{(py)_2_C(O)_2_}_2_] + 8 MeCO_2_H+ 14 H_2_O(49)
**70**


The molecular structures of **67**–**70** [24,72] are similar (Figure 55) and differ only in the nature of the carboxylato ligand (MeCO_2_^−^, PhCO_2_^−^) and the terminal inorganic anion (Cl^−^, Br^−^, NO_3_^−^). The bridging system comprises two 3.2_2_2_1,2_1_2_1_1_ (Figure 56) (py)_2_C(O)_2_^2−^ (the subscript 2 refers to Mn^III^ and 1 to Mn^II^) ligands, two 2.1110 (py)_2_CNO^−^(dpkox^−^) ligands (a more clear description is 2.1_2_1_1_1_1_0) and two 2.1_1_1_2_ PhCO_2_^−^ groups. Peripheral ligation is completed by two terminal halido (**67**–**69**) or two bidentate nitrato groups (**70**). The brief description of the core is {Mn_4_(μ_2_-OR’)_4_}^6+^, where the four bridging atoms belong to two (py)_2_C(O)_2_^2—^ligands. As might be expected, in **67**–**69** the Mn^III^ atoms are bound to the hard (HSAB) O_5_N donor set, while the Mn^II^ atoms to the less hard O_2_N_3_X (X = Cl, Br) set; the metal geometries are octahedral. As a result of the bidentate character of the terminal nitrato groups, the Mn^II^ atoms in **70** are seven-coordinate with a distorted pentagonal bipyramidal {Mn^II^O_4_N_3_} coordination sphere.

**Figure 55 molecules-30-00791-f055:**
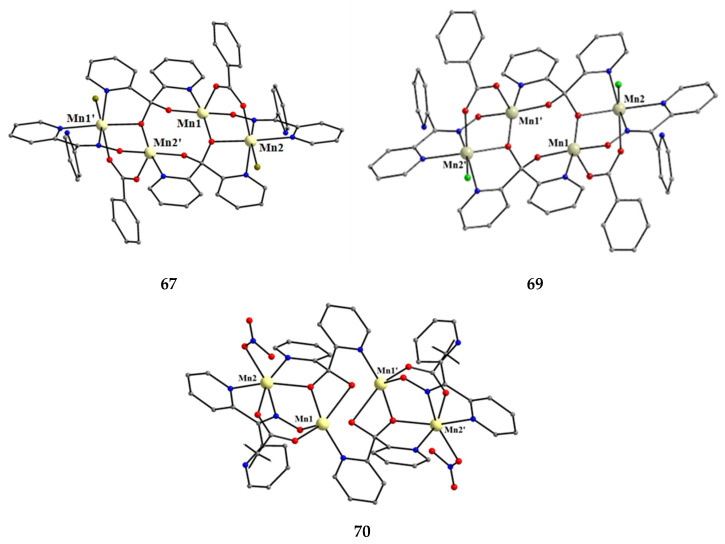
The molecular structure of [Mn^II,II^_2_Mn^III,III^_2_Br_2_(O_2_CPh)_2_(dpkox)_2_{(py)_2_C(O)_2_}_2_] (**67**), [Mn^II,II^_2_Mn^III,III^_2_Cl_2_(O_2_CPh)_2_(dpkox)_2_{(py)_2_C(O)_2_}_2_] (**69**) and [Mn^II,II^_2_Mn^III,III^_2_(NO_3_)_2_(O_2_CMe)_2_ (dpkox)_2_{(py)_2_C(O)_2_}_2_] (**70**).

**Figure 56 molecules-30-00791-f056:**
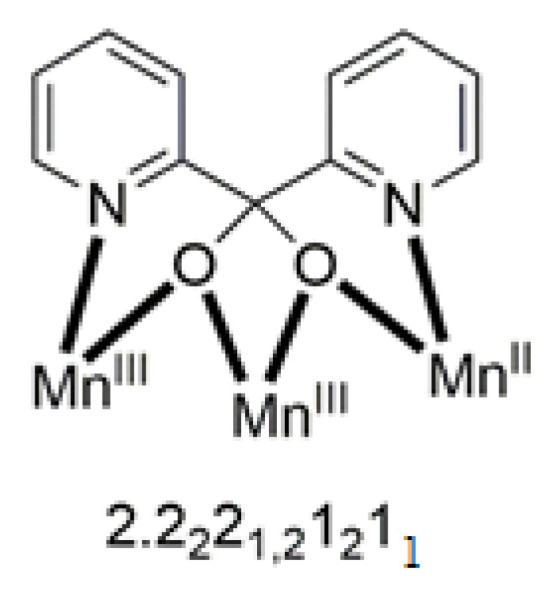
The coordination mode of (py)_2_C(O)_2_^−^ in complexes [Mn^II,II^_2_Mn^III,III^_2_Y_2_(O_2_CR)_2_(dpkox)_2_{(py)_2_C(O)_2_}_2_] and the detailed Harris notation that describes this mode. The subscript 2 refers to Mn^III^ and 1 to Mn^II^. R = Ph in **67** and **69**, and Me in **68** and **70**; Y = Br in **67** and **68**, Cl in **69,** and NO_3_ in **70**.

Variable-temperature magnetic susceptibility studies in the 2–300 K range for the representative complexes **67** and **68** reveal weak antiferromagnetic exchange Mn^II^∙∙∙Mn^III^ and Mn^III^∙∙∙Mn^III^ interactions, leading to non-magnetic *S* = 0 ground states [24,72].

The use of dpkoxH in iron(III) acetate chemistry provided access to two tetranuclear clusters (**71**, **72**) which have two different lattice solvent sets (2CH_2_Cl_2_∙H_2_O in **71** and 4.5MeNO_2_ in **72**), Equations (50) and (51); the yields were 60 and 45% for **71** and **72**, respectively [73]. The presence of N_3_^−^ ions in the reaction mixtures afforded complex **73** in typical yields in the range of 60–70%. This complex can be alternatively synthesized by the reaction of **71** with N_3_^−^; for the synthetic processes, see Equations (52)–(54).
4 Fe^III^Cl_3_∙6H_2_O + 4 dpkoxH+ 8 Na(O_2_CMe)∙3H_2_O $\overset{\mathrm{MeCN}/{\mathrm{CH}}_{2}{\mathrm{Cl}}_{2}}{\to }$

                         (50)
[Fe^III^_4_O_2_Cl_2_(O_2_CMe)_2_(dpkox)_4_]+ 8 NaCl + 6 MeCO_2_H + 2 HCl + 46 H_2_O
**71**


6 [Fe^III^_3_O(O_2_CMe)_6_(H_2_O)_3_]Cl+ 12 dpkoxH $\overset{{\mathrm{CH}}_{2}{\mathrm{Cl}}_{2}/{\mathrm{MeNO}}_{2}}{\to }$
3 [Fe^III^_4_O_2_Cl_2_(O_2_CMe)_2_(dpkox)_4_] +
(51)
**II**

2 “{Fe^III^_3_O(O_2_CMe)_6_(H_2_O)_2_(OH)}”+ 18 MeCO_2_H + 10 H_2_O


**72**



4 Fe^III^Cl_3_∙6H_2_O + 4 dpkoxH + 8 Na(O_2_CMe)∙3H_2_O + 2 NaN_3_ $\overset{\mathrm{MeCN}}{\to }$
      (52)                                                                                   [Fe^III^_4_O_2_(N_3_)_2_(O_2_CMe)_2_(dpkox)_4_] + 10 NaCl + 6 MeCO_2_H + 46 H_2_O                                                                                                                  **73**


6 [Fe^III^_3_O(O_2_CMe)_6_(H_2_O)_3_]Cl + 12 dpkoxH + 6 NaN_3_ $\overset{\mathrm{MeCN}}{\to }$
           (53)                            **ΙΙ**
            3 [Fe^III^_4_O_2_(N_3_)_2_(O_2_CMe)_2_(dpkox)_4_] + 2 “{Fe^III^_3_O(O_2_CMe)_6_(H_2_O)_2_(OH)}” + 6 NaCl + 18 MeCO_2_H + 10 H_2_O                                **73**


[Fe^III^_4_O_2_Cl_2_(O_2_CMe)_2_(dpkox)_4_]+ 2 NaN_3_ $\overset{{\mathrm{CH}}_{2}{\mathrm{Cl}}_{2}}{\to }$
[Fe^III^_4_O_2_(N_3_)_2_(O_2_CMe)_2_(dpkox)_4_]+ 2 NaCl                                          (54)
**71**


**73**



The structures of the tetranuclear molecules that are present in the crystal structures of **71** and **72** are almost identical and very similar to the structure of the molecule [Fe^III^_4_O_2_(N_3_)_2_(O_2_CMe)_2_(dpkox)_4_] in **73** [73]; thus, a common description is given. The structures of **71** and **73** are shown in Figure 57. The tetranuclear molecules contain the {Fe^III^_4_(μ_3_-O)_2_}^8+^ core comprising four Fe^III^ centers in a “butterfly” disposition and two triply bridging (μ_3_) oxido (O^2−^) ions. Ions Fe2 and Fe3 occupy the “body” sites, and Fe1 and Fe4 occupy the ”wingtip” sites. The four Fe^III^ atoms are essentially coplanar and the two μ_3_-O^2−^ ions are above and below the Fe^III^_4_ plane (ca. 0.5 Å). This is also reflected in the sums of the Fe-O-Fe angles around the μ_3_-O^2−^ ions which deviate from 360° (ca. 338°) and are close to the ideal value of 328.4° expected for sp^3^ hybridization. The two “body” Fe^III^ atoms are bridged by two μ_3_-O^2−^ ions, while a single μ_3_-O^2−^ also bridges a “wingtip” Fe^III^ atom. The four dpkox^−^ ligands adopt the 2.1110 coordination mode. Two of the dpkox^−^ ligands are coordinated to the “body” Fe2 through one 2-pyridyl and the oximato N atoms, and they use their oximato O atom to bridge one of the “wingtip” metal ions Fe1 and Fe4, and so Fe2 has a distorted octahedral O_2_N_4_ donor set. The other two dpkox^−^ ligands are coordinated through their 2-pyridyl and oximato N atoms to the “wingtip” Fe1 and Fe4 ions, whereas their oximato O atom is bonded to the “body” Fe3 ion. This “body” ion is also bridged to each of the “wingtip” Fe1 and Fe4 through a *syn*, *syn*-2.11 MeCO_2_^−^ group. The octahedral coordination around each of the “wingtip” Fe^III^ atoms is completed by a terminal chlorido (**71**, **72**) or azido (**73**) ion. In this manner, Fe3 has a {Fe^III^O_6_} coordination sphere, while Fe1 and Fe4 have an O_3_N_2_Cl (**71**, **72**) or O_3_N_3_ (**73**) octahedral coordination. 

**Figure 57 molecules-30-00791-f057:**
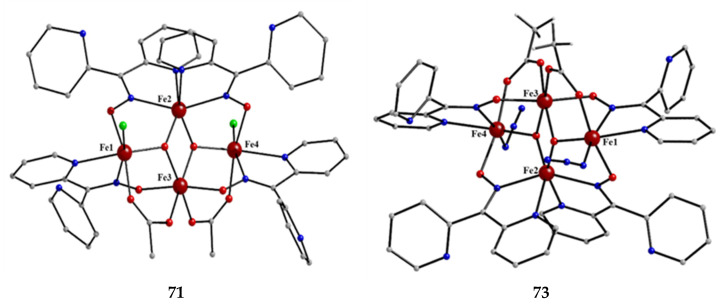
The molecular structures of the polymorph [Fe^III^_4_O_2_Cl_2_(O_2_CMe)_2_(dpkox)_4_] (**71**) [with a 2CH_2_Cl_2_∙H_2_O lattice solvent set] and the azido cluster [Fe^III^_4_O_2_(N_3_)_2_(O_2_CMe)_2_(dpkox)_4_] (**73**).

The ^57^Fe-Mössbauer spectra of **71** and **73** have *δ* and Δ*Ε*_Q_ parameters typical of high-spin Fe^III^ sites. Variable-temperature magnetic susceptibility studies on **71** revealed antiferromagnetic exchange interactions between the “body”-“body” and “wingtip”-“body” Fe^III^ ions resulting in an S = 1 ground state [73].

The use of dpkoxH in cobalt acetate chemistry has provided access to structurally interesting mixed-valence tetranuclear Co(II)/Co(III) and purely Co(III) clusters [74,75]. Complexes **74**–**76** were prepared by the reactions outlined in Equations (55)–(57) in good yields (50–70%). MeC(=O)-O-O-H is the powerful oxidant agent peracetic acid.
4 Co^II^(O_2_CMe)_2∙_4H_2_O + 4 dpkoxH + ½ O_2_ + 2 NaClO_4_ + 2 MeOH(55)$\overset{\mathrm{MeOH}/H_{2}O\;\left(5:1\;v/v\right)}{\to }\;$ [Co^II^_2_Co^III^_2_(OH)_2_(O_2_CMe)_2_(dpkox)_4_(MeOH)_2_](ClO_4_)_2_ + 2 Na(O_2_CMe) + 4 MeCO_2_H + 15 H_2_O                                                                    **74**
4 Co^II^(O_2_CMe)_2∙_4H_2_O + 4 dpkoxH + ½ O_2_ + 2 NaPF_6_ + 2 MeOH + 2 EtOH(56)$\overset{\mathrm{MeOH}/\mathrm{EtOH}\;\left(10:1\;v/v\right)}{\to }$ [Co^II^_2_Co^III^_2_(OMe)_2_(O_2_CMe)_2_(dpkox)_4_(EtOH)_2_](PF_6_)_2_ + 2 Na(O_2_CMe) + 4 MeCO_2_H + 17 H_2_O                                                         **75**
4 Co^II^(O_2_CMe)_2∙_4H_2_O + 4 dpkoxH + 2 MeC(=O)-O-O-H + 2 NaPF_6_(57)$\overset{{\mathrm{MeCO}}_{2}H/H_{2}O,80\;{}^{\circ}C}{\to }$ [Co^III^_4_(OH)_2_(O_2_CMe)_4_(dpkox)_4_](PF_6_)_2_ + 2 Na(O_2_CMe) + 4 MeCO_2_H + 16 H_2_O                                                         **76**


The centrosymmetric tetranuclear cation of **74** (Figure 58) has a rectangular arrangement of the four metal ions. The rectangle is defined by Co1∙∙∙Co2 [3.213(1) Å] and Co1∙∙∙Co2’ [4.441(1) Å] sides and their symmetric equivalents [74]. The Co1/Co1’ centers are low-spin Co^III^ atoms and the Co2/Co2’ ones are high-spin Co^II^ atoms. The cobalt centers are bridged along each short side of the rectangle by one hydroxido, one *syn*, *syn*-2.11 MeCO_2_^−^ and one oximato group while bridging along each long side is achieved through one oximato group only. The dpkox^−^ ligands are of two types arranged along the short and long sides of the rectangle. Short-side dpkox^−^ ions function as 2.1110 ligands (a more clear description is 2.1_1_1_2_1_2_0 where the subscript 1 refers to Co^II^ and the subscript 2 to Co^III^). Long-side dpkox^−^ ions adopt the 2.1111 (or better 2.1_1_1_2_1_2_1_1_) coordination mode. A terminal MeOH molecule occupies the sixth coordination site at each Co^II^ center. All the metal ions have a distorted octahedral geometry. Compound **74** realizes an inverse 12-MC-4 motif and can accommodate two OH^−^ ions within the metallacrown ring (the regular motifs accommodate metal ions). Using the metallacrown nomenclature, the cation of **74** can be formulated as {(OH)_2_[inv 12-MC_Co(II,III)(dpkox)_-4](O_2_CMe)_2_} [3,22,74]. The –[O-Co-O-N-Co-N]- repeat unit observed in this complex is perfectly acceptable for inverse metallacrown structures and cannot sustain a regular metallacrown (with an encapsulated fifth metal ion), as the latter would require adjacent six- and four-membered chelating rings.

**Figure 58 molecules-30-00791-f058:**
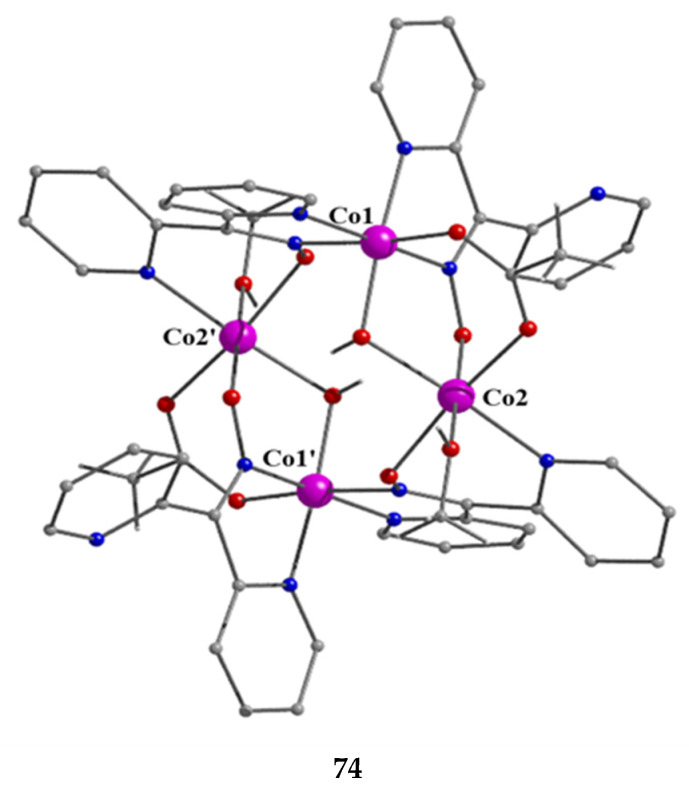
The structure of the cation [Co^II,II^_2_Co^III,III^_2_(OH)_2_(O_2_CMe)_2_(dpkox)_4_(MeOH)_2_]^2+^ that is present in complex **74**. The H atoms of the hydroxido, acetato, and methanol ligands have been drawn.

The cation of **75** has a molecular structure analogous to that of **74**, but with MeO^−^ and EtOH ligands in place of OH^−^ and MeOH ligands, respectively [74].

ESI-MS studies in MeCN suggest that the structures of the cations are retained in solution. Cyclic voltammetry experiments in the same solvent reveal a quasireversible Co^III^→Co^II^ reduction process and a resistance to oxidation of Co^II^. Because the paramagnetic Co^II^ atoms alternate with the diamagnetic Co^III^ atoms, solid-state dc magnetic susceptibility measurements in the 2–300 K range indicate that the Co^II^∙∙∙Co^II^ exchange interaction is negligible, if any [74].

Using the strong oxidizing agent peracetic acid (and contrary to the cyclic voltammetry studies of **74** and **75** mentioned above), the group of Masters was able to prepare the all-Co(III) version of **74** and **75**, i.e., compound **76**, Equation (57). Again, the sides of the centrosymmetric rectangle comprise two long and two short Co^III^∙∙∙Co^III^ distances [75]. Long-side Co^III^ ions are bridged by one oximato group of a 2.1111 dpkox^−^ ligand and short-side Co^III^ ions are bridged by one oximato group of a 2.1110 dpkox^−^, one hydroxido, and one *syn*, *syn*-2.11 MeCO_2_^−^ ligands. Two monodentate acetato groups are bonded to two metal ions (Co2 and Co2’ in Figure 59). The octahedral coordination spheres are {Co1/1’O_2_N_4_} and {Co2/Co2’O_5_N}. Complex **76** is a rather poor chain transfer catalyst for the polymerization of methylacrylate, but it does not catalyze cyclohexane oxidation in the presence or absence of co-catalysts [75]. In the former case, AIBN (azobisisobutyronitrile), {(CH_3_)_2_C(CN)}_2_N_2_, was used as the initiator, while in the latter case, NHPI, C_6_H_4_(CO)_2_NOH, was employed as co-catalyst with 100% O_2_ as oxidant.

The Ni(II)/dpkoxH chemistry is interesting [60,76,77,78,79,80]. Complex **77** is the only tetranuclear metal complex with dpkox^−^-ligation and no ancillary ligand, except the coordinated solvent molecules. The complex was prepared by the reaction shown in Equation (58) in 50% yield [76].
Ni(ClO_4_)_2_∙6H_2_O + 6 dpkoxH + 7 NaOH + 2 MeOH $\overset{MeOH}{\to }$
[Ni_4_(dpkox)_6_(MeOH)_2_](OH)(ClO_4_)+ 7 NaClO_4_ + 30 H_2_O(58)
**77**


**Figure 60 molecules-30-00791-f060:**
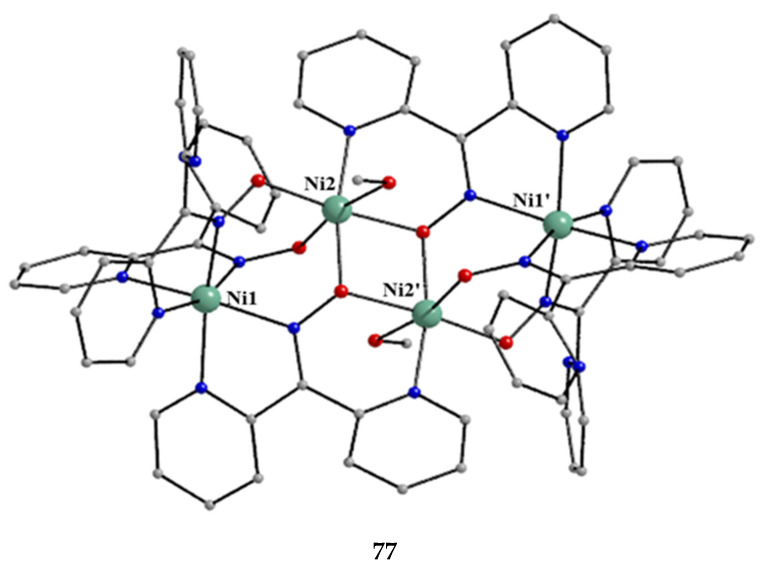
The structure of the cation [Ni_4_(dpkox)_6_(MeOH)_2_]^2+^ that is present in the crystal of **77**.

The four octahedral Ni^II^ atoms in the centrosymmetric cation (Figure 60) are held together by four 2.1110 and two 3.2111 dpkox^−^ ligands (Figure 14). The whole structure can be characterized as having a “metallacrown chair” topology [76]. The Ni2 and Ni2’ atoms are bridged by two oximato O atoms from the two 3.2111 ligands forming a central {Ni^II^_2_(μ_2_-OR)_2_}^2+^ subcore. The two “wing” metal ions have a {Ni^II^N_6_} coordination sphere, while for the “internal” Ni2/Ni2’ atoms the donor set is NO_5_ (with the participation of one MeOH ligand at each metal ion). The formation of the 12-membered metallacrown follows the pattern (Ni-N-O-Ni-O-N-Ni-N-O-Ni-O-N) and, in combination with the *syn*, *anti*-2.1110 dpkox^−^ ligands, allows the construction of the chair-like metallacrown motif. There are both ferromagnetic (Ni1∙∙∙Ni2/Ni1’∙∙∙Ni2’ and Ni1∙∙∙Ni2’/Ni1’∙∙∙Ni2) and antiferromagnetic (Ni2∙∙∙Ni2’) exchange interactions within the cation leading to an S = 0 ground state.

Our group studied the dpkox^−^/MeCO_2_^−^/SCN^−^ ligand “blend” in Ni(II) chemistry, which provided access to the cationic complex **78** in moderate yield [77], Equation (59).
4 Ni(O_2_CMe)_2_∙4H_2_O + 4 dpkoxH + NaSCN$\overset{MeOH}{\to }$
(59)[Ni_4_(O_2_CMe)_2_(dpkox)_4_](SCN)(OH)+ 5 MeCO_2_H+ Na(O_2_CMe) + 15 H_2_O**78**



The core of the complex consists of a tetrahedron of octahedral Ni^II^ atoms linked together by four 3.2111 dpkox^−^ ligands and two *syn*, *syn*-2.11 MeCO_2_^−^ groups [77]; thus, a distorted “{Ni_4_(NO)_4_}^4+^ “cube” is formed comprising single (O) and double (N-O) atom bridges. Peripheral ligation is provided by the eight 2-pyridyl N and the four acetato O atoms (Figure 61). The magnetic behavior of **78** is consistent with dominant antiferromagnetic interactions and an *S* = 0 ground state; the latter is corroborated by the appearance of a maximum in molar magnetic susceptibility at 18 K. 

Treatment of NiSO_4_∙6H_2_O with one equivalent of dpkoxH and one equivalent of Et_3_N in MeOH gave orange crystals of the neutral complex **79** in 57% yield [78], Equation (60).
4 NiSO_4_∙6H_2_O + 4 dpkoxH + 4 Et_3_N + 4 MeOH $\overset{\mathrm{MeOH}}{\to }$
[Ni_4_(SO_4_)_2_(dpkox)_4_(MeOH)_4_] + 2 (Et_3_NH)_2_SO_4_+ 24 H_2_O(60)
**79**


In the molecular structure of centrosymmetric **79** (Figure 62), the Ni^II^ centers are held together by two 3.2111 and two 2.1110 dpkox^−^ ligands, as well as two 2.1100 SO_4_^2−^ ions [78]. Four MeOH molecules act as terminal ligands completing octahedral coordination at each metal ion. The chromophores are {Ni(1,1’)(N_py_)(N_ox_)(O_ox_)_2_(O_sulf_)(O_met_)} and {Ni(2,2’)(N_py_)_2_(N_ox_)(O_ox_)(O_sulf_)(O_met_)}, where the abbreviations py, ox, sulf, and met are for the 2-pyridyl, oximato, sulfato, and methanol groups, respectively. The molecule has a metallacrown topology [3]. A *pseudo* 12-MC-4 ring is formed, because the *true* 12-MC-4 motif is “destroyed” by the bridging character of the two oximato O atoms that belong to the 3.2111 dpkox^−^ ligands (Figure 63).

The use of the ancillary ligand ethanolamine (eaH, Figure 15) in Ni(II)/dpkoxH chemistry led to the isolation of compound **80** in 68% yield [79], Equation (61).
4 Ni(ClO_4_)_2_∙6H_2_O + 4 dpkoxH + 4 eaH + 6 Et_3_N $\overset{\mathrm{MeOH},\;T}{\to }$
[Ni_4_(ea)_2_(eaH)_2_(dpkox)_4_](ClO_4_)_2_+ 6 (Et_3_NH)(ClO_4_) + 24 H_2_O(61)
**80**


ESI-MS spectra (in the positive ion mode) for **80** demonstrate its respective molecular ion peaks due to the [M-ClO_4_-4H_2_O]^+^ ionic species. The structure of the cation of **80** [79] is shown in Figure 64. The presence of a crystallographically imposed inversion center within the cation implies the equivalence of all four octahedral Ni^II^ atoms which are related by a S_4_ axis of symmetry. The donor set of each metal ion is O_3_N_3_. The dpkox^−^ ligands are coordinated in the 2.1110 manner, while eaH and ea^−^ adopt the ligation mode 2.21. Thus, two neighboring Ni^II^ atoms are connected by a pair of monoatomic O_alkoxido_ and diatomic (NO)_oximato_ bridges, generating an inverse 12-MC-4 topology. The four Ni^II^ centers form a perfect square with Ni∙∙∙Ni distances of 3.45 Å and Ni∙∙∙Ni∙∙∙Ni angles of 90°. Two alkoxy/alkoxido O atoms are above and below the Ni_4_ plane with O∙∙∙O axes orthogonal to each other, leading to a distorted tetrahedral arrangement for these O atoms inside the metallacrown cavity. The alkoxy (neutral) and alkoxido (deprotonated) O atoms cannot be distinguished because they form two very strong symmetrical H bonds of the -O∙∙∙H∙∙∙O- type both above and below the Ni_4_ plane; the O∙∙∙O separations are very short (ca. 2.47 Å).

Antiferromagnetic exchange interactions within the cation (derived from an 1-*J* model) result in an S = 0 ground state for **80** [79].

The unique tetranuclear Ni(II) cluster **81** was prepared and characterized by Kessissoglou’s and Pecoraro’s groups [80], Equation (62); shiH^2−^ is the dianion of salicylhydroxamic acid (shiH_3_, **K**; Figure 12). The yield was 60%. The appearance of MeOH in both the reactants and the products may be confusing. The MeOH molecule in the reactants arbitrarily denotes its incorporation as a ligand in the tetranuclear complex. The six free MeOH molecules in the products are assumed to be derived from the neutralization of the six MeO^−^ with six protons, two from dpkoxH and four from the shiH_3_ ligands.
4 NiCl_2_∙6H_2_O + 2 NH_4_SCN + 2 shiH_3_ + 2 dpkoxH $+6\;\mathrm{NaOMe}+\mathrm{DMF}+\mathrm{MeOH}\;\overset{MeOH/DMF\;\left(10:1\;v/v\right)}{\to }$
(62)[Ni_4_(SCN)_2_(shiH)_2_(dpkox)_2_(DMF)(MeOH)] + 2 NH_4_Cl + 6 NaCl + 6 MeOH + 24 H_2_O                                       **81**


Complex **81** (Figure 65) [80] is a rare example of a vacant metallacrown with mixed-ligand composition and can be written as {[12-MC_Ni(II)(shiH)2(dpkox)2_-4](SCN)_2_(DMF)(MeOH)]}. The molecule shows the connectivity pattern [-O-Ni-O-N-Ni-N-]_2_ (Figure 66). This pattern differs from other Ni(II) metallacrowns that follow the common [-Ni-O-N-]_4_ pattern. Whereas **81** shows the 6-5-6-5-6-5-6-5 arrangement of chelating rings, the other known Ni(II) metallacrowns (which are pentanuclear) exhibit the 6-6-5-5-6-6-5-5 or 6-6-5-5-6-5-6-5 patterns (vide infra); the numbers indicate the sizes (5- or 6-membered) of the chelating rings.

The dpkox^−^ and shiH^2−^ ligands adopt the coordination mode 2.1111 (Figure 14 and Figure 66), the former using three nitrogen and one oxygen atom, and the latter three oxygen and one nitrogen atom. The octahedral geometry at Ni1 is completed with one isothiocyanato and one MeOH ligand, and at Ni3 with one isothiocyanato and one DMF ligand. The Ni2 and Ni4 centers have a square planar coordination geometry. It was expected that no interaction would occur between the paramagnetic octahedral Ni^II^ centers (Ni1 and Ni3 in Figure 65), because there is not a short, through-bond pathway to make feasible superexchange between them (Ni2 and Ni4 are diamagnetic). The variable-temperature magnetic susceptibility data demonstrated that **81** follows the Curie Law behavior, suggesting that there is no coupling between the paramagnetic centers [80].

**Figure 65 molecules-30-00791-f065:**
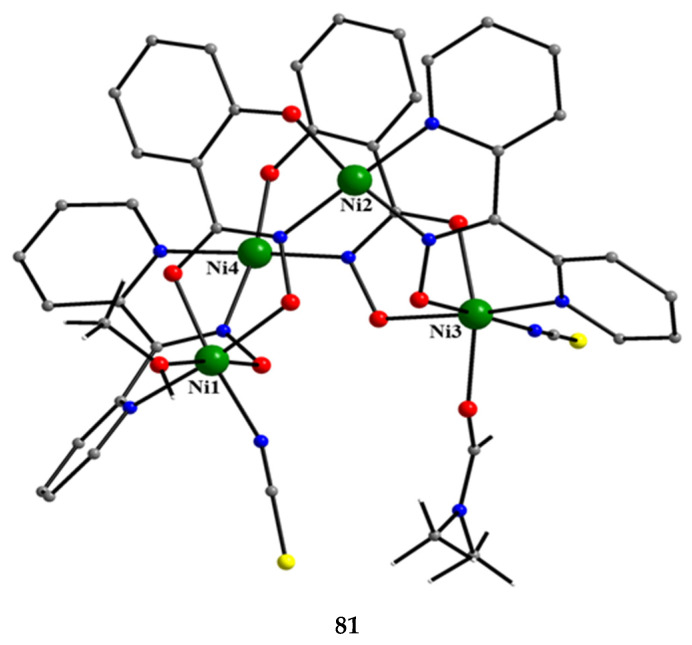
The molecular structure of [Ni_4_(SCN)_2_(shiH)_2_(dpkox)_2_(DMF)(MeOH)] (**81**).

**Figure 66 molecules-30-00791-f066:**
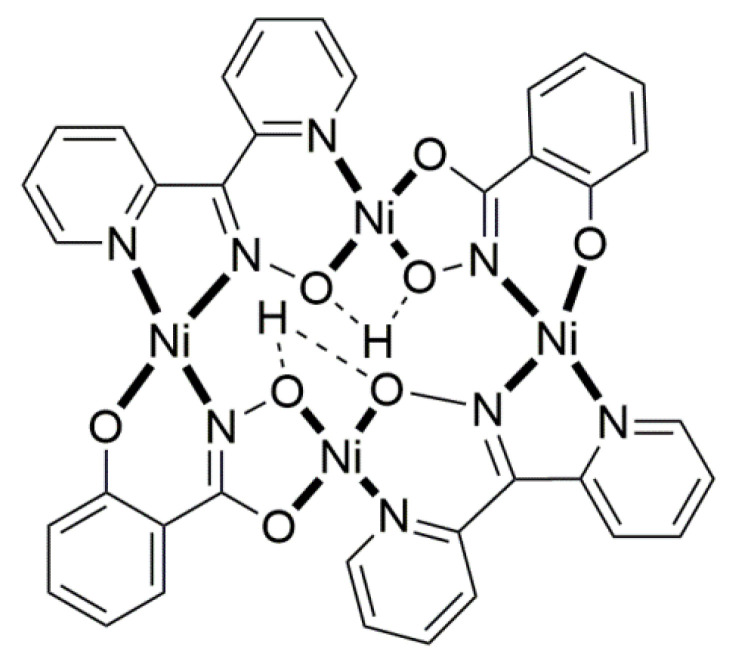
Drawing showing the connectivity pattern and the arrangement around the Ni^II^ centers in [Ni_4_(SCN)_2_(shiH)_2_(dpkox)_2_(DMF)(MeOH)] (**81**). The coordination bonds are indicated with bold lines.

The simultaneous use of the ancillary ligands dpt (Figure 15) and N_3_^−^ in Ni(II)/dpkoxH chemistry gave complex **82** [60], Equation (63); the yield was not reported.
4 NiCl_2_∙6H_2_O + 4 dpkoxH + 4 NaN_3_ + 2 dpt + 4 Et_3_N $\overset{MeCN}{\to }$
[Ni_4_(N_3_)_4_(dpt)_2_(dpkox)_4_]+ 4 NaCl + 4 (Et_3_NH)Cl + 24 H_2_O(63)
**82**


The molecular structure of **82** is shown in Figure 67. The four metal ions are in a zigzag topology [60]. The centrosymmetric molecule can be described as consisting of two dinuclear units. In each dinuclear unit, the two Ni^II^ atoms are doubly bridged by one μ_2_-1,1 (or 2.200) azido ligand and one oximato group of a 2.1110 dpkox^−^ ligand. The interdimer connection is provided by two oximato ligands of the rest dpkox^−^ ligands. The octahedral {Ni^II^ON_5_} coordination sphere of the terminal Ni^II^ atoms (Ni2/Ni2’) is completed by one N_3_-tridentate chelating (1.111) dpt molecule and one terminal (1.100) azido group. The octahedral coordination sphere at each central metal ion (Ni1/Ni1’) is also {Ni^II^ON_5_}, but the origin of the N atoms is different; the three N atoms of dpt and the nitrogen of the terminal azido group have been replaced by two oximato and two 2-pyridyl N atoms. The complex shows an overall antiferromagnetic response (S = 0) [60].

Zinc(II) forms a family of 12-MC-4 metallacrowns with inverse topology (vide infra), complexes **83**–**88** [33,81,82]. At this point, we would like to emphasize again the differences between the “regular” and “inverse” metallacrowns. In the latter, the metal ions, rather than anionic oxygen atoms, are oriented toward the central cavity. In the former, the coordination number and environment around the metal centers in the ring are usually uniform. In the inverse metallacrowns, the metal ions are site differentiated having different coordination numbers. Also, the connectivity around the ring is different from that in regular 12-MC-4 compounds where the linkage is consistently N-O-M-N-O-M; in contrast, in the inverse compounds the linkage is transposed to N-O-M-O-N-M and the ligand is coordinated with only three heteroatoms leaving one of the 2-pyridyl N atoms unbound. Preparative routes for **83**–**88** are summarized in Equations (64)–(68), where R = Me, Ph, acac = 2,4-pentadionato(−1) ion, and S = various solvents; the yields were higher than 50%.
4 Zn(O_2_CR)_2_∙2H_2_O + 4 dpkoxH $\overset{S}{\to }$
[Zn_4_(OH)_2_(O_2_CR)_2_(dpkox)_4_]+ 6 RCO_2_H + 6 H_2_O                              (64)
**83**, **84**




$4\;{\mathrm{ZnCl}}_{2}+4\;\mathrm{dpkoxH}+6\;\mathrm{NaOH}\;\overset{\mathrm{MeOH}}{\to }$

[Zn_4_(OH)_2_Cl_2_(dpkox)_4_]+ 6 NaCl + 4 H_2_O                                          (65)

**85**



4 Zn(NO_3_)_2_∙4H_2_O + 2 NaN_3_ + 4 dpkoxH + 6 NaOH $\overset{\mathrm{DMF}}{\to }$
[Zn_4_(OH)_2_(N_3_)_2_(dpkox)_4_]+ 8 NaNO_3_ + 20 H_2_O(66)

**86**



4 ZnCl_2_ + 2 NaOCN + 4 dpkoxH + 6 NaOH $\overset{\mathrm{DMF}/\mathrm{MeOH}}{\to }$
[Zn_4_(OH)_2_(NCO)_2_(dpkox)_4_]+ 8 NaCl + 4 H_2_O     (67)

**87**



4 Zn(acac)_2_∙H_2_O

$+4\;\mathrm{dpkoxH}\;\overset{{\mathrm{CH}}_{2}{\mathrm{Cl}}_{2},\;T}{\to }$

[Zn_4_(OH)_2_(acac)_2_(dpkox)_4_]+ 6 acacH + 2 H_2_O                               (68)
**JJ**


**88**



The molecular structures of **83**–**88** are all similar (or better analogous); the only difference is the identity of the peripheral ligands (MeCO_2_^−^, PhCO_2_^−^, Cl^−^, N_3_^−^, NCO^−^, acac^−^). The structures of selected complexes are shown in Figure 68; we describe in detail only the structure of **83**, which was the first inverse metallacrown reported in the literature [81]. The preparation of an inverse metallacrown was a direct consequence of the substitution of dpkox^−^ for the previously used salicylhydroxamate ligands (Figure 12).

The tetranuclear molecule of **83** lies on a crystallographic inversion center and has a planar, nearly rhombic arrangement of the Zn^II^ atoms. The metal centers are bridged along each side of the rhombus by one μ_3_-hydroxido group (3.3) and one diatomic oximato group from one 2.1110 dpkox^−^ ligand. The strict description of the core is thus {Zn^II^_4_(μ_3_-OH)_2_}^6+^. The coordination of the OH^−^ ligand is markedly pyramidal. Zn1 and Zn1’ are in a distorted O_2_N_4_ octahedral environment, whereas Zn2 and Zn2’ are in a severely distorted O_5_ environment, the carboxylato groups being anisobidentate chelating, i.e., one O forms a weak bond to the metal ion. As expected for an inverse metallacrown, the connectivity is N-O-Zn-O-N-Zn-N-O-Zn-O-N-Zn. Using the metallacrown nomenclature, the representation of the complex is {(OH)_2_[inv12-MC_Zn(II)N(dpkox)_-4](O_2_CMe)_2_}. The dpkox^−^ ligands have a propeller configuration that imposes absolute stereoisomerism, with Λ chirality on Zn1 and A chirality on Zn1’ [81].

The molecular structures of **84**–**88** are similar to the structure of **83**, except that the two anisobidentate chelating MeCO_2_^−^ ligands of **83** are replaced by two monodentate PhCO_2_^−^ ligands in **84** [82], two terminal chlorido groups in **85** [33], two monodentate azido groups in **86** [82], two monodentate isocyanato (i.e., N-bonded) groups in **87** [82], and two bidentate chelating acetylacetonato ligands in **88** [82]. The coordination geometries of the non-octahedral Zn^II^ atoms are distorted tetrahedral (**84** and **85**), tetrahedral (**86**, **87**), and distorted trigonal bipyramidal (**88**), the latter because of the chelating acac^−^ ligand which creates a true five-coordination at Zn2/Zn2’ (Figure 68).

Complex **89** [83] is structurally similar to compounds **83**–**88**, but its chemical identity differs. The two ancillary ligands in the latter have been replaced by two extra dpkox^−^ ligands in the former, but the hydroxido groups are retained. The complex was prepared by the reaction outlined in the balanced Equation (69).
4 ZnCl_2_ + 6 dpkoxH + 8 KOH $\overset{MeOH}{\to }$
[Zn_4_(OH)_2_(dpkox)_6_]+ 8 KCl + 6 H_2_O(69)
**89**


**Figure 69 molecules-30-00791-f069:**
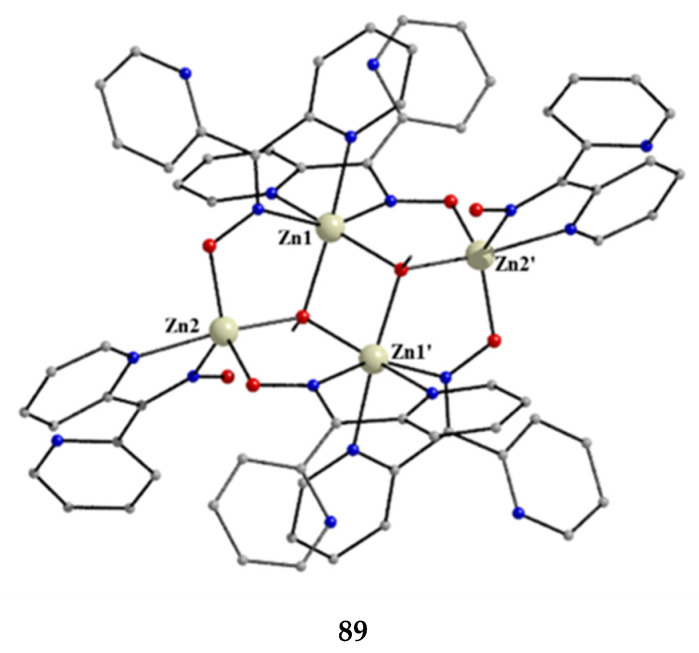
The molecular structure of [Zn_4_(OH)_2_(dpkox)_6_] (**89**).

The “{Zn_4_(OH)_2_(dpkox)_4_}^2+^” fragment of the structure of **89** (Figure 69) is almost identical to the corresponding unit of **83**–**88**, and especially to that of **88**. The only difference is the replacement of the bidentate chelating acac^−^ groups of **88** by two chelating 1.0110 dpkox^−^ ligands in **89** [83]. The distorted trigonal bipyramidal geometry at Zn(2)/Zn(2’) is retained. ^1^H NMR studies suggest that the molecule is stable in CDCl_3_.

Complex **90** is the only heterometallic tetranuclear compound based on dpkox^−^. It is the mixed-metal/mixed-ligand metallacrown [Ni_2_Mn^III^_2_(O_2_CMe)_2_(shi)_2_(dpkox)_2_(DMF)_2_], where shi^3−^ is the trianion of salicylhydroxamic acid (shiH_3_, **K**; Figure 12). Using the metallacrown representation, the complex can be written as {(O_2_CMe)_2_[12-MC_Ni(II)Mn(III)N(shi)2(dpkox)2_-4](DMF)_2_} [73]. Compound **90** was prepared by the reaction shown in Equation (70) in almost quantitative yield (ca. 95%).
2 Ni(O_2_CMe)_2_∙4H_2_O + 2 Mn^II^(O_2_CMe)_2_∙4H_2_O + 2 shiH_3_ + 2 dpkoxH $+2\;\mathrm{DMF}+½\;O_{2}\;\overset{\mathrm{DMF}}{\to }$
(70)[Ni_2_Mn^III^_2_(O_2_CMe)_2_(shi)_2_(dpkox)_2_(DMF)_2_] + 6 MeCO_2_H + 17 H_2_O                        **90**


The connectivity pattern of centrosymmetric **90** is illustrated in Figure 70. The dpkox^−^ and shi^3−^ ligands adopt the coordination modes 3.2_1_1_1_1_2_1_2_ (Figure 14 and Figure 70) and 2.1_1_1_1_1_2_1_2_ (Figure 70), respectively; the subscript 1 refers to Mn^III^ and the subscript 2 to Ni^II^. The shi^3−^ provides one O, O and one N, O chelating parts, whereas dpkox^−^ offers one N, O and one N, N chelating parts. The bridging character of the O atom of dpkox^−^ (it bridges two Mn^III^ atoms) causes a “collapsed” metallacrown structure. Like **81**, molecule **90** shows the 6-5-6-5-6-5-6-5 arrangement of chelating rings. Each acetato group bridges a Ni^II^Mn^III^ pair through its O atoms (2.1_1_1_2_), while a terminal DMF molecule completes octahedral coordination at Ni^II^. Complex **90** is antiferromagnetically coupled.

### 5.6. Pentanuclear Complexes

Pentanuclear dpkox^−^-based clusters are known for Ni(II), Cu(II) and Zn(II) [Table 6]. Our group has prepared complex 91 [84], Equation (71).
5 Ni(NO_3_)_2_∙6H_2_O + 5 dpkoxH + 5 LiOH∙H_2_O $\overset{\mathrm{EtOH}}{\to }$
[Ni_5_(dpkox)_5_(H_2_O)_7_](NO_3_)_5_+ 5 LiNO_3_ + 33 H_2_O(71)
**91**


The five Ni^II^ atoms in the cation of 91 are held together by four 3.2111 and one 2.1110 dpkox^−^ ligands. In addition, there are two *trans*-terminal H_2_O ligands each on Ni1, Ni3, and Ni4, and a unique H_2_O on Ni2 (Figure 71). The cations Ni, Ni3, Ni4, and Ni5 create a highly distorted (flattened) tetrahedron (Figure 72, left), with the fifth ion (Ni2) lying at the midpoint of the Ni3∙∙∙Ni4 edge (and not at the center of the tetrahedron). Each of the four 3.2111 dpkox^−^ groups spans an edge of the Ni_4_ tetrahedron and is coordinated to Ni2 through its doubly bridging oximato O atom. As a consequence, the Ni^II^∙∙∙Ni^II^ edges spanned by the μ_3_-dpkox^−^ groups become shorter (~4.8 Å) than the two unspanned Ni1∙∙∙Ni5 (6.01 Å) and Ni3∙∙∙Ni4 (7.01 Å) ones. Due to the insertion of Ni2 between Ni3 and Ni4, the Ni3∙∙∙Ni4 edge is the longest. The five octahedral chromophores are all different, i.e., Ni1O_4_N_2_, Ni2O_6_, Ni3O_3_N_3_, Ni4O_2_N_4,_ and Ni5ON_5_. The core of the cation is {Ni^II^_5_(μ_3_-ONR)_4_(μ_2_-ONR)}^5+^, where RNO^−^ = dpkox^−^ (Figure 72, right). A simple 2-*J* model, based on the Hamiltonian of Equation (72), was found to be satisfactory in describing the variable-temperature dc magnetic susceptibility data (*J*_1_ = −21.8 cm^−1^, *J*_2_ = −45.9 cm^−1,^ and *g* = 2.24). Magnetization data collected at 2.0 K support an *S* = 1 ground state [84].

**Figure 71 molecules-30-00791-f071:**
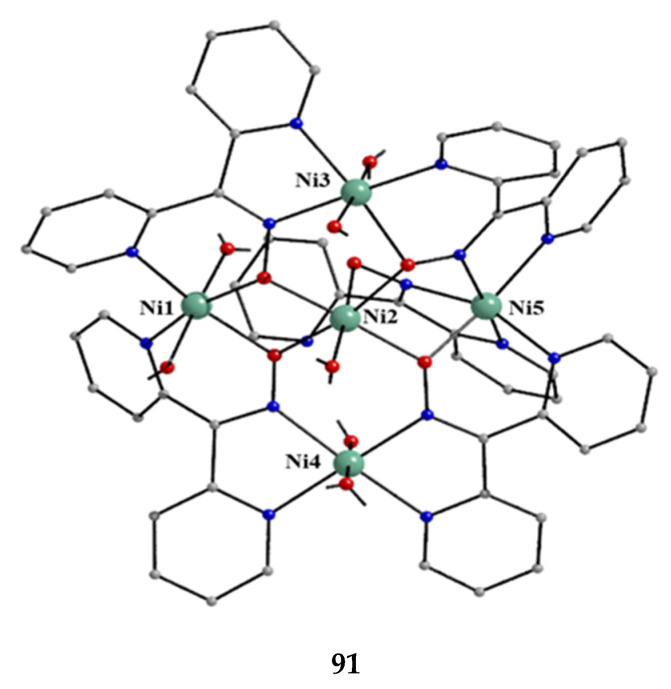
The cation [Ni_5_(dpkox)_5_(H_2_O)_7_]^2+^ that is present in the crystal structure of **91**.

**Figure 72 molecules-30-00791-f072:**
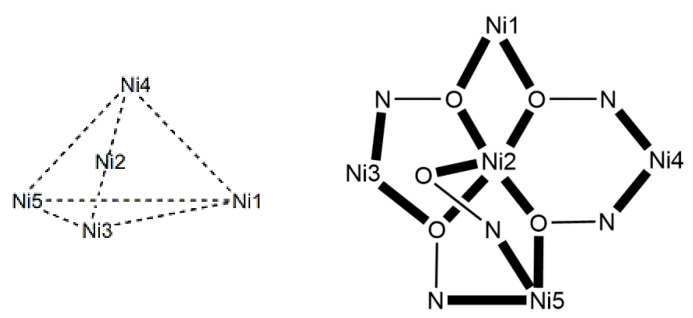
The Ni_5_ skeleton of **91** showing the very distorted tetrahedral metal topology (**left**) and the {Ni^II^_5_(μ_3_-ONR)_4_(μ_2_-ONR)}^5+^ core of **91** (**right**).


*H* = −*J*_1_(*Ŝ*_1_∙*Ŝ*_3_ + *Ŝ*_3_∙*Ŝ*_5_ + *Ŝ*_5_∙*Ŝ*_4_ + *Ŝ*_4_∙*Ŝ*_1_) − *J*_2_(*Ŝ*_1_∙*Ŝ*_2_ + *Ŝ*_2_∙*Ŝ*_3_ + *Ŝ*_2_∙*Ŝ*_4_ + *Ŝ*_2_∙*Ŝ*_5_)(72)


In addition to complex **78** [77], another Ni(II) cluster (complex **92**) with the binary dpkox^−^/MeCO_2_^−^ ligand “blend” was also prepared by our group [77] by changing the solvent and omitting SCN^−^ from the reaction mixture, Equation (73); the yield was 60%.
5 Ni(O_2_CMe)_2_∙4H_2_O + 3 dpkoxH $\overset{{\mathrm{Me}}_{2}\mathrm{CO}/H_{2}O\;\left(5:1,\;v/v\right)}{\to }$
[Ni_5_(O_2_CMe)_7_(dpkox)_3_(H_2_O)]+ 3 MeCO_2_H + 19 H_2_O(73)
**92**


The molecular structure of **92** (Figure 73) consists of five Ni^II^ centers in a closed, cage-like motif. The octahedral metal ions are held together by two 3.2111 and one 3.2110 dpkox^−^ ligands (Figure 14). The seven acetato groups adopt four different coordination modes. Four of them are coordinated through the common 2.11 mode, and the remaining three are in the 1.10, 2.20, and 2.21 modes. Two metal ions (Ni1 and Ni3) are further bridged by the O atom of the aquo ligand which is H-bonded to two uncoordinated acetato O atoms [77].

The mixed-ligand complex **93** was prepared by the reaction shown in Equation (74); the yield was ca. 85% [80].
5 Ni(O_2_CMe)_2_∙4H_2_O + 2 shiH_3_ + 2 dpkoxH + 2 DMF $\overset{\mathrm{DMF}}{\to }$
[Ni_5_(O_2_CMe)_2_(shi)_2_(dpkox)_2_(DMF)_2_]+ 8 MeCO_2_H + 20 H_2_O(74)
**93**


The difference between **81** (Figure 65 and Figure 66) and **93** (Figure 74) lies in the different encapsulated cations (H^+^ for **81** and Ni^II^ for **93**) and coordinated anions (NCS^−^ for **81** and MeCO_2_^−^ for **93**). The two 3.2111 shi^3−^ and dpkox^−^ ligands are arranged in a *trans* configuration constructing a 12-metallacrown-4 motif with an encapsulated Ni^II^ atom (Ni5). Unlike **81**, the common [-Ni-N-O-]_4_ repeating unit is observed for **93**. Two *syn*, *anti*- 2.11 acetato groups bridge the encapsulated Ni^II^ atom to two-ring metal ions and thus the overall complex is neutral [80]. The dpkox^−^ ligands are not planar due to steric hindrance between the two 2-pyridyl rings. As in **81**, two Ni^II^ atoms (Ni4, Ni2) have a square planar geometry, the other being octahedral; the central encapsulated metal ion has a {Ni^II^O_6_} coordination sphere, with four oxygens coming from the metallacrown cavity (shi^3−^ and dpkox^−^ ligands) and two from the bridging MeCO_2_^−^ groups. The complex does not show the alternating scheme of five- and six-membered chelating rings that are observed for other metallacrowns (*vide supra*). In **93**, two five-membered chelating rings surround its square planar Ni^II^ atom, and two six-membered ones surround the ring metal ions. The exchange is weak antiferromagnetic, while in solution the complex is shown to be stable both to decomposition and to ligand exchange [80].

As mentioned in Section 5.3, the Cu(II)/dpkoxH reaction system is fertile. Change of the reaction conditions that led to the dinuclear complexes **27** and **28** (Table 2) gave the pentanuclear complex **94**, Equation (75); the yield was not reported [48].
5 Cu^II^(ClO_4_)_2_∙6H_2_O + 7 dpkoxH + 7 NaOH $\overset{\mathrm{MeOH}}{\to }$
[Cu^II^_5_(dpkox)_7_](ClO_4_)_3_+ 7 NaClO_4_ + 37 H_2_O(75)
**94**


The structure and the core of the cation of cluster **94** are shown in Figure 75 and Figure 76, respectively. The structure of the pentanuclear cation can be described as consisting of one dimeric (Cu2Cu3) and one trimeric (Cu1Cu4Cu5) unit bridged by dpkox^−^ ligands [48]. The Cu_3_ unit can be considered a scalene triangle. The chromophores are {Cu1NNNOO} (square pyramidal), {Cu2NNNNN} (trigonal bipyramidal), {Cu3NNOOO} (trigonal bipyramidal), {Cu4NNNNO} (trigonal bipyramidal), and {Cu5NNOOO} (square pyramidal); all the coordination polyhedra are distorted. The dpkox^−^ ligands adopt four coordination modes. Three ligands are 2.1110 (one intradimer, one intratrimer, and one interdimer/trimer), one is 3.1111 (interdimer/trimer), two are 3.2110 (one intratrimer and one interdimer/trimer) and a unique ligand behaves in a 4.2111 manner (interdimer/trimer); these coordination modes can be seen in Figure 14. Thus, the dimeric and trimeric units are linked through one 2.1110, one 3.2110, one 3.1111, and one 4.2111 dpkox^−^ ligands. The core of the cation is {Cu^II^_5_(μ_3_-ΝO)_3_(μ_2_-NO)_4_}^3+^, with N representing the oximato nitrogen. Variable-temperature, solid-state dc magnetic susceptibility results are indicative of dominant antiferromagnetic exchange interactions within the cation. The lowest temperature χ_Μ_Τ value (0.38 cm^3^ K mol^−1^) suggests an S = ½ ground state [48]. UV/Vis experiments in DMF solutions have demonstrated an equilibrium between **94** and [Cu^II,II^_2_(dpkox)_4_] (**27**) upon adding Cu^II^(ClO_4_)_2_∙6H_2_O in the reaction solution [48]. 

Our group initially came across the Zn(II) pentanuclear clusters **95** and **96** when we tried to prepare the isothiocyanato analogs of **85**–**87** (Table 5). This at first glance trivial effort led to remarkable nuclearity and structural changes. A mixture of **95** and **96** was obtained from a reaction mixture containing ZnCl_2_, a half equivalent of NaSCN, and one equivalent of Na(dpkox) in MeOH [82]. The manual separation of colorless crystals of **95** from their mixture with the pale yellow crystals of **96** remains the only source of the former to date; the latter was obtained in pure form by the reactions shown in Equations (76) and (77) in yields of ca. 80 and 30%, respectively.
6 Zn(O_2_CMe)_2_∙2H_2_O + 6 NaSCN + 6 dpkoxH + MeOH $\overset{\mathrm{MeOH}}{\to }$
(76)                                                [Zn_5_(NCS)_2_(dpkox)_6_(MeOH)][Zn(NCS)_4_] + 6 Na(O_2_CMe) + 6 MeCO_2_H + 12 H_2_O                                                                                 **96**
[Zn_5_Cl_2_(dpkox)_6_][ZnCl(NCS)_3_] $+3\;\mathrm{NaSCN}+\mathrm{MeOH}\;\overset{\mathrm{MeOH}/H_{2}O\;\left(10:1\;v/v\right),\;45{{}^{\circ}}_{C,\;\mathrm{exc}.\mathrm{NaSCN}}}{\to }$
(77)                          **95**
                                                                                                    [Zn_5_(NCS)_2_(dpkox)_6_(MeOH)][Zn(NCS)_4_] + 3 NaCl                                                                                                                                 **96**


The structures of the cations of clusters **95** and **96** (Figure 77; note that the numbering scheme of the Zn^II^ atoms is not the same) are similar [82] in many aspects and, thus, only the structure of the former will be described in detail. The complexes are cationic and the positive charge is balanced by the tetrahedral anions [ZnCl(NCS)_3_]^2−^ in **95** and [Zn(NCS)_4_]^2−^ in **96**. The five Zn^II^ atoms are held together by six dpkox^−^ ligands, which adopt three different coordination modes (two are 2.1110, two 3.1111, and two 3.2111). The metal ions define two nearly equilateral triangles sharing a common apex at Zn1 (Figure 77, left). The “central” ion (Zn1 in **95**, Zn3 in **96**) is in a distorted *cis-cis-trans* O_2_(N_2-pyridyl_)_2_(N_oximato_)_2_ octahedral environment. Zn2 and Zn3 possess very distorted trigonal bipyramidal coordination spheres consisting of O_2_(N_2-pyridyl_)_2_(N_oximato_) sets of donor atoms, while Zn4 and Zn5 are bound to a square pyramidal O(N_2-pyridyl_)_2_(N_oximato_)Cl set. The metal topology can be described as consisting of two “collapsed” 9-metallacrown-3 motifs sharing a common apex at Zn1. In **96**, the two terminal chlorido groups of **95** have been replaced by two terminal isothiocyanato ligands. A minor difference is also the fact that one five-coordinate metal ion in **95** has become six-coordinate in **96** through the coordination of one terminal MeOH molecule (Zn5; Figure 77, right).

### 5.7. Hexanuclear Clusters

Four hexanuclear dpkox^−^ -based clusters (**97**–**100**) with first-row transition metal ions have been reported (Table 6). The use of dpkoxH in manganese benzoate chemistry provided access to a {Mn^II^_3_Mn^III^_3_} complex which has a rare oxidation-state combination [85]. The reaction of Mn^II^(ClO_4_)_2_∙6H_2_O, PhCO_2_H, (Bu^n^_4_N)MnO_4,_ and (py)_2_CNOH(dpkoxH) in a 4:5:1:5 molar ratio in CH_2_Cl_2_ gave [Mn^II^_3_Mn^III^_3_O_2_(O_2_CPh)_6_(dpkox)_2_{(py)_2_C(OH)(O)}_2_](ClO_4_) (**97**) in 65% yield; (py)_2_C(OH)(O) is the monoanion of the *gem*-diol form of (py)_2_CO, the latter being produced by the metal ion-assisted/-promoted transformation of dpkoxH, Equations (40)–(42) and Figure 15. A simplified reaction illustrating the changes in the oxidation states of Mn is shown in Equation (78) with the assumption that Mn^VII^O_4_^2−^ is the exclusive oxidant, i.e., that atmospheric oxygen does not participate in the oxidation of Mn(II).
27 Mn^II^ + 3 Mn^VII^ → 5 {Mn^II^_3_Mn^III^_3_} (78)

The cation of **97** (Figure 78, left) contains three Mn^II^ (Mn2, Mn4, Mn6) and three Mn^III^ (Mn1, Mn3, Mn5) atoms, which are held together by six 2.11 PhCO_2_^−^, two 2.1110 dpkox^−^ (the Mn^II^ atoms are coordinated to the 2-pyridyl and the oximato N atoms, while the O atom is bonded to an oxophilic Mn^III^ center), two 3.3011 (py)_2_C(OH)(O)^−^ (one deprotonated triply bridging O atom is bonded to the three Mn^II^ atoms, and the other to two Mn^II^ and one Mn^III^ atoms) and two μ_4_ (4.4) O^2−^ (each bonded to two Mn^II^ and two Mn^III^ atoms) ligands. The core is {Mn^II^_3_Mn^III^_3_(μ_3_-O)_2_(μ_3_-OR)_2_}^9+^, where RO^−^ = (py)_2_C(OH)(O)^−^. It consists (Figure 78, right) of a central {Mn^II^_3_Mn^III^(μ_3_-O)_2_(μ_3_-OR)_2_}^3+^ cubane subcore, with the remaining two Mn^III^ atoms attached to the two vertices of the cube that are occupied by the O^2−^ groups, the latter becoming μ_4_ as a result [85]. The metal topology and the core of the complex remain novel to date. The χ_Μ_Τ vs. T curve indicates ferromagnetic coupling with an S = 15/2 ± 1 ground state; the complex is SMM.

The hexanuclear complex **98** [48], which can be roughly considered as a “dimer” of the trinuclear complex **48** (Table 3) [62], was prepared by the reaction shown in Equation (79); the yield was not reported.
$6\;{\mathrm{Cu}}^{\mathrm{II}}{\mathrm{Cl}}_{2}\cdot 2H_{2}O+6\;\mathrm{dpkoxH}+8\;\mathrm{NaOH}+2\;\mathrm{Na}({\mathrm{BPh}}_{4})\;\overset{DMF}{\to }$[Cu^II^_6_(OH)_2_Cl_2_(dpkox)_6_]+ 10 NaCl + 18 H_2_O(79)
**98**


The cation of 98 consists of two triangular {Cu^II^_3_(OH)Cl(dpkox)_3_}^+^ isosceles units (Figure 79). Each triangular unit is capped by the O atom of one 3.3 hydroxido group. Each edge is bridged by an oximato group. The oximato groups that bridge the Cu1∙∙∙Cu2 and Cu1∙∙∙Cu3 edges (and their symmetry equivalents) belong to two 2.1110 dpkox^−^ ligands; the oximato group which bridges the Cu2∙∙∙Cu3 edge (and its symmetry equivalent) belongs to a 3.1111 dpkox^−^ ligands. Thus, two 3.1111 dpkox^−^ ligands provide the intertrimer association. An edge of each triangle (Cu1∙∙∙Cu3 and its symmetry equivalent) is additionally bridged by a non-symmetric chlorido group. The distorted square pyramidal coordination spheres are {Cu(1,3)NNOOCl} and {Cu3NNNOO} [48]. Each triangular unit shows the pattern of 9-MC-3 complexes accommodating a coordinated OH^−^ anion. The Cu^II^∙∙∙Cu^II^ exchange interactions are antiferromagnetic.

A structurally interesting Cu(II) cluster containing two crystallographically distinct [18-MC_Cu(II)(N)dpkox_-6] cations was prepared by Pecoraro’s and Kessissoglou’s groups [48,86], Equation (80); the yield was ca. 60%.
12 Cu^II^(ClO_4_)_2_∙6H_2_O + 12 Na(dpkox) + 10 MeCN $\overset{\mathrm{MeCN}}{\to }$

(80)[Cu^II^_6_(ClO_4_)(dpkox)_6_(MeCN)_6_][Cu^II^_6_(ClO_4_)_3_(dpkox)_6_(MeCN)_4_](ClO_4_)_8_+ 12 NaClO_4_ + 72 H_2_O                        **99**



In both cations [Cu^II^_6_(ClO_4_)(dpkox)_6_(MeCN)_6_]^5+^ (**99a**) and [Cu^II^_6_(ClO_4_)_3_(dpkox)_6_(MeCN)_4_]^3+^ (**99b**), six Cu^II^ atoms and six 2.1111 dpkox^−^ ligands create the 18-MC-6 ring [48,86]. The cation **99a** is shown in Figure 80. The six square pyramidal Cu^II^ atoms of both rings are in a chair configuration. For each cation, the juxtaposed five- and six-membered chelating rings form the basis of the 18-MC-6 core through the [-Cu^II^-N-O-]_6_- linkage. The cations create a cavity, having a trigonal prism shape with trigonal bases created by 2-pyridyl C atoms. Each cavity contains an encapsulated, *non-coordinated* ClO_4_^−^ ion (Figure 81). The only difference between the two metallacrown cations is in the ligands bound to the Cu^II^ atoms on the outside of the ring. In **99a**, there are six MeCN molecules, one at each Cu^II^, whereas in **99b**, four Cu^II^ atoms have MeCN ligands and two possess monodentate perchlorato groups. Thus, from the six ClO_4_^−^ ions in **99b**, one is encapsulated and two are coordinated and an alternative formulation of this cation could be [Cu^II^_6_(ClO_4_)(dpkox)_6_(ClO_4_)_2_(MeCN)_4_]^3+^. The use of anions other than ClO_4_^−^ does not template the same 18-MC-6 motifs, but leads to complexes (such as **98**), though all reaction conditions are kept constant. The Cu^II^ centers in **99** are strongly antiferromagnetically coupled, resulting in an S = 0 ground state. When the complex is dissolved in DMF, the two cations (probably with DMF ligands instead of coordinated MeCN molecules) exist separately and are involved in several equilibria, e.g., the equilibrium represented by Equation (81).
5 [Cu^II^_6_(ClO_4_)(dpkox)_6_(solvent)_6_]^5+^ + 12 dpkox^−^ ⇆ 6 [Cu^II^_5_(dpkox)_7_]^3+^ + 5 ClO_4_^−^ + 30 solvents(81)

The reaction of ZnCl_2_ with dpkoxH and flufenamic acid (flufH; the structural formula of its anion is shown in Figure 15) in a basic medium led to the isolation of the hexanuclear cluster **100**, Equation (82) [87].
6 ZnCl_2_ + 4 dpkoxH + 6 flufH + 12 KOH $\overset{\mathrm{MeOH}}{\to }$
[Zn_6_(OH)_2_(dpkox)_4_(fluf)_6_]+ 12 NaCl + 10 H_2_O(82)
**100**


Single-crystal X-ray crystallography revealed that complex **100** contains molecules consisting of six Zn^II^ atoms bridged by the *syn*, *syn*-2.11 carboxylato groups of six flufenamato ligands, creating a 24-membered metallocoronate ring with a repeat –[Zn-O-C-O]- unit (Figure 82) [80]. Two 3.2110 and two 3.2111 dpkox^−^ ligands, and two μ_3_ (3.3) hydroxido groups contribute to the bridging pattern, stabilizing the six octahedral Zn^II^ atoms in a distorted trigonal antiprismatic arrangement. 

**Figure 80 molecules-30-00791-f080:**
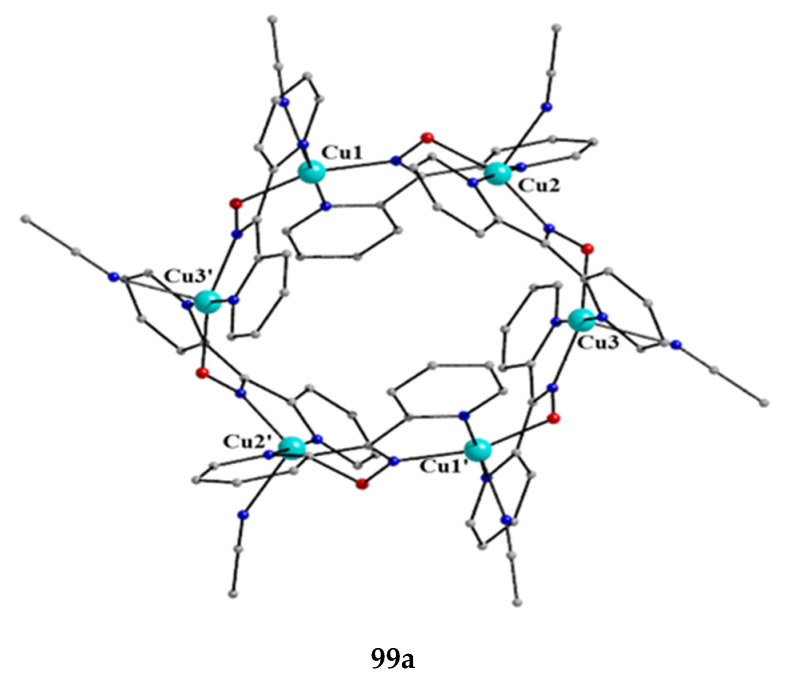
The structure of [Cu^II^_6_(dpkox)_6_(MeCN)_6_]^6+^ that is present in the cation **99a**; the encapsulated (trapped) ClO_4_^−^ of **99a** has not been drawn.

**Figure 81 molecules-30-00791-f081:**
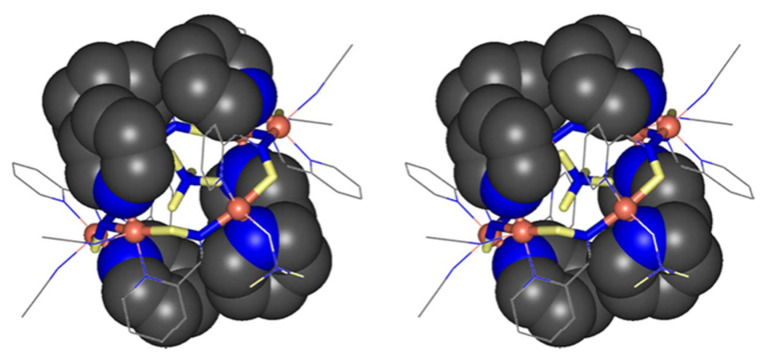
Stereoview of the cation [Cu^II^_6_(ClO_4_)_3_(dpkox)_6_(MeCN)_4_]^3+^ (**99b**) showing the cavity of the ring and the encapsulated ClO_4_^−^ ion. Reproduced from Ref. [86]. Copyright 2005 Elsevier.

**Figure 82 molecules-30-00791-f082:**
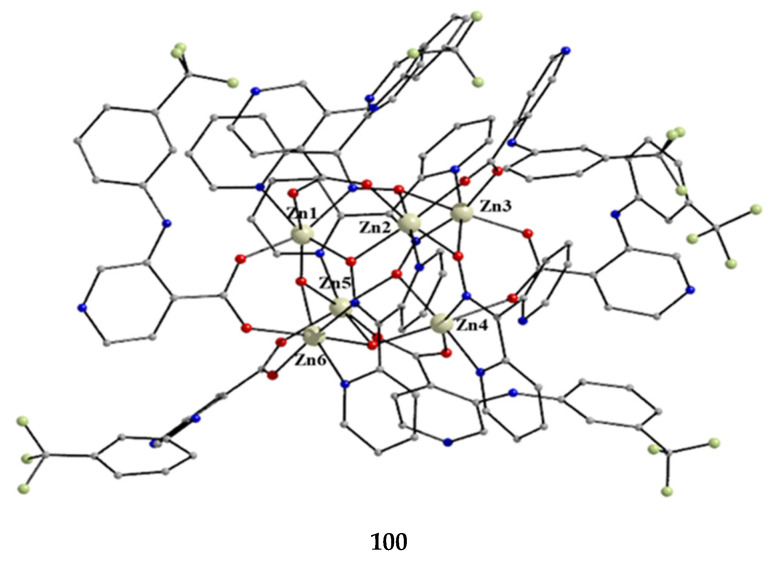
The molecular structure of [Zn_6_(OH)_2_(dpkox)_4_(fluf)_6_] (**100**).

### 5.8. A Cationic 24-MC-8 Dodecanuclear Manganese Complex with Metals Possessing Three Oxidation States

The reaction of Mn^II^(ClO_4_)_2_∙6H_2_O with dpkoxH in 1:1 molar ratio in MeOH in the presence of NaOH gave cluster 101 (Table 7), Equation (83); the yield was ~40% [88].
12 Mn^II^(ClO_4_)_2_∙6H_2_O + 12 dpkoxH + 21 NaOH + 2.5 O_2_ + 2 MeOH $\overset{\mathrm{MeOH}}{\to }$
(83)                                                                [Mn^II^_4_Mn^III^_6_Mn^IV^_2_O_6_(OH)_4_(OMe)_2_(dpkox)_12_](OH)(ClO_4_)_3_ + 21 NaClO_4_ + 87 H_2_O                                                                                                        **101**

**Figure 83 molecules-30-00791-f083:**
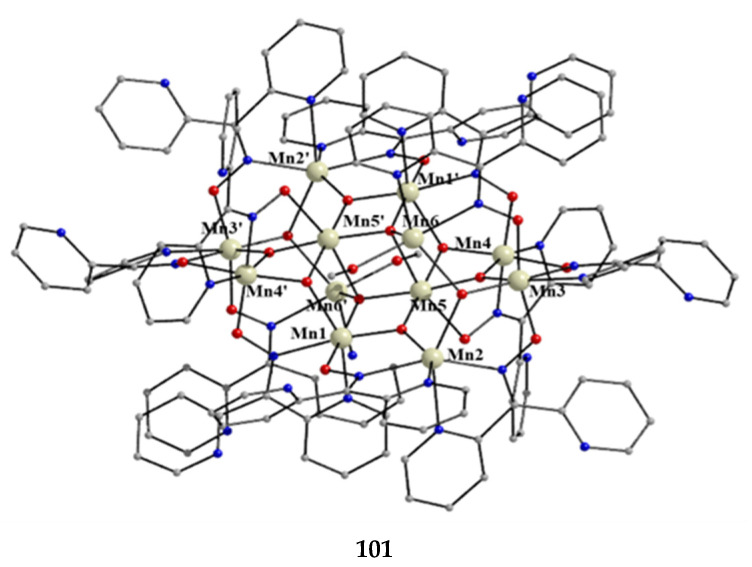
The structure of the cation [Mn^II^_4_Mn^III^_6_Mn^IV^_2_O_6_(OH)_4_(OMe)_2_(dpkox)_12_]^4+^ that is present in complex **101**.

**Table 7 molecules-30-00791-t007:** Coordination clusters of dpkoxH and dpkox^−^ with nuclearities higher than 6.

Complex	Coordination Mode of dpkoxH/dpkox^− e^	Ref.
[Mn^II^_4_Mn^III^_6_Mn^IV^_2_O_6_(OH)_4_(OMe)_2_(dpkox)_12_](OH)(ClO_4_)_3_ (**101**)	2.1110	[88]
[Ni_7_(N_3_)_2_(O_2_CMe)_6_(dpkox)_6_(H_2_O)_2_] (**102**)	1.0110, 3.2111	[77]
(N_3_)C[Ni_9_(N_3_)_9_(dpkox)_6_(dpt)_6_](ClO_4_)_2_ (**103**) ^a^	2.1110	[60]
[Ni_10_(mcpa)_2_(shi)_5_(dpkox)_3_(MeOH)_3_(H_2_O)] (**104**) ^b,c^	3.2111, 4.3111	[80,89]
[Ni_10_(2,4-D)_2_(shi)_5_(dpkox)_3_(MeOH)_3_(H_2_O)] (**105**) ^d^	3.2111, 4.3111	[90]

^a^ dpt = dipropylenetriamine. ^b^ shi is the triply deprotonated form of salicylhydroxamic acid (shiH_3_; Figure 12, **K**). ^c^ mcpa = 2-methyl-4-chlorophenoxyacetate(−1). ^d^ 2,4-D = 2,4-dichlorophenoxyacetate(−1). The structural formulae of dpt, mcpa^−^ and 2,4-D^−^ are illustrated in Figure 15. ^e^ Using the Harris notation [2]. Lattice solvent molecules have not been incorporated into the formulae of the compounds.

The structure (Figure 83) consists of a dodecanuclear cation with charge +4, which is counterbalanced by one OH^−^ and three ClO_4_^−^ ions [88]. The core of the complex is {Mn^II^_4_Mn^III^_6_Mn^IV^_2_(μ_3_-O)_4_(μ_4_-O)_2_(μ_3_-OH)_4_}^18+^. The cluster contains a 24-membered metallacrown (24-MC-8) subunit constructed by eight metal ions (two Mn^II^, four Mn^III,^ and two Mn^IV^) and eight 2.1110 dpkox^−^ ligands; more detailed descriptions of these modes are 2.1_1_1_2_1_2_0 and 2.1_3_1_1_1_1_0, where the subscripts 1, 2, and 3 refer to Mn^III^, Mn^II,^ and Mn^IV^, respectively. Therefore, this “external” part can be formally described as cationic [24-MC_Mn(II/III/IV)N(dpkox)_-8]^4+^ subunit. The specific connectivity of the ring atoms is [Mn^III^-O-N-Mn^II^-N-O-Mn^III^-N-O-Mn^IV^-O-N]_2_. Atoms Mn1, Mn1’, Mn3, Mn3’ are Mn^III^, atoms Mn2, Mn2’, are Mn^II,^ and atoms Mn4, Mn4’ are Mn^IV^. The 24-membered metallacrown ring wraps a 16-membered star-shaped {Mn^II^_2_Mn^III^_4_Mn^IV^_2_(μ_3_-O)_4_(μ_3_-OH)_4_}^12+^ ring with the connectivity pattern [Mn1-O-Mn2-(OH)-Mn3-O-Mn4-(OH)]_2_. It should be noted that the Mn centers in the two rings are common, while the heteroatoms forming the two rings come from non-related ligands. Those of the metallacrown ring come from eight dpkox^−^ N-O^−^ moieties, whereas those of the star-shaped ring from μ_4_-O^2−^ and μ_3_-OH^−^ groups. The metallacrown ring accommodates a tetranuclear mixed-valence cluster subunit of the formula {Mn^II^_2_Mn^III^_2_(μ_3_-O)_4_(μ_4_-O)_2_(μ_3_-OH)_4_(OMe)_2_(dpkox)_4_}^12−^. The dpkox^−^ anions behave as 2.1_1_1_2_1_2_0 ligands. The connection between the Mn clusters of the metallacrown ring and those of the “internal” tetranuclear subunit is achieved by μ_4_-O^2−^ and μ_3_-OH^−^ groups. The methoxido groups are terminal and form bonds to the Mn^III^ atoms of the tetranuclear mixed-valence subunit. The χ_Μ_Τ vs T plot for **101** indicates an overall antiferromagnetic interaction [88].

### 5.9. A Heptanuclear Azido-Bridged Ni(II) Cluster

Incorporation of N_3_^−^ ions into the reaction system that led to 92, Equation (73) and Figure 73, but using MeCN/H_2_O (5:1, *v*/*v*) instead of Me_2_CO/H_2_O (5:1, *v*/*v*), gave dark red crystals of [Ni_7_(N_3_)_2_(O_2_CMe)_6_(dpkox)_6_(H_2_O)_2_] (102) in 55% yield (Table 7), Equation (84) [77].
7 Ni(O_2_CMe)_2_∙4H_2_O + 6 dpkoxH + 2 NaN_3_ $\overset{\mathrm{MeCN}/H_{2}O\;\left(5:1,\;v/v\right)}{\to }$
(84)    [Ni_7_(N_3_)_2_(O_2_CMe)_6_(dpkox)_6_(H_2_O)_2_] + 6 MeCO_2_H + 2 Na(O_2_CMe) + 26 H_2_O                                 **102**

The molecular structure of **102** (Figure 84) consists of an assembly of seven octahedral Ni^II^ centers; there is a crystallographic 2-fold axis passing through Ni1. Two end-on (2.110) azido groups, four 3.2111 dpkox^−^ ligands, and six *syn*, *syn*-2.11 acetato ions hold together the seven metal ions. Peripheral ligation is provided by two 1.0110 dpkox^−^ ligands and two aqua groups [77]. Without considering the bridging acetato groups as part of the core, the latter is {Ni_7_(η^1^:μ_2_-Ν_3_)_2_(μ_3_-OΝ)_4_}^8+^. The molecule can be considered as two {Ni_3_(μ_2_-Ν_3_)(μ_2_-O_2_CMe)_2_(η^1^-O_2_CMe)(μ_3_-dpkox)(μ_2_-dpkox)(η^1^:η^1^-dpkox)}^−^ subunits, each linked to the central Ni1 atom. Variable-temperature magnetic susceptibility data and magnetization studies indicate an S = 3 ground state.

### 5.10. A Remarkable Enneanuclear Ni(II) Metallacycle

The group of Escuer prepared a {Ni_9_} dpkox^−^-based metallacycle that is capable of selective encapsulation of an azido anion in a cryptand-like cavity through H-bonding interactions [60], Equation (85); dpt is dipropylenetriamine (Figure 15). The yield of the reaction is ca. 40%.
9 Ni(ClO_4_)_2_∙6H_2_O + 6 dpkoxH + 10 NaN_3_ + 6 dpt + 6 Et_3_N $\overset{\mathrm{MeCN}}{\to }$
(85)     (N_3_)C[Ni_9_(N_3_)_9_(dpkox)_6_(dpt)_6_](ClO_4_)_2_ + 10 NaClO_4_ + 6 (Et_3_NH)(ClO_4_) + 54 H_2_O                                      **103**


The cation of **103** (Figure 85) can be described as consisting of three triangular subunits linked by μ-1,3 (end-to-end or 2.11) azido groups generating a {Ni_9_} ring. Coordination of N_2-pyridyl_, N_oximato_ chelating parts from two 2.1110 dpkox^−^ ligands to Ni1 and tridentate chelating dpt ligands to Ni2 lead to a “trimer of trimers” description of the cation. Each of the two connected Ni∙∙∙Ni edges of the triangular units is supported by a μ-1,1 (end-on or 2.20) azido group. Thus, there are three end-to-end and six end-on N_3_^−^ ligands. The {-Ni-(μ_1,1_-Ν_3_)-Ni-(μ_1,1_-Ν_3_)-Ni-(μ_1,3_-Ν_3_)-}_3_ linkage sequence defines a 24-membered ring. The ring is not planar due to the arrangement of the μ-1,3-azido bridges, exhibiting a zigzag conformation that creates a large internal prismatic cavity functionalized by six –NH_2_ groups coming from six dpt ligands. These –NH_2_ functionalities are donors of six H bonds with the terminal N atoms of one N_3_^−^ ion as acceptors; in this way, the single N_3_^−^ ion is trapped inside the cavity along the C_3_ axis. The three positive charges on the ring are compensated by the guest N_3_^−^ ion and two ionic perchlorates. The self-assembly for the formation of **103** is the combination of several factors such as charge balance and guest H-bonding interactions [54], leading to the stabilization of this unusual supramolecular system.

### 5.11. Unique {Ni_10_} Dimeric Metallacrowns

Compound **104** was prepared by the reaction represented in Equation (86) in ca. 60% yield; shiH_3_ is salicylhydroxamic acid (K, Figure 12), and mcpa is 2-methyl-4-chlorophenoxyacetate(−1) (the structural formula of this anion is illustrated in Figure 15).
10 NiCl_2_∙6H_2_O + 3 dpkoxH + 5 shiH_3_ + 2 Na(mcpa) + 18 NaOH + 3 MeOH $\overset{\mathrm{MeOH}}{\to }$
(86)                                  [Ni_10_(mcpa)_2_(shi)_5_(dpkox)_3_(MeOH)_3_(H_2_O)] + 20 NaCl + 77 H_2_O                                                                 **104**


**Figure 86 molecules-30-00791-f086:**
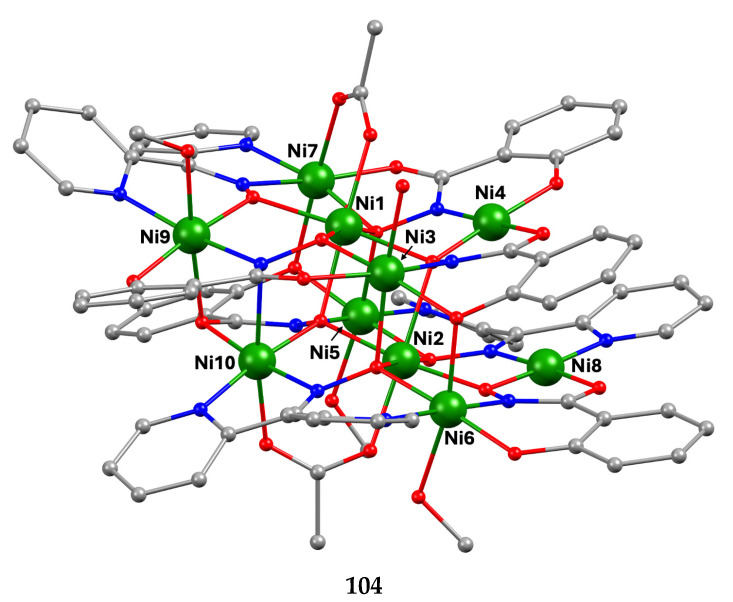
The molecular structure of [Ni_10_(mcpa)_2_(shi)_5_(dpkox)_3_(MeOH)_3_(H_2_O)] (**104**). Atoms Ni4 and Ni8 have square planar geometries.

The molecule of **104** (Figure 86) consists of two [12-MC_Ni(II)(ox)(ligand)_-4] units (ox = oximato and ligand = shi^3−^). These are {Ni^II^(mcpa)(MeOH)_2_[12-MC_Ni(II)N(shi)2(dpkox)2_-4]} and {Ni^II^(mcpa)(MeOH)(H_2_O)[12-MC_Ni(II)N(shi)3(dpkox)_-4]} with charges +1 and −1, respectively [80,89]. The shi^3−^ ligands participate with five different ligation modes (Figure 87), while the dpkox^−^ ligands are 3.2111 and 4.3111 (Figure 14). The neutral [12-MC_Ni(II)N(shi)2(dpkox)2_-4] part of the structure has alternating shi^3−^ and dpkox^−^ ligands, forming a neutral 12-MC-4 ring; all Ni^II^ atoms are octahedral, except Ni4, which can be described as square planar. An overall +1 charge results because a fifth Ni^II^ atom is encapsulated and a *syn, syn*-2.11 mcpa^−^ ligand is bound. The second metallacrown [12-MC_Ni(II)N(shi)3(dpkox)_-4] part of the molecule is anionic with -2 charge. An overall -1 charge results when a fifth Ni^II^ atom is encapsulated and a *syn*, *syn*-2.11 mcpa^−^ ligand is bound; again, all Ni^II^ atoms are octahedral, except Ni8, whose geometry is better described as square planar. Thus, each part of the molecule has four ring Ni^II^ atoms and one additional encapsulated metal ion. The coordination sphere of each ring Ni^II^ includes both five- and six-membered chelating rings in the neutral metallacrown part of the molecule. In contrast, in the anionic metallacrown part, the coordination sphere of two-ring Ni^II^ atoms includes both five- and six-membered chelating rings, one Ni^II^ atom includes only five-membered chelating rings and the fourth ring Ni^II^ atom contains only six-membered chelating rings, that form the basis of the two metallacrown parts through a [-Ni^II^-N-O-]_4_ core system. The planar configurations of both metallacrown parts allow for the encapsulation of a fifth Ni^II^ atom. The trapped Ni^II^ lies in the plane of the four O-ring atoms. The cavity radii are 0.68 and 0.70 Å for the anionic and cationic units, respectively. The magnetic “structure” of **104** is complicated, with some exchange interactions being strongly antiferromagnetic [89].

The 2,4-D^−^ analog of **104**, i.e., [Ni_10_(2,4-D)_2_(shi)_5_(dpkox)_3_(MeOH)_3_(H_2_O)] (**105**), has also been structurally characterized; 2,4-D^−^ is the anion of 2,4-dichlorophenoxyacetic acid (Figure 15) [90].

Complexes **104** and **105** exhibit strong antibacterial activity against Gram-positive and Gram-negative microorganisms [90].

**Figure 87 molecules-30-00791-f087:**
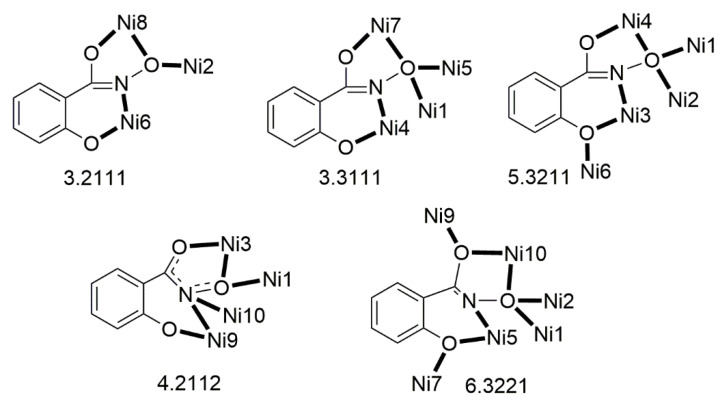
The five coordination modes of the shi^3−^ ligands in the structure of [Ni_10_(mcpa)_2_(shi)_5_(dpkox)_3_(MeOH)_2_(H_2_O)] (**104**) and the Harris notation that describes these modes. The coordination bonds are drawn with bold lines. The numbering scheme of the Ni^II^ atoms is presented for clarity. The ligation modes 4.2112 and 6.3221 are, to-date, novel in the coordination chemistry of salicylhydroxamic acid.

### 5.12. Coordination Polymers

The to-date structurally characterized polymeric complexes containing the deprotonated dpkox^−^ ligand (**106**) and the neutral dpkoxH ligand (**107**–**110**) are listed in Table 8.

Complex **106** is the only compound whose polymeric character is created (at least in part) by di-2-pyridyl ketoxime. The complex [71] was prepared by the reaction shown in Equation (87) in ca. 60% yield; (py)_2_CO is di-2-pyridyl ketone and (py)_2_C(O)_2_^2−^ is the dianion of its *gem*-diol form (Figure 15).
4n MnCl_2_∙4H_2_O + 2n dpkoxH + n (py)_2_CO + 6n NaN_3_ + 2n NaOH + 1/2n O_2_ + 2n MeOH $\overset{\mathrm{MeOH}}{\to }$
(87)                          {[Mn^II^_2_Mn^III^_2_(N_3_)_6_(dpkox)_2_{(py)_2_C(O)_2_}(MeOH)_2_]}_n_ + 8n NaCl + 18n H_2_O                                                              **106**


The repeat unit of **106** is shown in Figure 88. The topology of the octahedral metal ions is distorted tetrahedral. Atoms Mn1 and Mn3 are Mn^II^, and the rest (Mn2, Mn4) are Mn^III^ [71]. The Mn centers are linked by a series of diatomic (N-O) and O-monoatomic bridges coming from the two 2.1110 dpkox^−^ ligands, the single 4.2_1,2_2_1,2_1_2_1_2_ (the subscript 1 refers to Mn^II^ and 2 to Mn^III^) (py)_2_C(O)^2−^ ligand and three end-on (2.20) azido groups. The bidentate chelating N_2-pyridyl_, N_oximato_ part of each dpkox^−^ binds a Mn^II^ atom, while their oximato O atoms bind Mn^III^. Each of the two deprotonated O atoms of (py)_2_C(O)_2_^2−^ bridges one Mn^II^ and one Mn^III^ atom. There are three types of azido groups in the tetranuclear unit; one is terminal (1.10), three are end-on (2.20), and two are end-to-end (2.11); the latter two appear as terminal in Figure 88. Each of the end-to-end azido groups provides a bridge to a metal ion of an adjacent tetranuclear unit, thus creating the 1D chain in the solid (Figure 89); these bridges are of the {Mn^III^-N_azido_-Mn^III^} type.

Variable-field magnetic susceptibility data for **106** at 4.5 K reveal that there is not an isolated ground state [71]. Ac measurements provide the estimation of an S = 2 ground state. The compound is weak SMM. XANES spectroscopy was used to confirm the correct combination of the Mn oxidation states, i.e., {Mn^II^_2_Mn^III^_2_}. Simulation of the EXAFS data using the known crystallographic coordinates was satisfactory [71].

The Cu(I) coordination polymer **107** was prepared [51] by the reaction shown in Equation (88); the yield was low (ca. 25%). Cu^I^SCN had been prepared by the 1:1 reaction of Cu^II^(NO_3_)_2_∙3H_2_O and NaSCN in the presence of the good reductant agent L(+)-ascorbic acid.
n Cu^I^SCN + n dpkoxH $\overset{H_{2}O/\mathrm{EtOH}}{\to }$
{[Cu^I^(SCN)(dpkoxH)]}_n_(88)
**107**

**Figure 90 molecules-30-00791-f090:**
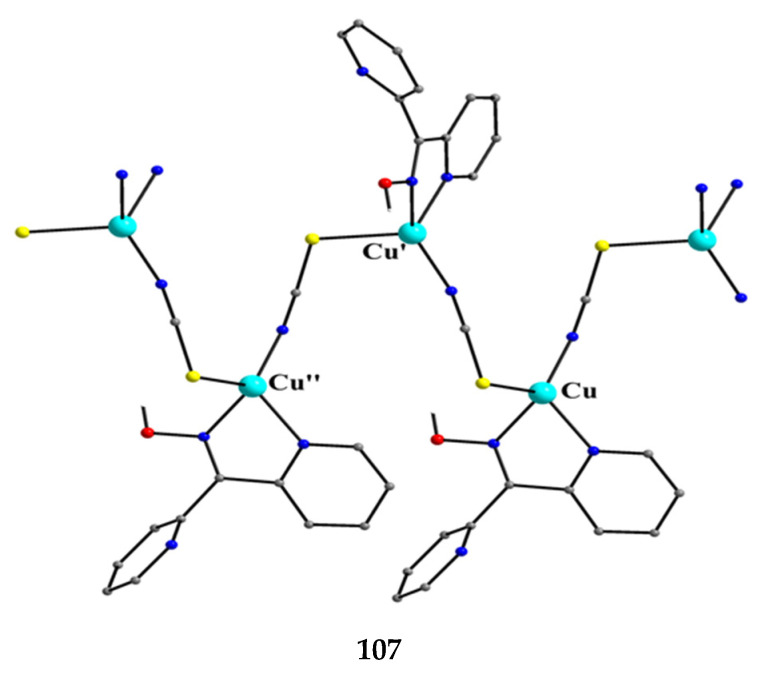
A small portion of one zigzag chain that is present in the crystal structure of {[Cu^I^(SCN)(dpkoxH)]}_n_ (**107**).

The crystal structure of **107** features tetrahedral Cu^I^ atoms singly bridged by η^1^:η^1^:μ_2_ thiocyanido ligands forming zigzag chains (Figure 90) [51]. A N_2-pyridyl,_ N_oximato_-bidentate chelating dpkoxH ligand (1.0110) completes the coordination sphere of each metal ion, which is {Cu^I^N_thiocyanido_S_thiocyanido_N_2-pyridyl_N_oximato_}. The interchain H bond, with the O atom of the oxime group as donor and the uncoordinated 2-pyridyl N atom as acceptor, creates the architecture of the crystal structure. The *ν*(C≡Ν) mode in the IR spectrum of the complex appears as a very strong band at 2098 cm^−1^. The compound does not display emission light in the solid state or in solution at room temperature, which would be due to MLCT excitation. The non-emissive character may be caused by the oxime group which alters the energy levels of the aromatic rings to make other, radionless decay mechanisms more efficient [51]. 

Our group recently studied [91] the reactions of cadmium halides and dpkoxH in order to assess the structural role (if any) of the halido ligand and compare the products with their Zn(II) analogs (Table 1). Complexes **108**–**110** were prepared by the general reaction shown in Equation (89). X = Cl, Br, I; y = 2 for X = Cl, n = 4 for X = Br, and n = 0 for X = I. The solvents were MeOH for **108** and **110**, and MeCN for **109**. The yields were 50–60%. Reactions with a large excess of the ligand, e.g., Cd(II):dpkoxH = 1:3, led again to the isolation of the 1:1 complexes.
n CdX_2_∙yH_2_O + n dpkoxH $\overset{solvent}{\to }$
{[CdX_2_(dpkoxH)]}_n_+ ny H_2_O(89)
**108**–**110**


The structure of the compounds consists of structurally similar 1D zigzag chains, but only **109** (Figure 91) and **110** are strictly isomorphous. Neighboring metal centers are alternately doubly bridged by halido (or halo) and neutral dpkoxH ligands, the latter adopting the 2.0111 coordination mode. A terminal halido group completes distorted octahedral coordination at each Cd^II^ atom; thus, the coordination sphere of the metal ions is {Cd(η^1^-Χ)(μ_2_-Χ)_2_(Ν_2-pyridyl_)_2_(N_oxime_)}. The *trans*-donor-atom pairs in **108** are Cl_terminal_/N_oxime_ and two Cl_bridging_/N_2-pyridyl_; on the contrary, these donor-atom pairs are X_terminal_/N_2-pyridyl_, X_bridging_/N_oxime_ and X_bridging_/N_2-pyridyl_ for **109** and **110**.

The solid-state structures of the complexes are not retained in DMSO, as proven via NMR (^1^H, ^13^C, and ^13^Cd NMR) spectroscopy and molar conductivity data. The complexes completely release the coordinated dpkoxH ligand from the coordination sphere of Cd^II^. The dominant solution species are [Cd(DMSO)_6_]^2+^ for **108** and **109**, and [CdI_2_(DMSO)_4_] in the case of **110**, Equations (90) (X = Cl, Br) and (91). The ^113^Cd NMR spectra of the complexes are shown in Figure 92.
{[CdX_2_(dpkox)]}_n_ + 6n DMSO $\overset{\mathrm{DMSO}}{\to }$ n [Cd(DMSO)_6_]^2+^ + 2n X^−^ + n dpkoxH (90)     **108**, **109**              {[CdI_2_(dpkox)]}_n_ + 4n DMSO $\overset{\mathrm{DMSO}}{\to }$ n [CdI_2_(DMSO)_4_] + n dpkoxH (91)                         **110**

## 6. Unpublished Results from Our Group

Our research activity towards the completion of the “Periodic Table” of dpkoxH continues. At the time of submission of this review we have at hand almost 50 structurally characterized complexes of dpkoxH and dpkox^−^ which remain unpublished. In this section we give a brief overview of *some* of our results; a few of the compounds are listed in Table 9. The compounds discussed below do not necessarily appear in this table. Since this material is unpublished, we refrain from giving details and presenting many figures.

The Cd(II)/dpkoxH reaction system is extremely fertile. In addition to the 1D compounds **108**–**110**, the 1:2 reactions between Cd(NO_3_)_2_∙4H_2_O and dpkoxH in MeCN gave two different products depending on the crystallization conditions. Storage of the reaction solution at room temperature gave complex **111**, Equation (92), while layering the reaction solution with Et_2_O gave complex **112**, Equation (93). The products were originally characterized by microanalyses (C, H, N), single-crystal X-ray crystallography and vibrational spectroscopy (IR, Raman).
Cd(NO_3_)_2_∙4H_2_O + 2 dpkoxH $\overset{\mathrm{MeCN}}{\to }$ [Cd(1.100-NO_3_)_2_(dpkoxH)_2_] + 4 H_2_O (92)                                                                                            **111**Cd(NO_3_)_2_∙4H_2_O + 2 dpkoxH $\overset{\mathrm{MeCN}/{\mathrm{Et}}_{2}O}{\to }$ [Cd(1.110-NO_3_)_2_(dpkoxH)_2_] + 4 H_2_O (93)                                                                                                  **112**

**Table 9 molecules-30-00791-t009:** A few unpublished metal complexes of dpkoxH and dpkox^−^ from our group.

Complex	Coordination Mode of dpkoxH/dpkox^− c^	Ref.
[Hg^II,II^_2_(O_2_CMe)_4_(dpkoxH)_2_] (**113**)	2.0111	[92]
[Hg^II,II^_2_(O_2_CPh)_4_(dpkoxH)_2_] (**114**)	2.0111	[92]
[Hg^II,II^_2_(O_2_CPh)_3_(dpkox)(dpkoxH)_2_] (**115**)	1.0110	[92]
{[Hg^II^Cl(O_2_CPh)(dpkoxH)]}_n_ (**116**)	2.0111	[92]
[Pr_4_(NO_3_)_8_(dpkox)_4_] (**117**)	^a^	[93]
[Pr_4_(OH)_2_(NO_3_)_4_(dpkox)_6_(MeOH)_2_] (**118**) ^b^	2.1110, 2.2110, 3.2110	[93]
[Pr_4_(OH)_2_(NO_3_)_4_(dpkox)_6_(EtOH)_2_] (**119**) ^b^	2.1110, 2.2110, 3.2110	[93]
[Pr^III^_8_Pr^IV^O_4_(OH)_4_(NO_3_)_4_(dpkox)_12_(H_2_O)_4_] (**120**)	2.1110, 2.2110, 3.2111	[93]

^a^ The formula of this tetranuclear cluster has been derived from a poor-quality single-crystal X-ray structure, and hence, we avoid specifying the coordination mode(s) of dpkox^−^. ^b^ These clusters are diastereoisomers depending on the relative positions of the nitrato ligands in the coordination sphere of the Pr^III^ center. ^c^ Using the Harris notation [2]. Lattice solvent molecules have not been incorporated into the formulae of the compounds.

The molecular structures of **111** and **112 [91]** are shown in Figure 93. The Cd^II^ atom in **111** is 6-coordinate with a distorted octahedral stereochemistry. The metal ion in **112** is 8-coordinate with a distorted hexagonal bipyramidal geometry. In both complexes, each dpkoxH molecule behaves as an N_2-pyridyl_, N_oxime_-bidentate chelating ligand (1.0110). The *trans*-nitrato groups are monodentate (1.100) in **111**, whereas the *cis*-nitrato groups are bidentate chelating (1.110) in **112**. Thus, the two complexes exhibit a type of linkage isomerism; since the donor atoms of the nitrato ligands are exclusively oxygens (i.e., this group is not ambidentate), we have termed these complexes as “coordination number isomers”. Since the vibrational (IR, Raman) spectra of the samples from the two preparations are identical, we suspected that each sample was practically a mixture of the two isomers. This would also be expected on synthetic grounds since the only difference in the two preparations was the absence (**111**) or presence (**112**) of Et_2_O. Our fears proved true, because the solid-state ^113^Cd NMR spectra of the two samples (Figure 94) have repeatedly shown a striking similarity with the main peak (attributed to one isomer) at *δ* 209 ppm. The spectra also show the presence of another minor Cd^II^ site (attributed to the other isomer). DFT calculations are in progress to discover the predominant isomer in the samples (**111** or **112**) and to find the synthetic–crystallization conditions for the isolation of the pure isomers.

The study of the Hg(II) carboxylate chemistry with dpkoxH has resulted in dinuclear and polymeric complexes in yields of 45–60%, Equations (94)–(97).
2 Hg^II^(O_2_CMe)_2_∙2H_2_O + 2 dpkoxH $\overset{MeOH}{\to }$ [Hg^II,II^_2_(O_2_CMe)_4_(dpkoxH)_2_] + 4 H_2_O (94)                                                                                       **113**2 Hg^II^(ClO_4_)_2_∙3H_2_O + 4 Na(O_2_CPh) + 2 dpkoxH $\overset{\mathrm{MeOH}}{\to }$ [Hg^II,II^_2_(O_2_CPh)_4_(dpkoxH)_2_] + 4 NaClO_4_ + 6 H_2_O (95)                                                                                                            **114**2 Hg^II^Cl_2_ + 4 Na(O_2_CPh) + 2 dpkoxH + MeOH $\overset{MeOH}{\to }$
(96)                                [Hg^II,II^_2_(O_2_CPh)_3_(dpkox)(dpkoxH)(MeOH)] + 4 NaCl + PhCO_2_H                                                              **115**n Hg^II^Cl_2_ + n Na(O_2_CPh) + dpkoxH $\overset{\mathrm{MeOH}}{\to }$ {[Hg^II^Cl(O_2_CPh)(dpkoxH)]}_n_ + n NaCl (97)                                                                                                **116**

The structures of **113** and **114** are similar. The Hg^II^ atoms are bridged by two head-to-tail 2.0111 dpkoxH ligands; a bidentate chelating and a monodentate carboxylato ligands complete octahedral coordination at each metal ion. The molecular structure of **115** is analogous to that of **114**; the only differences are the presence of one 2.0111 dpkoxH and one 2.0111 dpkox^−^ ligands, and the replacement of one monodentate PhCO_2_^−^ group with one terminal MeOH molecule at one of the Hg^II^ centers in **115**. Complex **116** is a zigzag 1D coordination polymer. The Hg^II^ atoms are alternately bridged by two 2.0111 dpkoxH ligands in a head-to-tail arrangement and two μ_2_-chlorido (2.2) groups. 

In the last 2–3 years, our group has been studying the coordination chemistry of dpkoxH with lanthanoid(III) ions [93]. Four of the more than 30 structurally characterized complexes (we confine our discussion to one metal, namely Pr) are listed in Table 9. Compounds **117**–**120** were prepared in low yields by the reactions shown in balanced Equations (98)–(100); R = Me, Et.
4 Pr(NO_3_)_3_∙6H_2_O + 4 dpkoxH + 4 (Bu^n^_4_N)(OMe) $\overset{MeCN}{\to }$
(98)                                                  [Pr_4_(NO_3_)_8_(dpkox)_4_] + 4 (Bu^n^_4_N)(NO_3_) + 4 MeOH + 24 H_2_O                                                                  **117**4 Pr(NO_3_)_3_∙6H_2_O + 6 dpkoxH + 8 (Bu^n^_4_N)(OR) $\overset{\mathrm{MeCN}}{\to }$
(99)                               [Pr_4_(OH)_2_(NO_3_)_4_(dpkox)_6_(ROH)_2_] + 8 (Bu^n^_4_N)(NO_3_) + 6 ROH + 22 H_2_O                                                    **118**, **119**9 Pr^III^(NO_3_)_3_∙6H_2_O + 12 dpkoxH + 23 Et_3_N + ¼ O_2_ $\overset{EtOH}{\to }$
(100)                   [Pr^III^_8_Pr^IV^O_4_(OH)_4_(NO_3_)_4_(dpkox)_12_(H_2_O)_4_] + 23 (Et_3_NH)(NO_3_) + 42.5 H_2_O                                                   **120**

The X-ray dataset for a crystal of **117** was poor and only the connectivity and the gross structural features are visible; we thus refrain from briefly describing its molecular structure. Complexes **118** and **119** are isostructural. The molecular structure of the former is shown in Figure 95. The complex is neutral and the centrosymmetric molecule consists of four Pr^III^ atoms with a butterfly arrangement, bridged by two μ_3_ hydroxido groups. There are six dpkox^−^ ligands which comprise two 2.1110, two 2.2110, and two 3.2110 binding modes. Ten coordination around Pr2 and Pr2’ is completed by two chelating (1.110) nitrato groups, while one MeOH molecule is bound to each of the 9-coordinate Pr1 and Pr1’ centers. The carbon-nitrogen and nitrogen-oxygen bond lengths of the oximato groups in the 2.2110 dpkox^−^ ligands, which form the three-membered chelating ring, suggest delocalization (
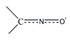
).

Complex **120** (Figure 96) is neutral and crystallizes in the tetragonal space group *I*4_1_/*a* with *Z* = 4. It consists of three crystallographically independent Pr^III^ atoms (Pr1, Pr2, Pr3) and contains one 4-fold axis which passes through the central Pr1 atom. The eight Pr^III^ atoms and the unique Pr^IV^ center (Pr1) are held together by four μ_3_-O^2−^, four μ_3_-OH^−,^ and twelve dpkox^−^ ligands; the latter are arranged in four 2.1110, four 2.2110, and four 3.2111 groups. The coordination sphere of each Pr3 atom is completed with one chelating nitrato group and one terminal H_2_O molecule. The inorganic {Pr^III^_8_Pr^IV^(μ_3_-O)_4_(μ_3_-OH)_4_}^16+^ core (Figure 97) consists of four “butterflies”, the central Pr^IV^ atom being a common site for them. It is remarkable that the {Pr^IV^(μ_3_-O)_4_(μ_3_-OH)_4_}^8−^ moiety contains only “hard” (HSAB) oxido- and hydroxido-O atoms. This is no doubt the reason for the stabilization of the IV oxidation state of Pr1. Mixed valency is only known in praseodymium oxides and not in molecular compounds. The carbon-nitrogen and nitrogen-oxygen bond lengths in the three-membered chelating ring of the 2.1110 dpkox^−^ ligands are 1.399 and 1.278 Å, respectively, indicating a –C-N=O description of the oximato group. The confirmation of this unusually high (for molecular complexes) Pr(IV) oxidation state was achieved by EPR spectroscopy and BVS calculations.

We mentioned in paragraph 5.1 that an interesting aspect of the chemistry of dpkoxH is its activation by 3d-metal ions, resulting in complexes that often possess a novel ligand different than dpkoxH. From the aerobic mixtures, Pr(NO_3_)_3_∙6H_2_O/dpkoxH, with or without Et_3_N in MeCN or MeNO_2,_ were subsequently isolated as pale green crystals of various solvates of complex [Pr_2_(NO_3_)_4_(L)_2_] in low yields (~15%). The structure of one such complex (compound **121**) is shown in Figure 98. L^−^ represents the anionic ligand shown in Figure 99.

The ligand was derived from the in situ transformation of dpkoxH, which is relatively sensitive to hydrolysis in basic media. In our case, half of the dpkoxH molecules undergo deprotonation, while the other half undergo hydrolysis to form the ketone followed by nucleophilic attack of dpkox^−^ at the positive (δ+) carbonyl carbon. The process seems to be Pr^III^-mediated with the metal ion polarizing the carbonyl group by coordination of its O atom and/or the 2-pyridyl N atoms (in solution). Molecule **121** consists of two Pr^III^ atoms doubly bridged by the methoxido O atoms of the 2.2011110 L^−^ ligands in a head-to-head arrangement. Two chelating nitrato groups complete tetradecahedral ten-coordination at each metal center. 

**Figure 98 molecules-30-00791-f098:**
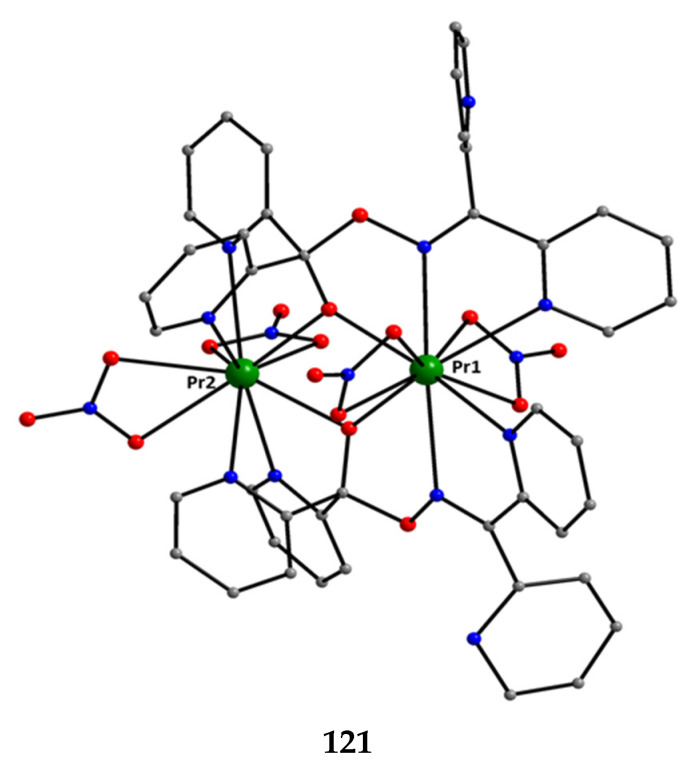
The molecular structure of one of the solvates of [Pr_2_(NO_3_)_4_(L)_2_] (**121**); the structural formula of the anionic ligand L is illustrated in Figure 99.

**Figure 99 molecules-30-00791-f099:**
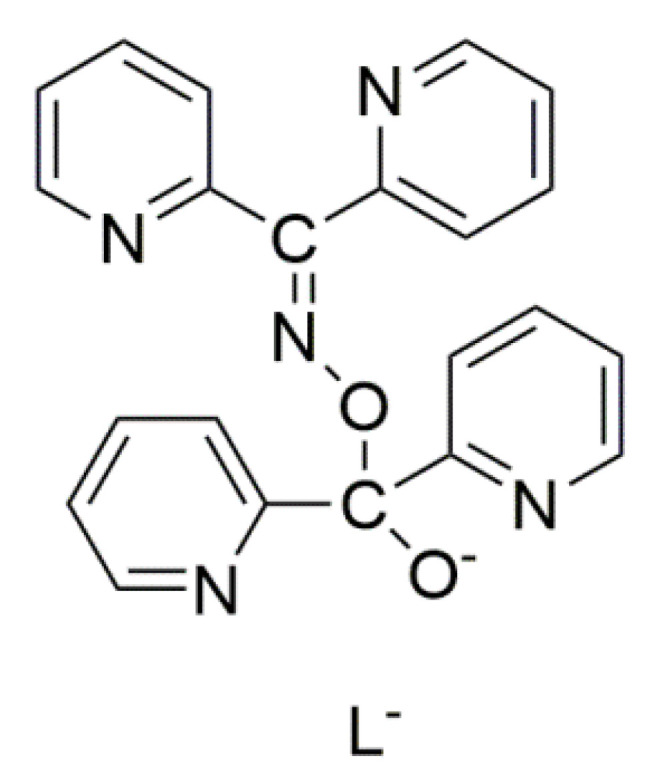
The structural formula and the abbreviation of the anionic ligand that is present in [Pr_2_(NO_3_)_4_(L)_2_] (**121**).

There is one Au(III)/dpkoxH complex in the literature [42], Equation (8), Figure 25, Table 1. This has the formula [Au^III^Cl(dpkoxH)_2_] (**23**) with a five-coordinate Au^III^ center and was prepared in H_2_O. Our group studied the Au(III)/dpkoxH reaction system in organic solvents. In refluxing MeOH, the product is [Au^I^Cl(dpkoxH)] (**122**), which was isolated in moderate yield; the starting material was H[Au^III^Cl_4_]. The Au(III)→Au(I) reduction is presumably the result of the use of MeOH (solvent), whose reduction ability increases with temperature. The crystal structure of the complex consists of [Au^I^Cl(dpkoxH)] molecules (Figure 100) which have a neutral 1.0110 dpkoxH ligand and a terminal chlorido group. The molecule has a T-shape; the N_2-pyridyl_-Au^I^-N_oxime_, N_oxime_-Au^I^-Cl, and N_2-pyridyl_-Au^I^-Cl bond angles are 72.9, 112.5, and 174.5°, respectively. The Au^I^ atom is exactly coplanar with the three donor atoms.

**Figure 100 molecules-30-00791-f100:**
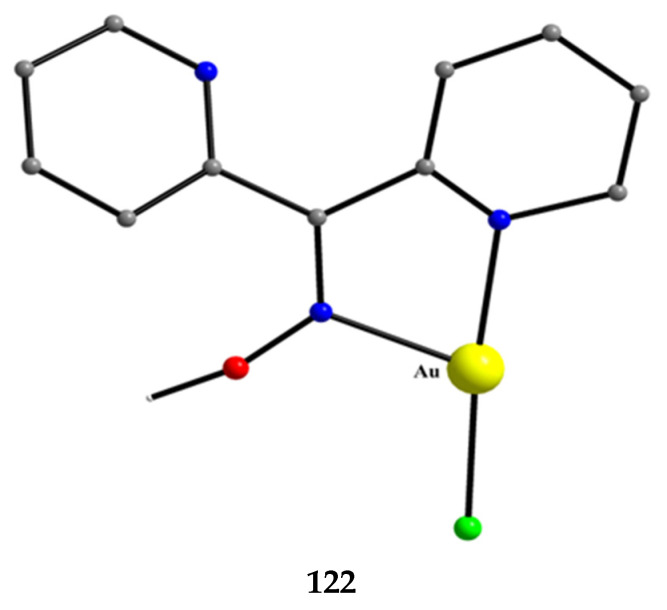
The molecular structure of [Au^I^Cl(dpkoxH)] (**122**).

## 7. Concluding Comments and Perspectives

In this report, we have comprehensively reviewed the to-date published coordination chemistry of dpkoxH (Section 5) and briefly mentioned a few unpublished results from our group (Section 6). According to our opinion, this ligand is very interesting, having a variety of exciting scientific aspects, from important synthetic and reactivity metal chemistry to interesting molecular structures and to remarkable properties (magnetic, biological, etc.). We hope that we have convinced the reader that the coordination chemistry of dpkoxH is distinctly different from that of other simple 2-pyridyl oximes (Figure 3, left/R = H, Me, Ph, and Figure 10). The presence of an extra donor group (the second 2-pyridyl ring) offers extra coordinating possibilities (Figure 14), resulting in a variety of mononuclear, dinuclear, oligomeric (coordination clusters), and polymeric complexes. It is also remarkable that the ligand can often adopt two (or sometimes three) coordination modes in the same complex providing unusual structures. The to-date developed “periodic table” of dpkoxH is shown in Figure 101.

There appear to be no solid rules concerning the way in which metal centers with different ionic radii react with dpkoxH. In general, the larger metal ions of the 2^nd^- and 3^rd^-row transition series form mononuclear and dinuclear complexes, as well as a much smaller number of coordination clusters. It is tentatively proposed that large sizes of the metal centers decrease the coordination flexibility of dpkox^−^. A final factor that influences the reactivity of the metal ions is the nature and coordination possibilities of the ancillary organic and inorganic ligands that are present in the reaction media. As a general remark, the deprotonated di-2-pyridyl ketoxime favors higher nuclearity species than the neutral ligand.

**Figure 101 molecules-30-00791-f101:**
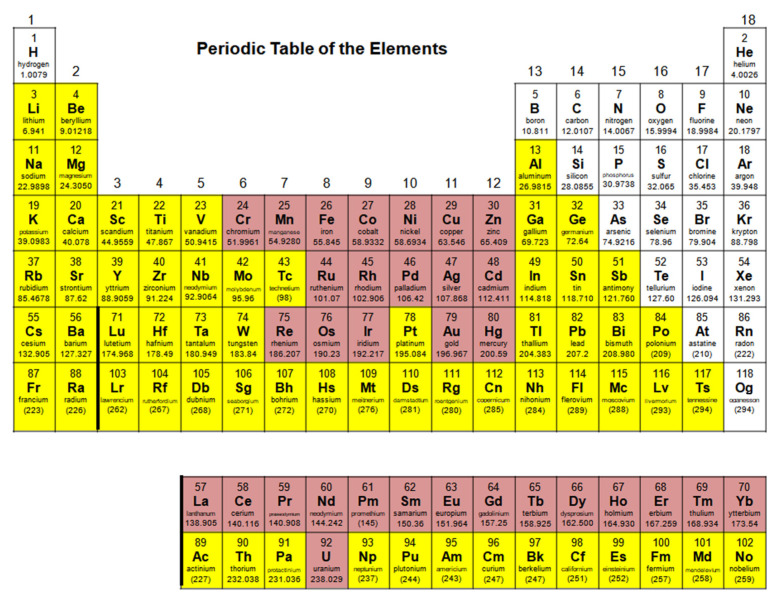
The present form of the “periodic table” of dpkoxH. Color code: Pale red; metals whose complexes with dpkoxH and/or dpkox^−^ have been published, or structurally characterized by our group and remain unpublished. Yellow; metals with unpublished coordination chemistry of dpkoxH/dpkox^−^.

The perspectives of the use of dpkoxH as a primary ligand in inorganic chemistry appear brilliant. Other perspectives are as follows: (a) We easily notice in Figure 101 that no p-metal complexes, e.g., with Ga(III), In(III), Sn(IV), Pb(II), Sb(III), Bi(III), etc., of this ligand have been reported, and work on this subarea might be interesting. (b) With the exception of the trinuclear {M^II^Ln^III^_2_} clusters (M = Ni, Cu, Pd) of Table 4 and complex **90**, the heterometallic chemistry based on dpkox^−^ is limited, and further studies are required. (c) The characterization of complexes **67**–**70** (Table 5), **L**, **M**, **N,** and **O** (Section 5.1, Figure 13) and **121** (Section 6) emphasizes the capability of dpkoxH to undergo metal ion-assisted/promoted transformations and thus further investigations on this topic are warranted; and (d) no 5f-metal chemistry of dpkoxH (with the “non-dangerous” actinoids) is known, and this area might be fruitful.

We currently work hard to fill in as many as possible empty positions in the table of Figure 101. This means that our efforts to complete the “periodic table” of di-2-pyridyl ketoxime are continued. If we are successful, a future review in “Molecules” will have the title “Completion of the ‘Periodic Table’ of Di-2-pyridyl Ketoxime”. To prove the great potential of dpkoxH, we close this comprehensive article by mentioning that during the days of the submission of this work: (1) Heterometallic dpkox^−^ -based {Cu^II^Fe^III^} and {Ni^II^Fe^III^} clusters have been prepared [perspective (b) above]; (2) in the uranyl complex [U^VI^O_2_(dpkox)_2_(MeOH)_2_] just structurally characterized [perspective [d] above], the anionic ligand adopts the hitherto not reported coordination mode 1.1010 (Figure 102).

## Figures and Tables

**Figure 1 molecules-30-00791-f001:**
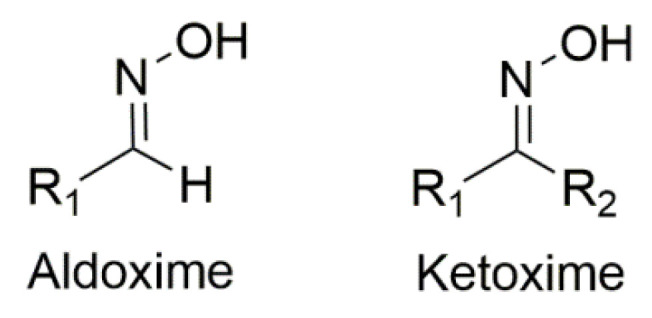
The structural formulae of aldoximes and ketoximes.

**Figure 2 molecules-30-00791-f002:**
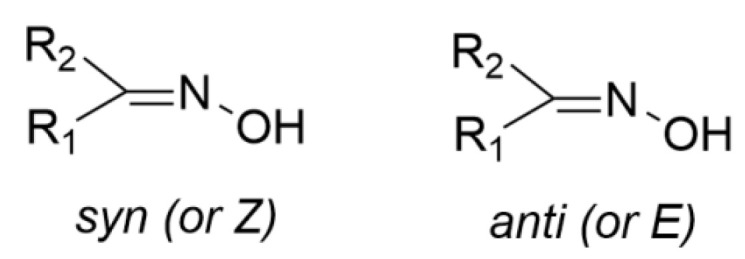
The *syn*-*anti* isomerism of the simple oxime group assuming that R_1_ takes precedence over R_2_ according to the Cahn-Ingold-Prelog system.

**Figure 3 molecules-30-00791-f003:**
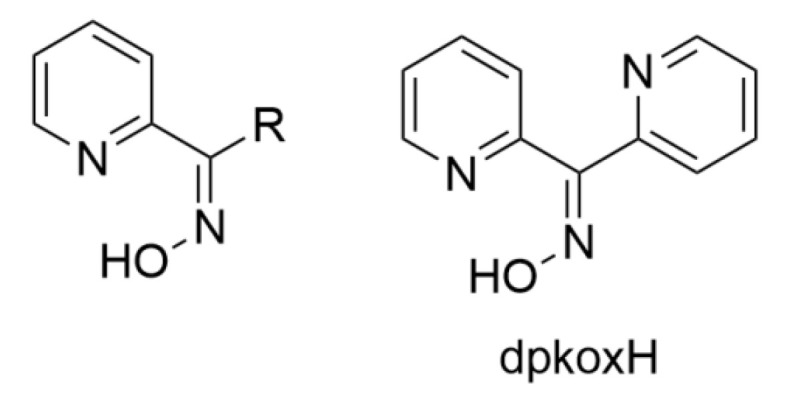
The structural formulae of simple 2-pyridyl (aldo)ketoximes (**left**; R = H, Me, Ph, …) and di-2-pyridyl ketoxime.

**Figure 4 molecules-30-00791-f004:**
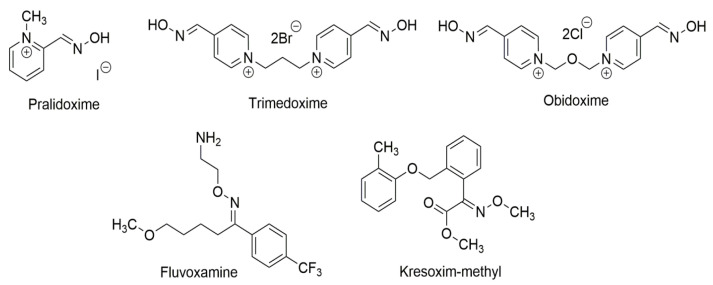
Few oxime derivatives which find use in medicine and agriculture (see text for details).

**Figure 5 molecules-30-00791-f005:**
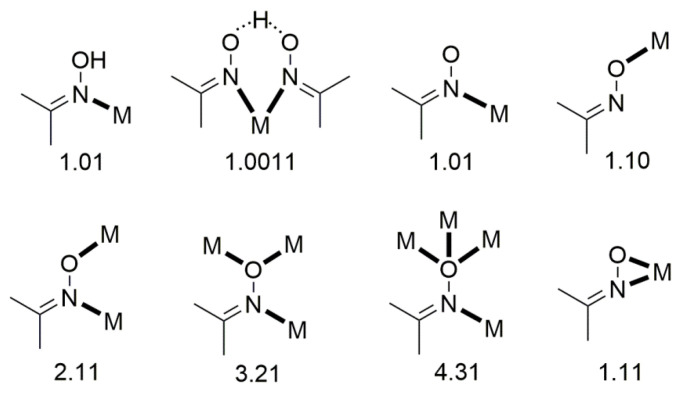
The to-date crystallographically observed coordination modes of the oxime and oximate groups, and the Harris notation [2] that describes these modes. The 1.0011 ligation mode represents one formally oximate group and one formally neutral oxime group; this moiety is present in some complexes, including bis(dimethylglyoximato)nickel(II).

**Figure 6 molecules-30-00791-f006:**
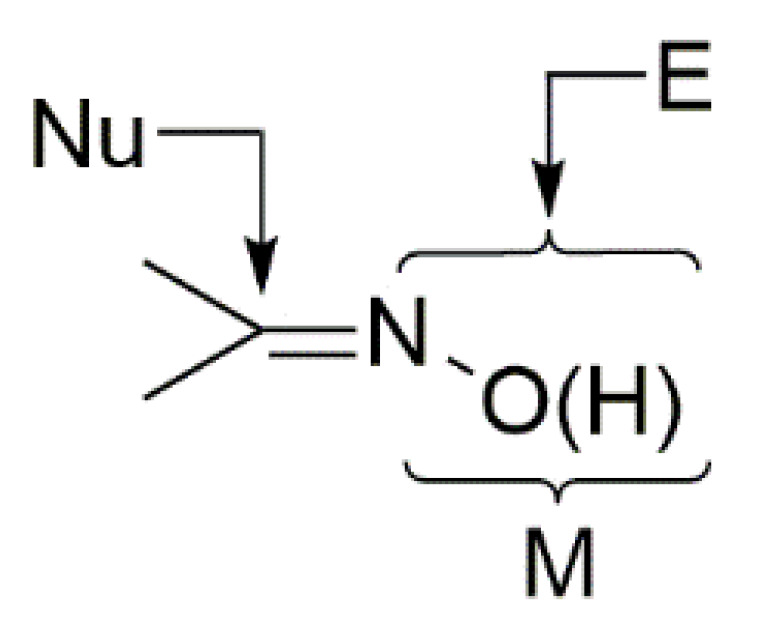
Reactivity modes of the coordinated oxime or oximate groups. M = metal ion, Nu = nucleophile, E = electrophile.

**Figure 10 molecules-30-00791-f010:**
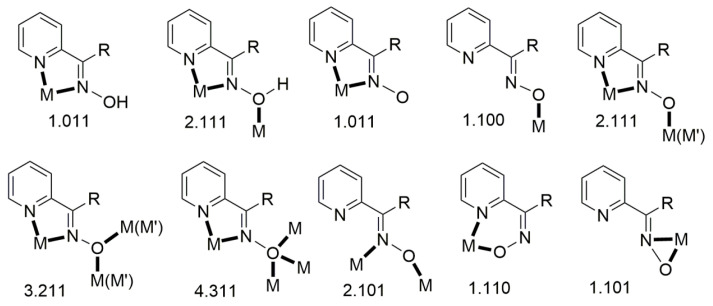
The to-date crystallographically confirmed coordination modes of neutral and anionic 2-pyridyl oximes, and the Harris notation [2] that describes these modes. R (H, Me, Ph, …) is a non-donor group. The 2.111 mode of neutral oximes and the 4.311 mode of the 2-pyridyl oximate ligands have been found only in cases where R = H. M and M’ are metal ions.

**Figure 11 molecules-30-00791-f011:**
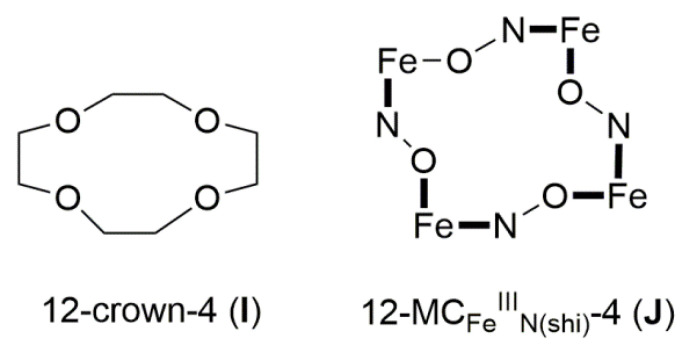
An example of the MC-crown ether analogy. The symbol shi represents the trianion of salicylhydroxamic acid (shiH_3_, Figure 12). The metal ion can be only a first-row transition metal ion. The drawing was adapted from Ref. [3].

**Figure 12 molecules-30-00791-f012:**
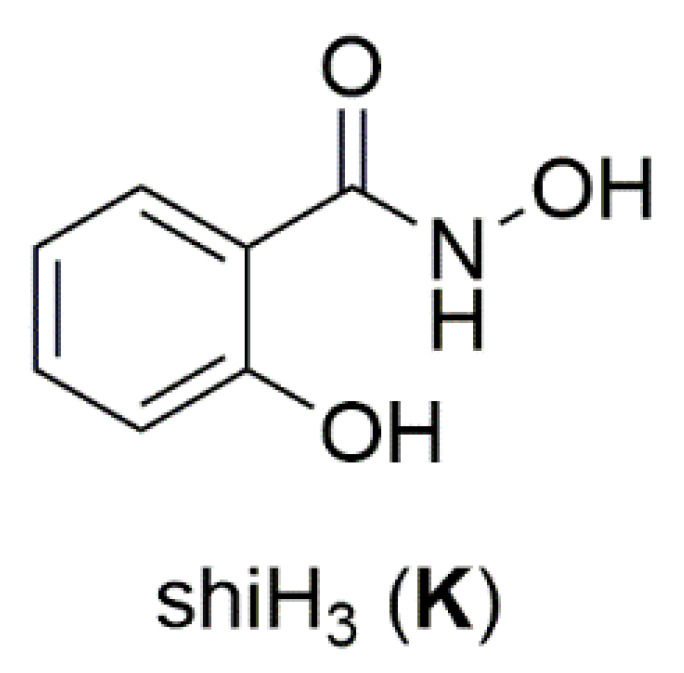
The structural formula and abbreviation of salicylhydroxamic acid, one of the most commonly used ligands for the construction of MCs (vide infra).

**Figure 13 molecules-30-00791-f013:**
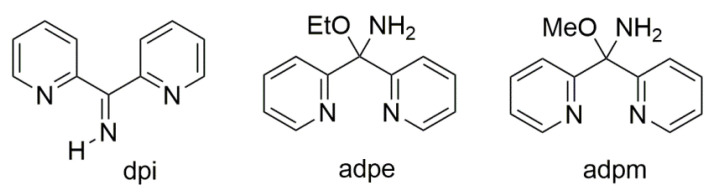
The molecules dpi, adpm, and adpe that have resulted from the synthetic investigation of the [V^III^Cl_3_(THF)_3_]/dpkoxH reaction system. The molecules are coordinated in the {V^IV^O}^2+^ (vanadyl) complexes **L**, **M**, **N,** and **O** as described in the text.

**Figure 14 molecules-30-00791-f014:**
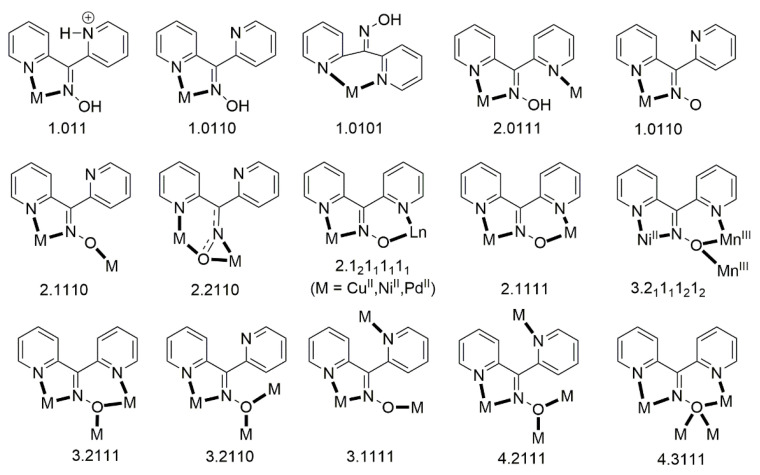
The to-date crystallographically observed coordination modes of dpkoxH_2_^+^, dpkoxH, and dpkox^−^, and the Harris notation [2] that describes these modes. M is a metal ion and Ln is a trivalent lanthanoid.

**Figure 15 molecules-30-00791-f015:**
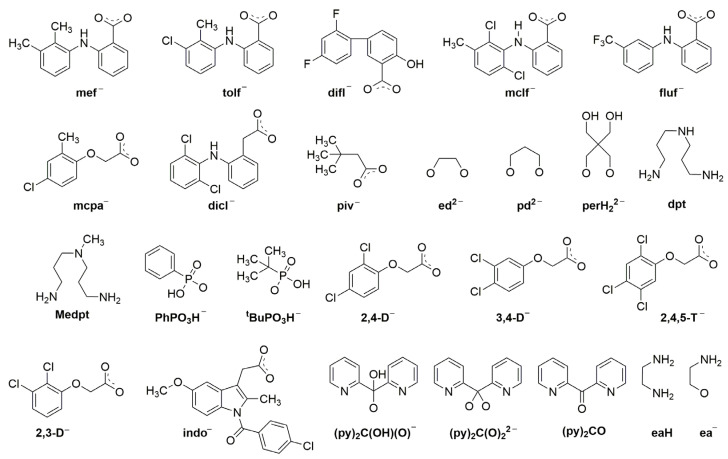
The structural formulae of most of the ancillary ligands discussed in the text. For the anionic ligands, the negative charge is indicated only in their abbreviations and not at the corresponding atoms in the structural formulae.

**Figure 19 molecules-30-00791-f019:**
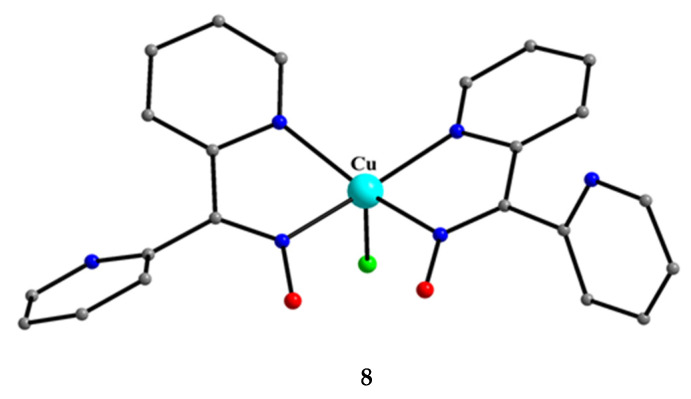
The molecular structure of [Cu^II^Cl(dpkox)(dpkoxH)] (**8**).

**Figure 20 molecules-30-00791-f020:**
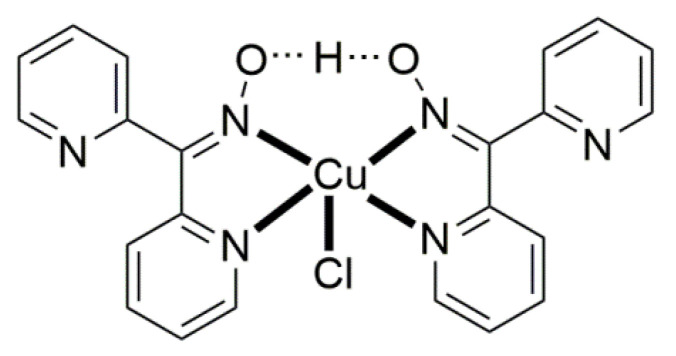
The N_4_-tetradentate chelating viewpoint of the moiety (dpkox∙∙∙H∙∙∙dpkox)^−^ in complex [Cu^II^Cl(dpkox)(dpkoxH)] (**8**).

**Figure 25 molecules-30-00791-f025:**
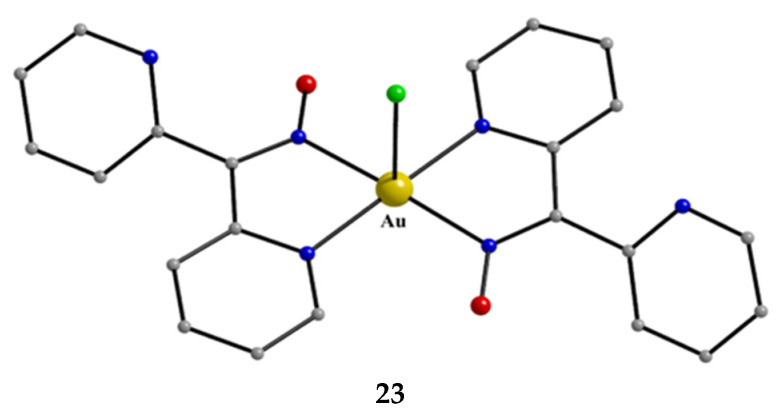
The molecular structure of [Au^III^Cl(dpkox)_2_] (**23**).

**Figure 26 molecules-30-00791-f026:**
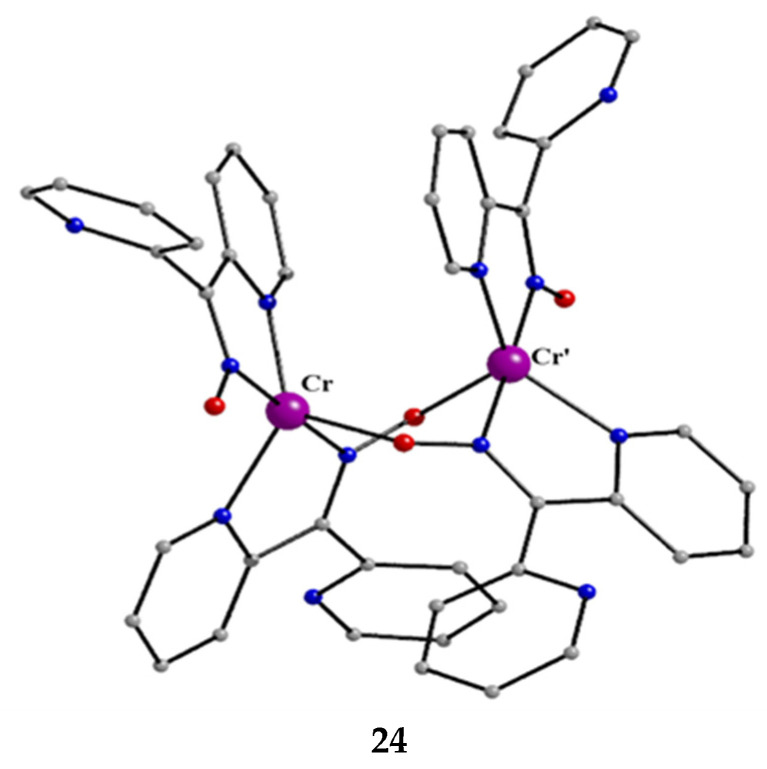
The structure of the molecule [Cr^II,II^_2_(dpkox)_4_] in the crystal of **24**.

**Figure 59 molecules-30-00791-f059:**
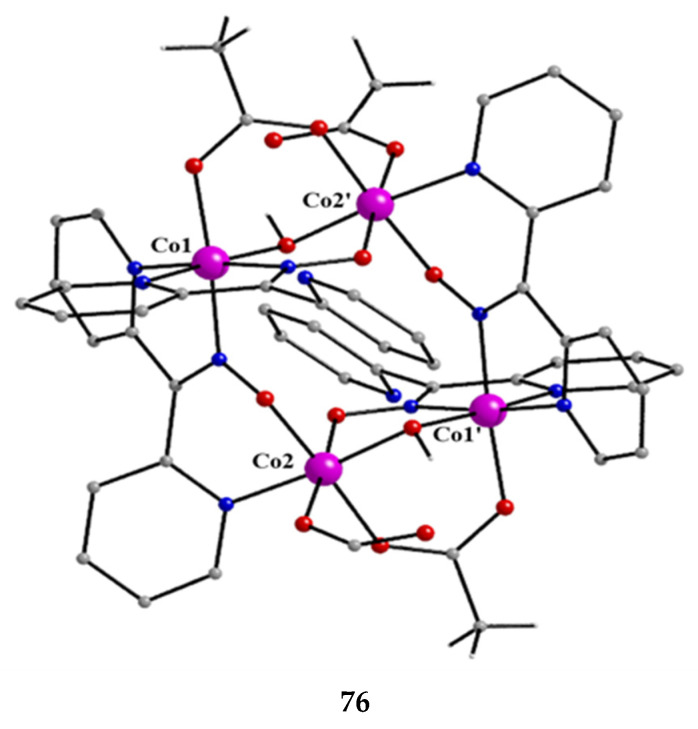
The structure of the cation [Co^III^_4_(OH)_2_(O_2_CMe)_4_(dpkox)_4_]^2+^ that is present in complex **76**. The H atoms of the hydroxidο and acetato ligands have been drawn.

**Figure 61 molecules-30-00791-f061:**
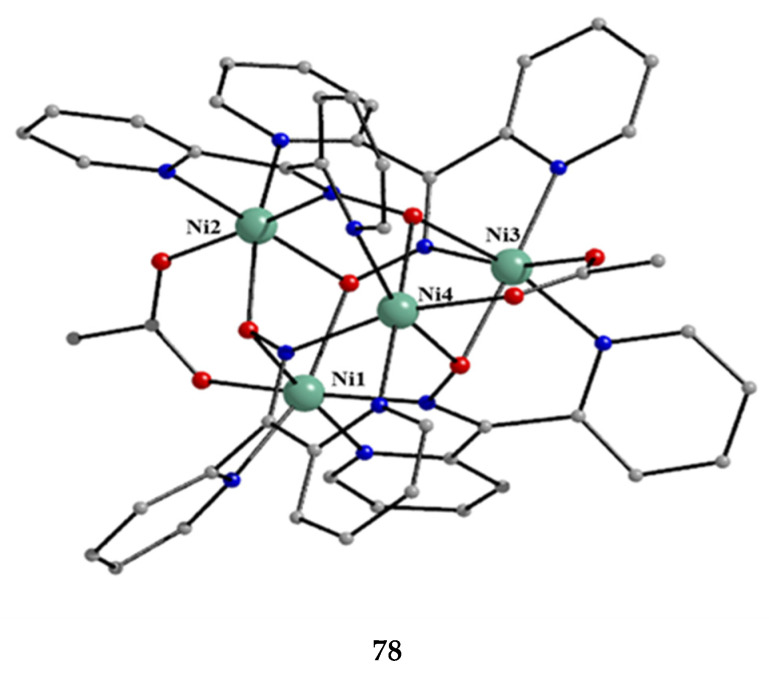
The structure of the cation [Ni_4_(O_2_CMe)_2_(dpkox)_4_]^2+^ that is present in the crystal of **78**.

**Figure 62 molecules-30-00791-f062:**
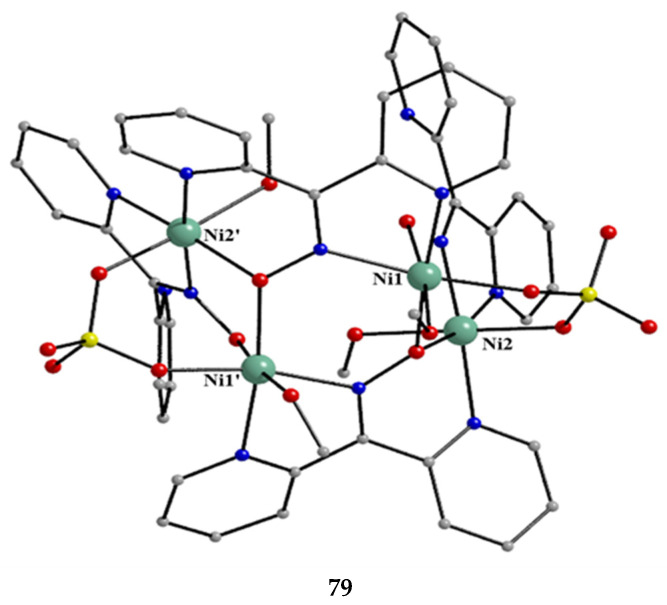
The molecular structure of [Ni_4_(SO_4_)_2_(dpkox)_4_(MeOH)_4_] (**79**).

**Figure 63 molecules-30-00791-f063:**
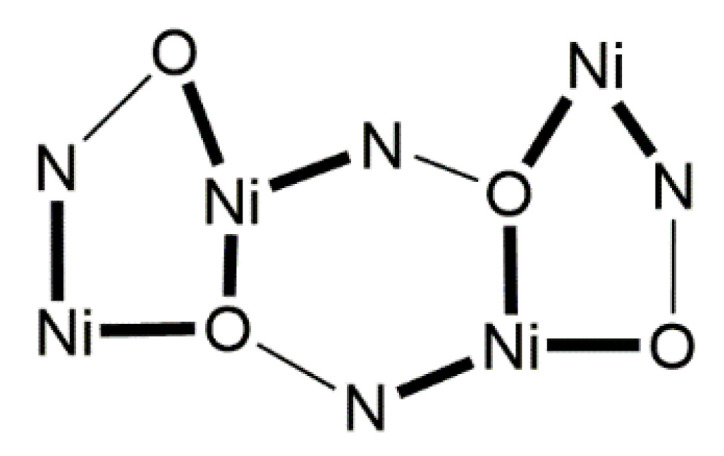
The pseudo 12-MC-4 ring of complex [Ni_4_(SO_4_)_2_(dpkox)_4_(MeOH)_4_] (**79**). The coordination bonds are drawn with bold lines.

**Figure 64 molecules-30-00791-f064:**
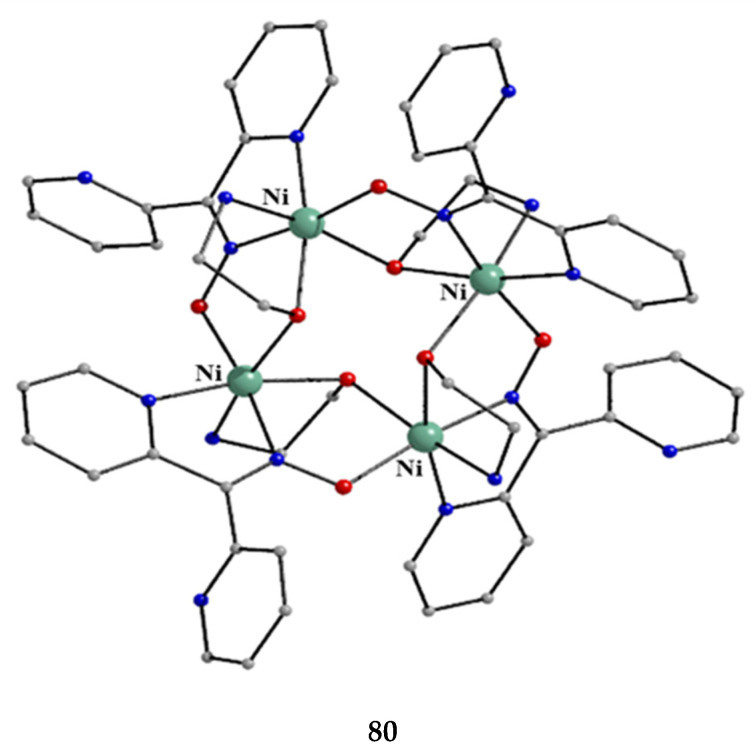
The structure of the cation [Ni_4_(ea)_2_(eaH)_2_(dpkox)_4_]^2+^ that is present in the crystal of its perchlorate salt **80**.

**Figure 67 molecules-30-00791-f067:**
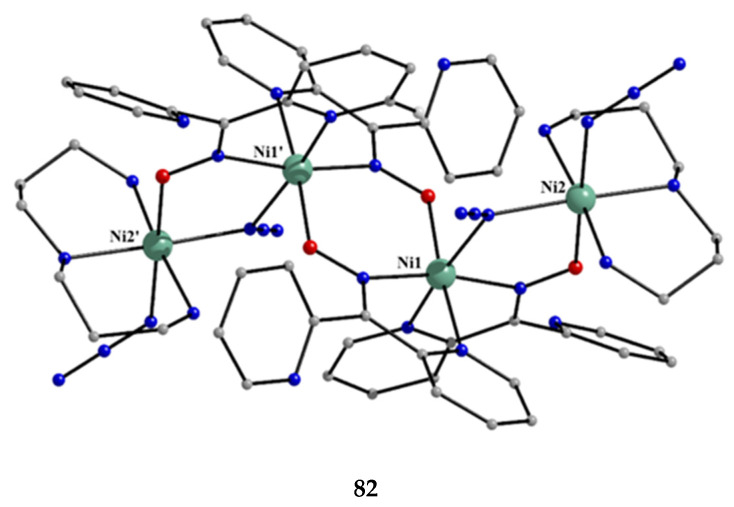
The molecular structure of [Ni_4_(N_3_)_4_(dpt)_2_(dpkox)_4_] (**82**).

**Figure 68 molecules-30-00791-f068:**
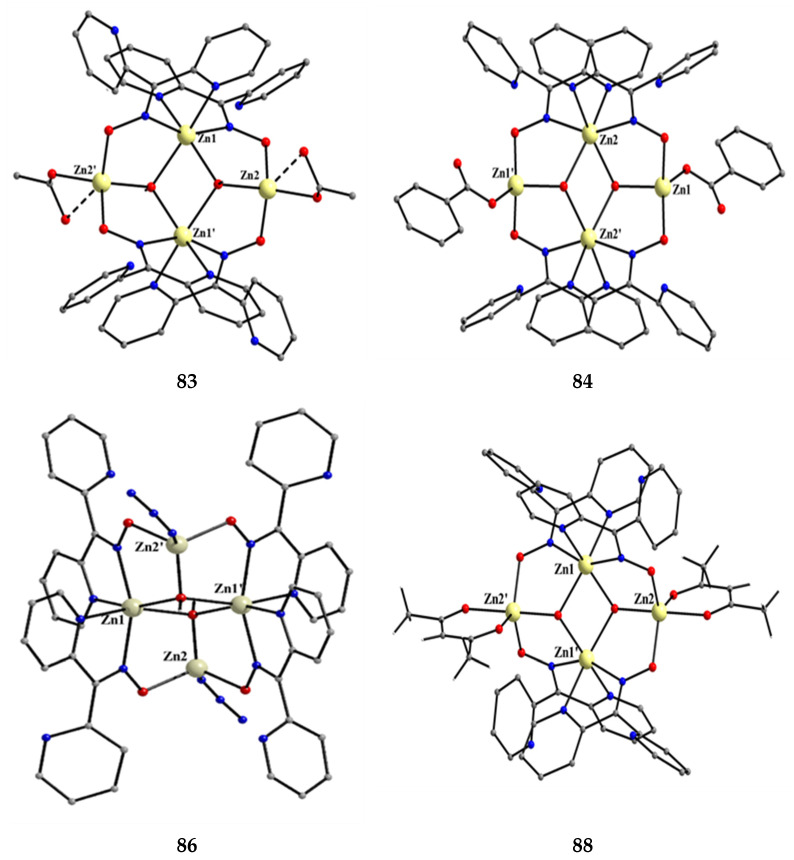
The molecular structures of [Zn_4_(OH)_2_(O_2_CMe)_2_(dpkox)_4_] (**83**), [Zn_4_(OH)_2_(O_2_CPh)_2_(dpkox)_4_] (**84**), [Zn_4_(OH)_2_(N_3_)_2_(dpkox)_4_] (**86**), and [Zn_4_(OH)_2_(acac)_2_(dpkox)_4_] (**88**). The dashed line indicates a weak bonding interaction of one carboxylato O atom to Zn2/Zn2’ (Zn-O = ~2.6 Å).

**Figure 70 molecules-30-00791-f070:**
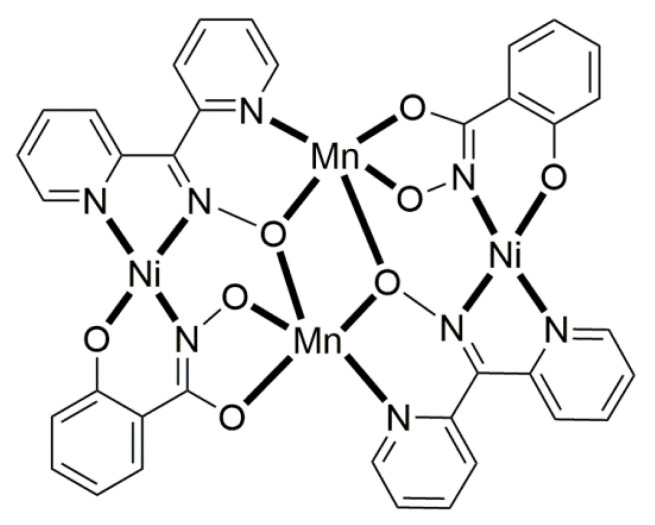
Drawing shown the connectivity pattern and the arrangement of donor atoms around the Mn^III^ and Ni^II^ centers in [Ni_2_Mn^III^_2_(O_2_CMe)_2_(shi)_2_(dpkox)_2_(DMF)_2_] (**90**). The coordination bonds are indicated with bold lines.

**Figure 73 molecules-30-00791-f073:**
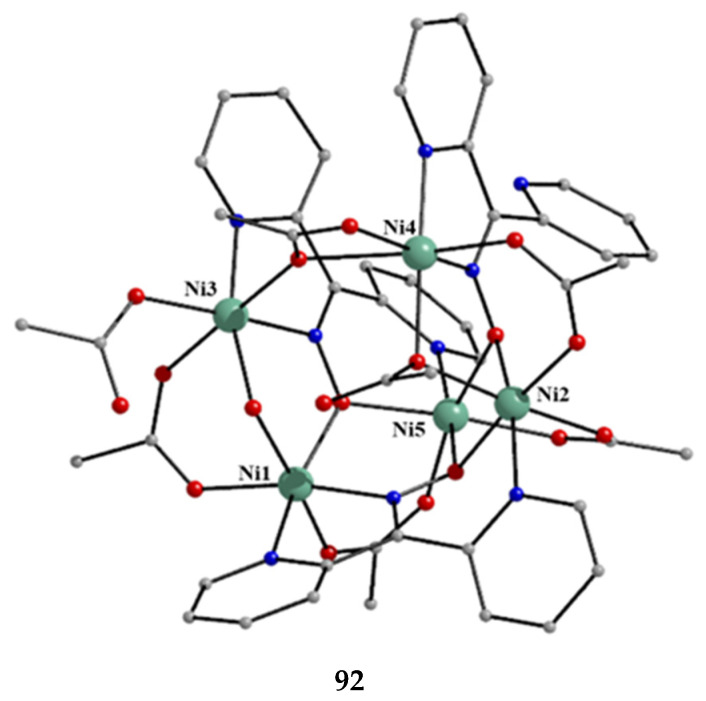
The molecular structure of [Ni_5_(O_2_CMe)_7_(dpkox)_3_(H_2_O)] (**92**).

**Figure 74 molecules-30-00791-f074:**
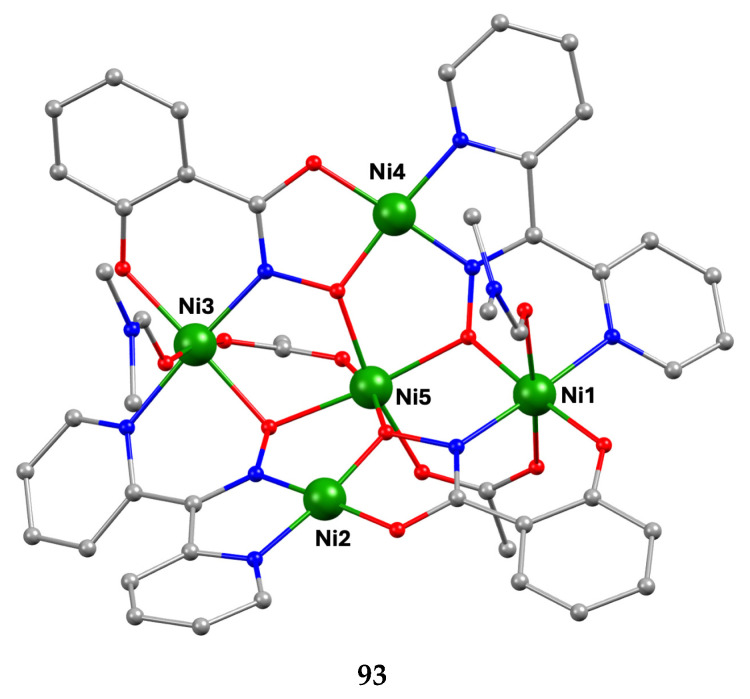
The molecular structure of [Ni_5_(O_2_CMe)_2_(shi)_2_(dpkox)_2_(DMF)_2_] (**93**).

**Figure 75 molecules-30-00791-f075:**
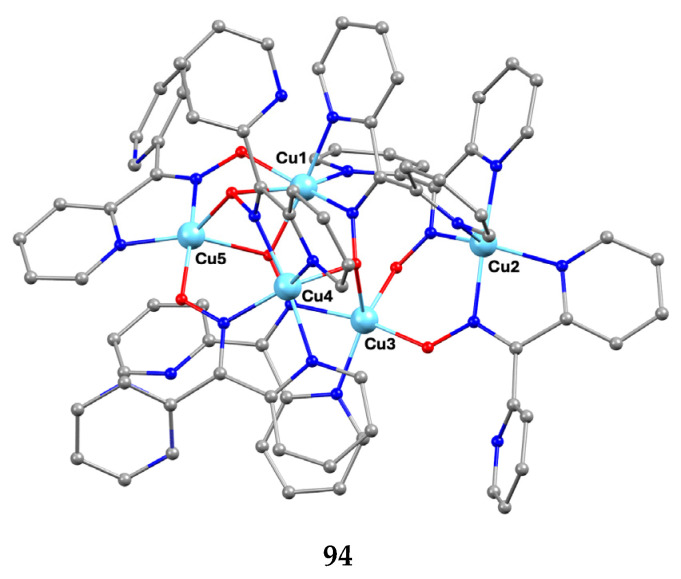
The pentanuclear cation [Cu^II^_5_(dpkox)_7_]^3+^ that is present in the crystal structure of **94**.

**Figure 76 molecules-30-00791-f076:**
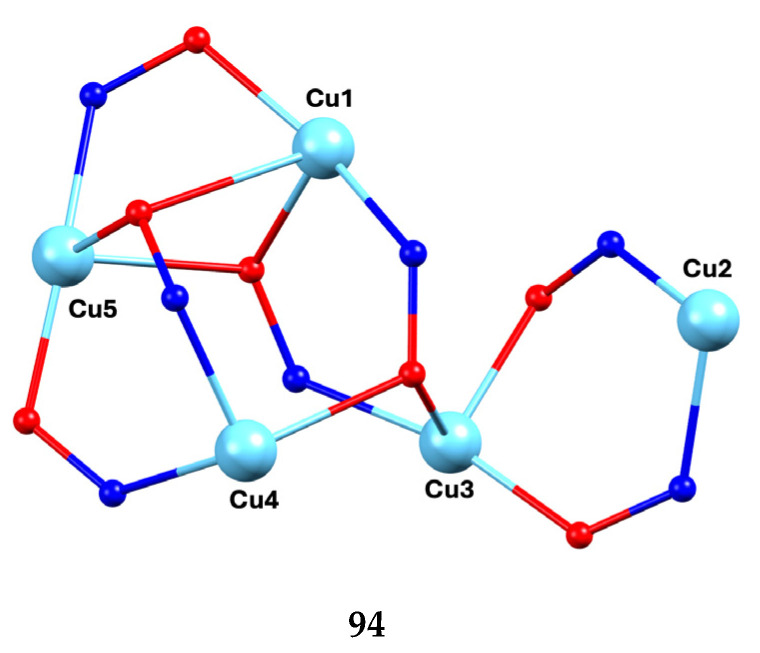
The {Cu^II^_5_(μ_3_-ΝO)_3_(μ_2_-NO)_4_}^3+^ core of the cation of complex **94**; NO represents the oximato group.

**Figure 77 molecules-30-00791-f077:**
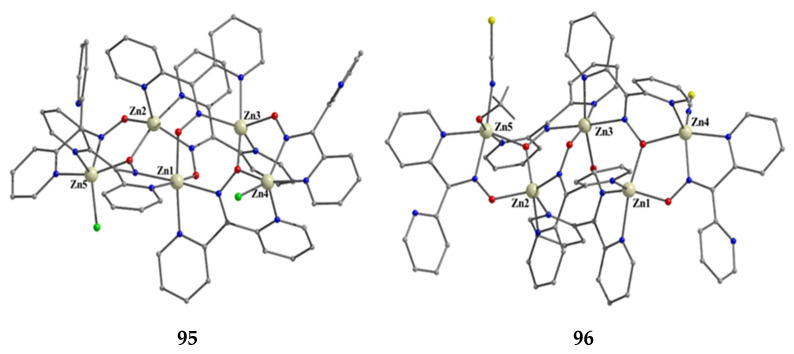
The pentanuclear cations that are present in the crystal structures of the ionic complexes [Zn_5_Cl_2_(dpkox)_6_][ZnCl(NCS)_3_] (**95**) and [Zn_5_(NCS)_2_(dpkox)_6_(MeOH)][Zn(NCS)_4_] (**96**).

**Figure 78 molecules-30-00791-f078:**
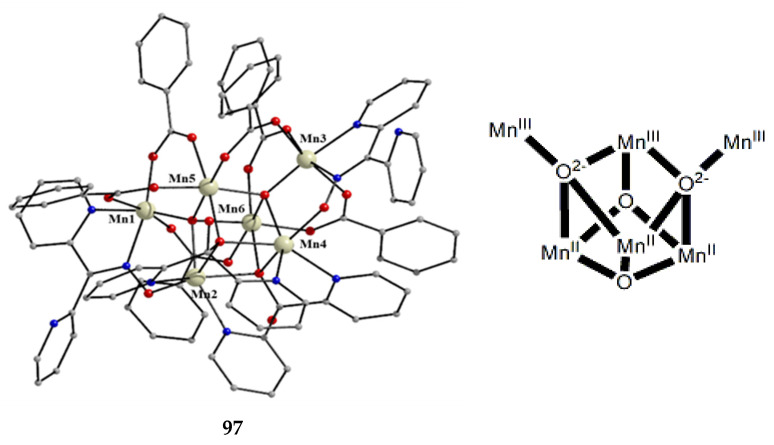
The structure of the cation [Mn^II^_3_Mn^III^_3_O_2_(O_2_CPh)_6_(dpkox)_2_{(py)_2_C(OH)(O)}_2_]^+^ that is present in complex **97** (**left**) and its core (**right**). The coordination bonds are drawn with bold lines. The quadruply bridging oxygens represent the oxido (O^2−^) groups, while the triply bridging oxygens come from the O atoms of the 3.3011 (py)_2_C(OH)(O)^−^ ligands.

**Figure 79 molecules-30-00791-f079:**
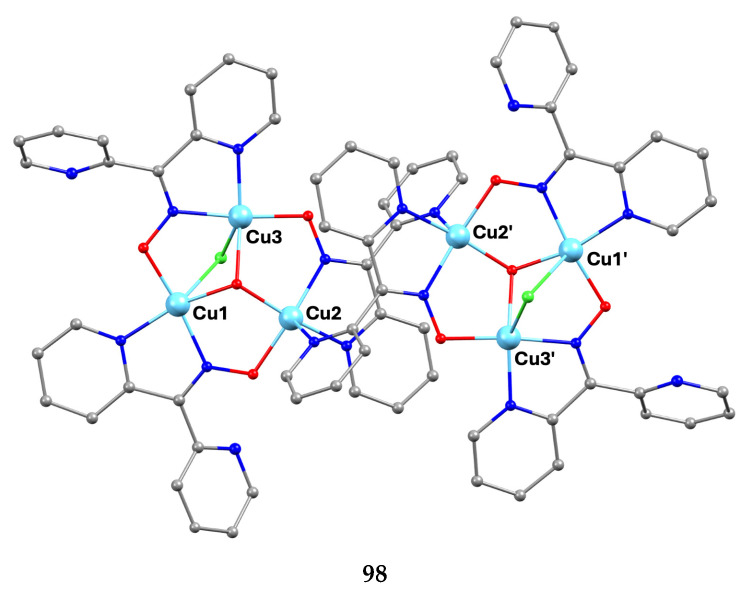
The structure of the cation [Cu^II^_6_(OH)_2_Cl_2_(dpkox)_6_]^2+^ that is present in the crystal of **98**.

**Figure 84 molecules-30-00791-f084:**
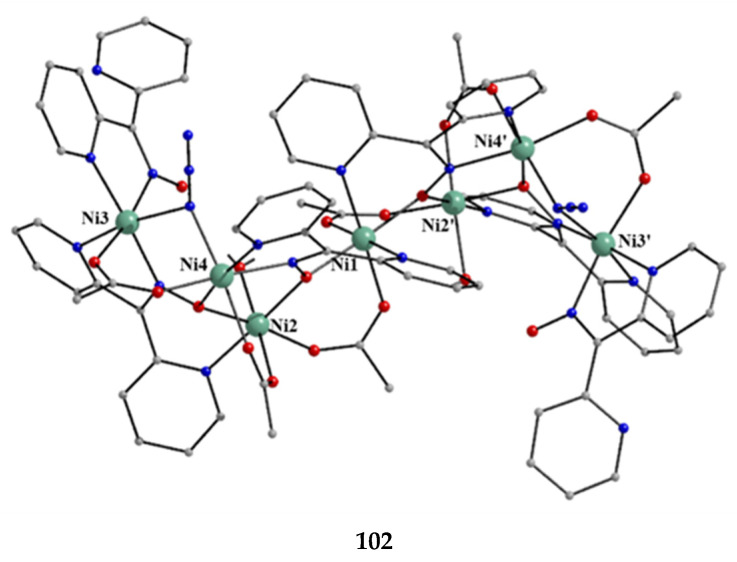
The molecular structure of [Ni_7_(N_3_)_2_(O_2_CMe)_6_(dpkox)_6_(H_2_O)_2_] (**102**).

**Figure 85 molecules-30-00791-f085:**
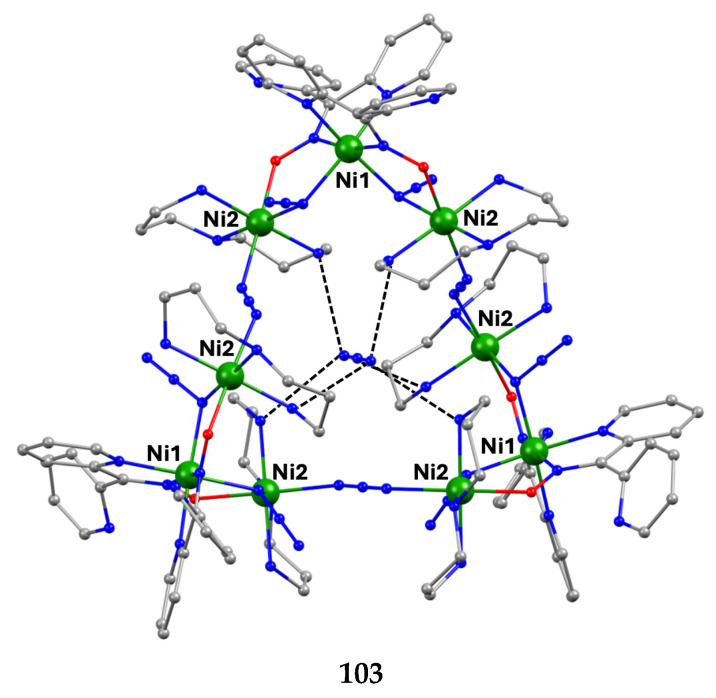
The structure of the cation [Ni_9_(N_3_)_9_(dpkox)_6_(dpt)_6_]^3+^ that is present in cluster **103**; the H-bonded encapsulated N_3_^−^ ion (dashed lines) is also shown. An identical numbering scheme (i.e., without primes) is used for the Ni^II^ atoms generated by symmetry.

**Figure 88 molecules-30-00791-f088:**
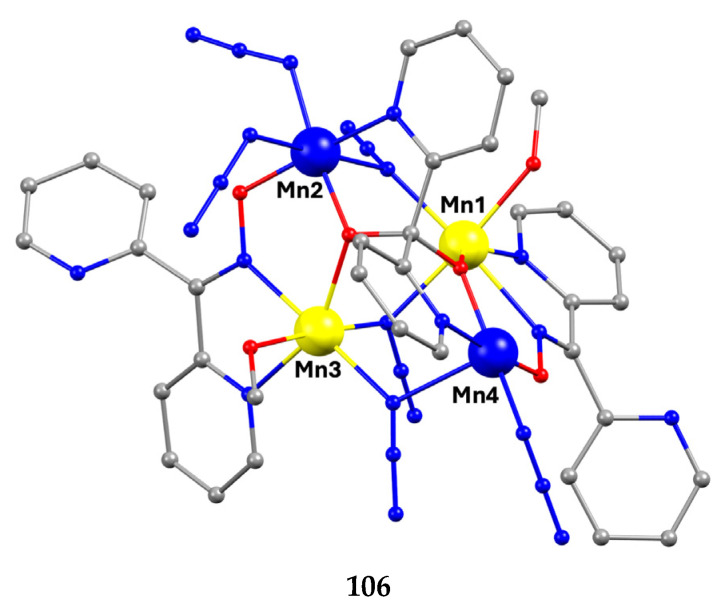
The tetranuclear unit that creates the 1D coordination polymer {[Mn^II^_2_Mn^III^_2_(N_3_)_6_(dpkox)_2_{(py)_2_C(O)_2_}(MeOH)_2_]}_n_ (**106**). Two terminal azido groups on Mn2 and Mn4 are becoming end-to-end in the 1D chain providing the intercluster linkage.

**Figure 89 molecules-30-00791-f089:**
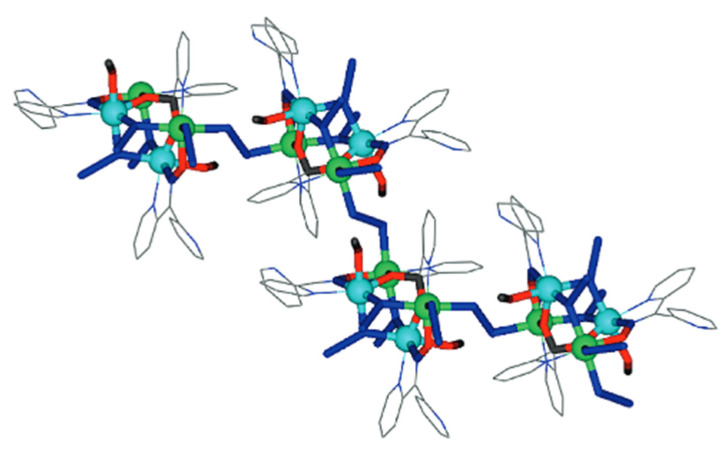
Chain structure of {[Mn^II^_2_Mn^III^_2_(N_3_)_6_(dpkox)_2_{(py)_2_C(O)_2_}(MeOH)_2_]}n (**106**). Adjacent tetranuclear units are linked via an end-to-end (2.11) azido group, which connects two Mn^III^ atoms; these groups are terminal in each tetranuclear unit and are becoming end-to-end to generate the polymer. Reproduced from Ref. [71]. Copyright 2008 American Chemical Society.

**Figure 91 molecules-30-00791-f091:**
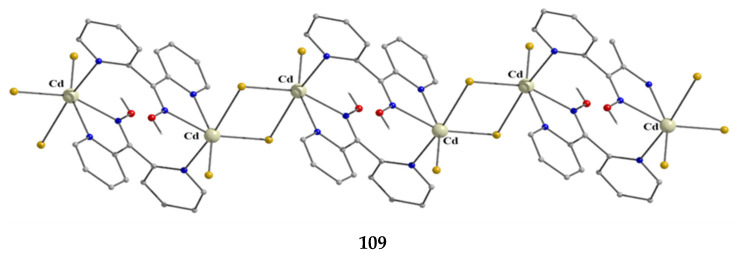
A portion of one zigzag chain that is present in the crystal structure of {[CdBr_2_(dpkox)]}_n_ (**109**).

**Figure 92 molecules-30-00791-f092:**
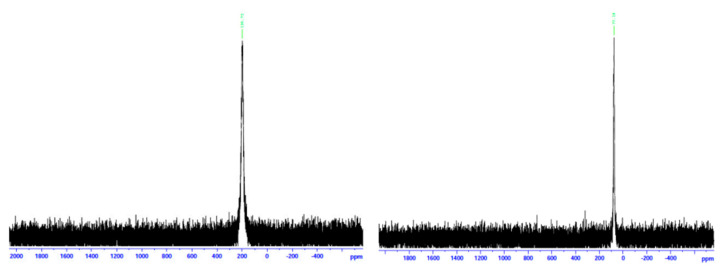
The ^113^Cd NMR spectra of {[CdX_2_(dpkoxH)]}_n_ (X = Cl, **108**; X = Br, **109**) (**left**) and {[CdI_2_(dpkoxH)]}_n_ (**110**) (**right**).

**Figure 93 molecules-30-00791-f093:**
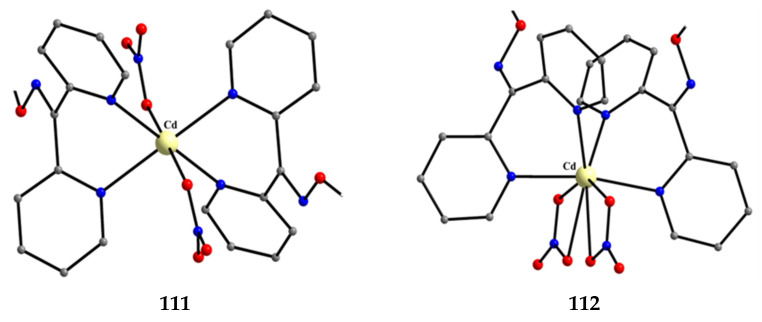
The molecular structures of [Cd(1.100-NO_3_)_2_(dpkoxH)_2_] (**111**) and [Cd(1.110-NO_3_)_2_(dpkoxH)_2_] (**112**).

**Figure 94 molecules-30-00791-f094:**
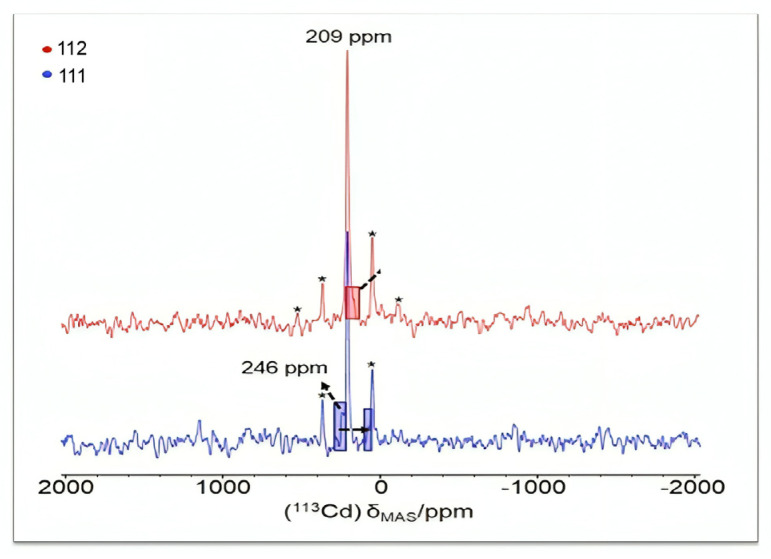
Solid-state MAS ^113^Cd NMR spectra of the samples isolated during the preparations of complexes **111** and **112**.

**Figure 95 molecules-30-00791-f095:**
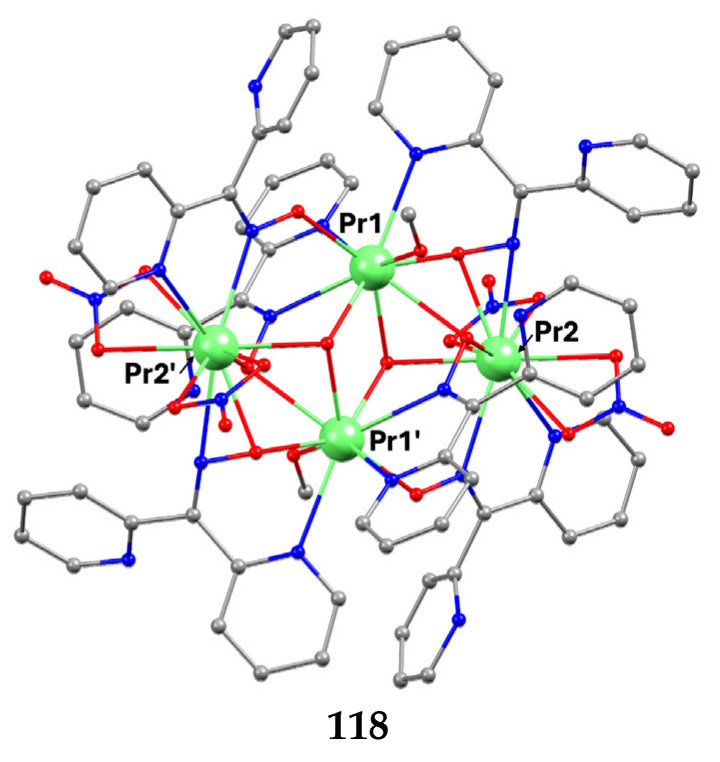
The molecular structure of the centrosymmetric complex [Pr_4_(OH)_2_(NO_3_)_4_(dpkox)_6_(MeOH)_2_] (**118**).

**Figure 96 molecules-30-00791-f096:**
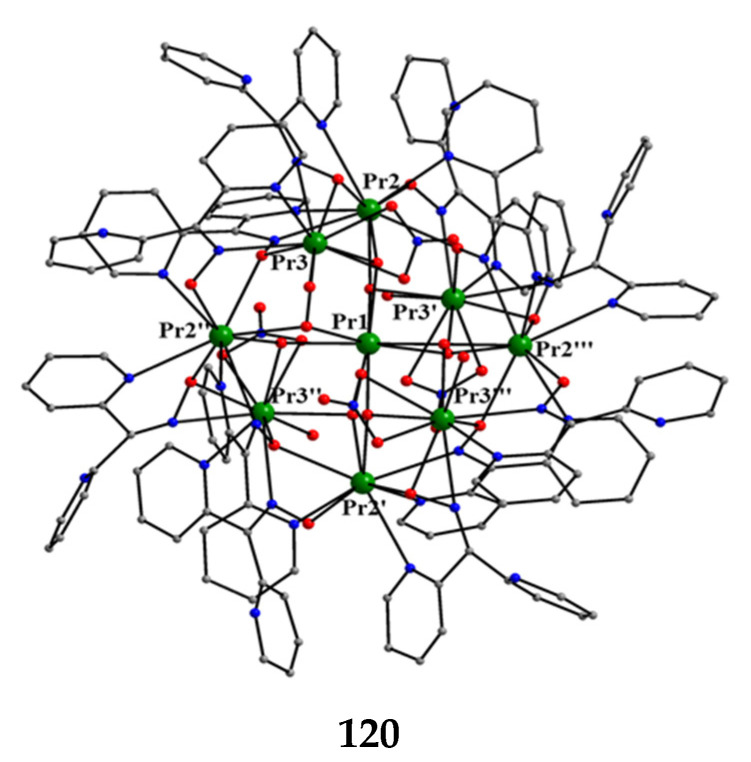
The molecular structure of the mixed-valence coordination cluster [Pr^III^_8_Pr^IV^O_4_(OH)_4_(NO_3_)_4_(dpkox)_12_(H_2_O)_4_] (**120**).

**Figure 97 molecules-30-00791-f097:**
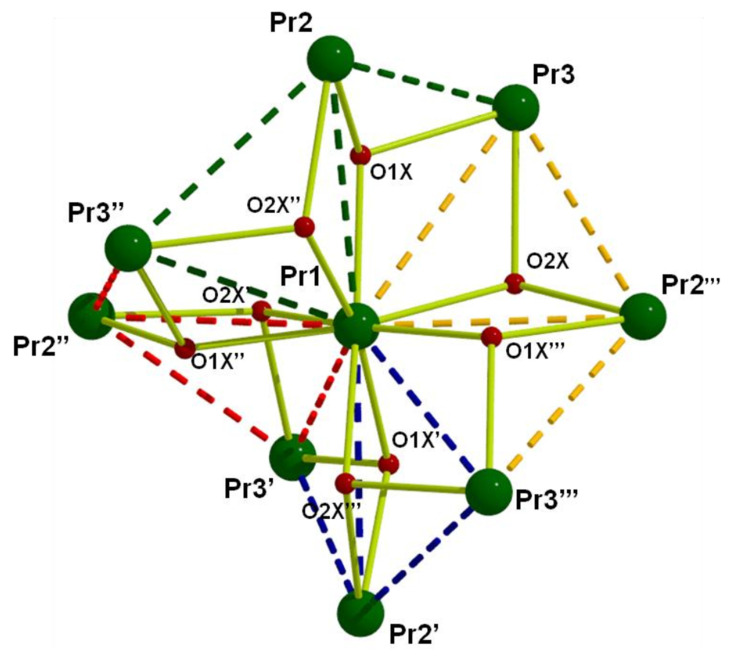
The topology (dashed multicolored lines) and the inorganic {Pr^III^_8_Pr^IV^(μ_3_-O)_4_(μ_3_-OH)_4_}^16+^ core in **120**. O1X (and symmetry equivalents) and OX2 (and symmetry equivalents) are oxido and hydroxido oxygens.

**Figure 102 molecules-30-00791-f102:**
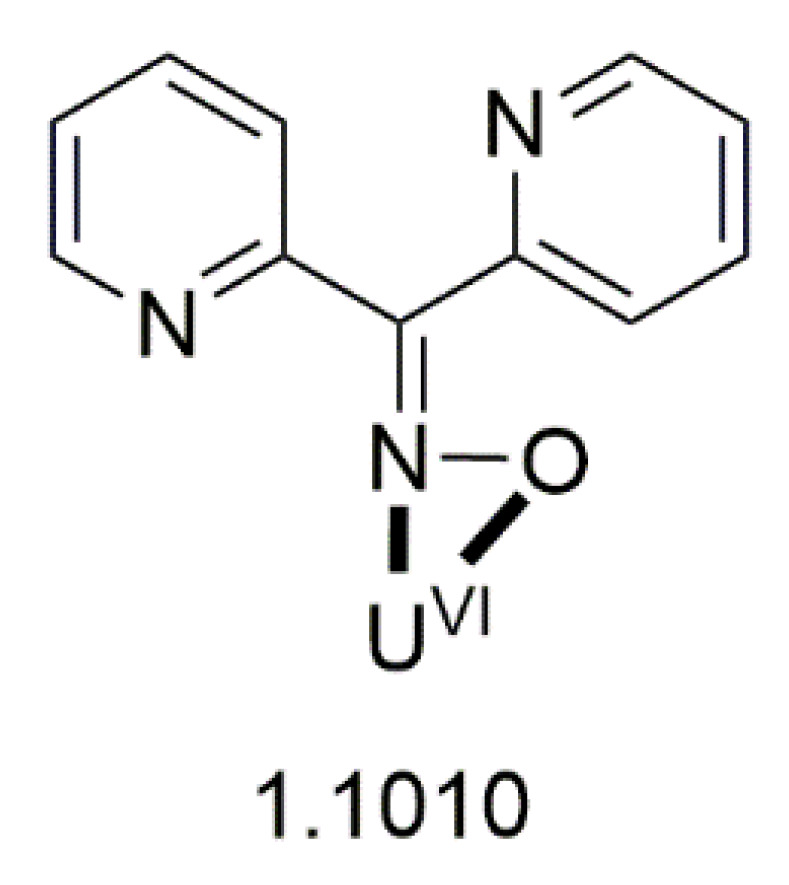
The 1.1010 (Harris notation) coordination mode confirmed in the just prepared and structurally characterized hexagonal bipyramidal complex [U^VI^O_2_(dpkox)_2_(MeOH)_2_]. The coordination bonds are drawn with solid lines.

**Table 1 molecules-30-00791-t001:** Mononuclear metal complexes of dpkoxH and/or dpkox^−^.

Complex ^a,b^	Coordination Mode of dpkoxH/dpkox^− c^	Ref.
[Mn(O_2_CPh)_2_(dpkoxH)_2_] (**1**)	1.0110	[24]
[Ni(mef)_2_(dpkoxH)_2_] (**2**)	1.0110	[25]
[Ni(tolf)_2_(dpkoxH)_2_] (**3**)	1.0110	[26]
[Ni(difl)_2_(dpkoxH)_2_] (**4**)	1.0110	[27]
[Ni(mclf)_2_(dpkoxH)_2_] (**5**)	1.0110	[28]
[Ni(dicl)(diclH)(dpkoxH)_2_](dicl) (**6**)	1.0110	[29]
[Ni(indo)_2_(dpkoxH)_2_] (**7**) ^d^	1.0110	[30]
[Cu^II^Cl(dpkox)(dpkoxH)] (**8**)	1.0110, 1.0110	[31]
[CuCl_2_(dpkoxH)_2_] (**9**) ^d^	1.0101	[32]
[CuBr_2_(dpkoxH)_2_] (**10**) ^d^	1.0101	[32]
[ZnCl_2_(dpkoxH)] (**11**) ^e^	1.0101	[33]
[ZnCl_2_(dpkoxH)] (**12**) ^e,f^	1.0101	[34,35]
[ZnBr_2_(dpkoxH)] (**13**)	1.0101	[34,35]
[Zn(mef)_2_(dpkoxH)_2_] (**14**)	1.0110	[36]
[Zn(tolf)_2_(dpkoxH)_2_] (**15**)	1.0110	[37]
[Zn(difl)_2_(dpkoxH)_2_] (**16**)	1.0110	[38]
[Zn(fluf)_2_(dpkoxH)_2_] (**17**) ^d^	1.0110	[39]
[(Ph)Ru^III^Cl(dpkoxH)](PF_6_) (**18**)	1.0110	[40]
[(Cp*)Rh^III^Cl(dpkoxH)](PF_6_) (**19**)	1.0101	[40]
[Re^I^(CO)_3_Cl(dpkoxH)] (**20**)	1.0101	[41]
[(Cp*)Ir^III^Cl(dpkoxH)](PF_6_) (**21**) ^g^	1.0110	[40]
[(Cp*)Ir^III^Cl(dpkoxH)](PF_6_) (**22**) ^g^	1.0101	[40]
[Au^III^Cl(dpkox)_2_] (**23**)	1.0110	[42]

^a^ Abbreviations of the non-common ancillary ligands: mef = mefenamato(−1), tolf = tolfenamato(−1), difl = the anion of diflunisal, mclf = meclofenamato(−1), dicl = the anion of diclofenac, indo = the anion of indomethacin, fluf = flufenamato(−1), Cp* = pentamethylcyclopentadienate(−1). ^b^ The structural formulae of the non-common ancillary carboxylate ligands used for the preparation of some Ni(II) and Zn(II) complexes are illustrated in Figure 15. ^c^ Using the Harris notation [2]. ^d^ The structures of these complexes have been proposed based on spectroscopic data. ^e^ These complexes are polymorphs. ^f^ The structure of this compound has been reported twice. ^g^ Isomers with different coordination modes of dpkoxH, see text. Lattice solvent molecules have not been incorporated into the formulae of the compounds.

**Table 5 molecules-30-00791-t005:** Homotetranuclear (homotetrametallic) and heterotetranuclear (heterotetrametallic) complexes of dpkox^−^.

Complex	Coordination Mode of dpkox^− j^	Ref.
[Mn^II,II,II^_3_Mn^IV^O(3,4-D)_4_(dpkox)_4_] (**64**) ^a^	2.1110	[69]
[Mn^II,II,II^_3_Mn^IV^O(2,4,5-T)_4_(dpkox)_4_] (**65**) ^b^	2.1110	[70]
[Mn^II,II,II^_3_Mn^IV^O(2,3-D)_4_(dpkox)_4_] (**66**) ^c^	2.1110	[71]
[Mn^II,II^_2_Mn^III,III^_2_Br_2_(O_2_CPh)_2_(dpkox)_2_{(py)_2_C(O)_2_}_2_] (**67**) ^d^	2.1110	[24,72]
[Mn^II,II^_2_Mn^III,III^_2_Br_2_(O_2_CMe)_2_(dpkox)_2_{(py)_2_C(O)_2_}] (**68**) ^d^	2.1110	[24,72]
[Mn^II,II^_2_Mn^III,III^_2_Cl_2_(O_2_CPh)_2_(dpkox)_2_{(py)_2_C(O)_2_}_2_] (**69**) ^d^	2.1110	[24]
[Mn^II,II^_2_Mn^III,III^_2_(NO_3_)_2_(O_2_CMe)_2_(dpkox)_2_{(py)_2_C(O_2_)}_2_] (**70**) ^d^	2.1110	[24]
[Fe^III^_4_O_2_Cl_2_(O_2_CMe)_2_(dpkox)_4_] (**71**) ^e^	2.1110	[73]
[Fe^III^_4_O_2_Cl_2_(O_2_CMe)_2_(dpkox)_4_] (**72**) ^e^	2.1110	[73]
[Fe^III^_4_O_2_(N_3_)_2_(O_2_CMe)_2_(dpkox)_4_] (**73**)	1.0110	[73]
[Co^II,II^_2_Co^III,III^_2_(OH)_2_(O_2_CMe)_2_(dpkox)_4_(MeOH)_2_](ClO_4_)_2_ (**74**)	2.1110, 2.1111	[74]
[Co^II,II^_2_Co^III,III^_2_(OMe)_2_(O_2_CMe)_2_(dpkox)_4_(EtOH)_2_](PF_6_)_2_ (**75**)	2.1110, 2.1111	[74]
[Co^III^_4_(OH)_2_(O_2_CMe)_4_(dpkox)_4_](PF_6_)_2_ (**76**)	2.1110, 2.1111	[75]
[Ni_4_(dpkox)_6_(MeOH)_2_](OH)(ClO_4_) (**77**)	2.1110, 3.2111	[76]
[Ni_4_(O_2_CMe)_2_(dpkox)_4_](SCN)(OH) (**78**)	3.2111	[77]
[Ni_4_(SO_4_)_2_(dpkox)_4_(MeOH)_4_] (**79**)	2.1110, 3.2111	[78]
[Ni_4_(ea)_2_(eaH)_2_(dpkox)_4_](ClO_4_)_2_ (**80**) ^f^	2.1110	[79]
[Ni_4_(SCN)_2_(shiH)_2_(dpkox)_2_(DMF)(MeOH)] (**81**) ^g^	2.1111	[80]
[Ni_4_(N_3_)_4_(dpt)_2_(dpkox)_4_] (**82**) ^h^	2.0110	[60]
[Zn_4_(OH)_2_(O_2_CMe)_2_(dpkox)_4_] (**83**)	2.1110	[81]
[Zn_4_(OH)_2_(O_2_CPh)_2_(dpkox)_4_] (**84**)	2.1110	[82]
[Zn_4_(OH)_2_Cl_2_(dpkox)_4_] (**85**)	2.1110	[33]
[Zn_4_(OH)_2_(N_3_)_2_(dpkox)_4_] (**86**)	2.1110	[82]
[Zn_4_(OH)_2_(NCO)_2_(dpkox)_4_] (**87**)	2.1110	[82]
[Zn_4_(OH)_2_(acac)_2_(dpkox)_4_] (**88**) ^i^	2.1110	[82]
[Zn_4_(OH)_2_(dpkox)_6_] (**89**)	1.0110, 2.1110	[83]
[Ni_2_Mn^III^_2_(O_2_CMe)_2_(shi)_2_(dpkox)_2_(DMF)_2_] (**90**) ^k^	3.2_1_1_1_1_2_1_2_ ^b^	[80]

^a^ 3,4-D = 3,4-dichlorophenoxyacetate(−1). ^b^ 2,4,5-T = 2,4,5-trichlorophenoxyacetate(−1). ^c^ 2,3-D = 2,3-dichlorophenoxyacetate(−1). ^d^ (py)_2_C(O)_2_ is the doubly deprotonated *gem*-diol derivative of di-2-pyridyl ketone [(py)_2_CO]. ^e^ These complexes are pseudopolymorphs. ^f^ eaH is ethanolamine and ea is its anion. ^g^ shiH and shi are the doubly and triply deprotonated, respectively, forms of salicylhydroxamic acid (shiH_3_, Figure 12/**K**). ^h^ dpt = dipropylenetriamine. ^i^ acac is the acetylacetonato(−1) ligand. The structural formulae of most of these ligands are illustrated in Figure 15. ^j^ Using the Harris notation [2]. ^k^ The subscript 1 refers to Mn^III^ and the subscript 2 to Ni^II^. Lattice solvent molecules have not been incorporated in the formulae of the compounds.

**Table 6 molecules-30-00791-t006:** Pentanuclear and hexanuclear complexes of dpkoxH and dpkox^−^.

Complex	Coordination Mode of dpkoxH/dpkox^− c^	Ref.
[Ni_5_(dpkox)_5_(H_2_O)_7_] (**91**)	2.1110, 3.2111	[84]
[Ni_5_(O_2_CMe)_7_(dpkox)_3_(H_2_O)] (**92**)	3.2111, 3.2110	[77]
[Ni_5_(O_2_CMe)_2_(shi)_2_(dpkox)_2_(DMF)_2_] (**93**)	3.2111	[80]
[Cu^II^_5_(dpkox)_7_](ClO_4_)_3_ (**94**)	2.1110, 3.1111, 3.2110, 4.2111	[48]
[Zn_5_Cl_2_(dpkox)_6_][ZnCl(NCS)_3_] (**95**)	2.1110, 3.2111, 3.111	[82]
[Zn_5_(NCS)_2_(dpkox)_6_(MeOH)][Zn(NCS)_4_] (**96**)	2.1110, 3.2111, 3.1111	[82]
[Mn^II^_3_Mn^III^_3_O_2_(O_2_CPh)_6_(dpkox)_2_{(py)_2_C(OH)(O)}_2_](ClO_4_) (**97**) ^a^	2.1110	[85]
[Cu^II^_6_(OH)_2_Cl_2_(dpkox)_6_](BPh_4_)_2_ (**98**)	2.1110, 3.1111	[48]
[Cu^II^_6_(ClO_4_)(dpkox)_6_(MeCN)_6_][Cu^II^_6_(ClO_4_)_3_(dpkox)_6_(MeCN)_4_] (ClO_4_)_8_ (**99**)	2.1111	[48,86]
[Zn_6_(OH)_2_(dpkox)_4_(fluf)_6_] (**100**) ^b^	3.2111, 3.2110	[87]

^a^ (py)_2_C(OH)(O) is the singly deprotonated *gem*-diol derivative of di-2-pyridyl ketone [(py)_2_CO]. ^b^ fluf = flufenamate(−1). The structural formulae of fluf^−^ and (py)_2_C(OH)(O)^−^ are illustrated in Figure 15. ^c^ Using the Harris notation [2]. Lattice solvent molecules have not been incorporated in the formulae of the compounds.

**Table 8 molecules-30-00791-t008:** Coordination polymers of dpkoxH and dpkox^−^.

Complex	Coordination Mode of dpkoxH/dpkox^− b^	Ref.
{[Mn^II^_2_Mn^III^_2_(N_3_)_6_(dpkox)_2_{(py)_2_C(O)_2_}(MeOH)_2_]}_n_ (**106**) ^a^	2.1110	[71]
{[Cu^I^(SCN)(dpkoxH)]}_n_ (**107**)	1.0110	[51]
{[CdCl_2_(dpkoxH)]}_n_ (**108**)	2.0111	[91]
{[CdBr_2_(dpkoxH)]}_n_ (**109**)	2.0111	[91]
{[CdI_2_(dpkoxH)]}_n_ (**110**)	2.0111	[91]

^a^ (py)_2_C(O)_2_ is the doubly deprotonated *gem*-diol derivative of di-2-pyridyl ketone; see Figure 15. ^b^ Using the Harris notation [2]. Lattice solvent molecules have not been incorporated in the formulae of the compounds.

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
