# Peer review of "Towards Completion of the “Periodic Table” of Di-2-Pyridyl Ketoxime"

_molecules, 2025, doi:10.3390/molecules30040791_

Round 1
Reviewer 1 Report
Comments and Suggestions for Authors
In this review, the structures and properties of different metal complexes based on 2-pyridyl oxime ligands were reported. The studies should be of interest for the readers in coordination chemistry and biochemistry. It can be accepted in molecules after some issues are addressed:
1. The author mentioned that in mononuclear compounds the ligands are either in neutral or deprotonated versions. So, what are the effects of these two different coordination modes on the structures or properties of the corresponding complexes?
2. Are there some rules when metals center with different ionic radii react with the ligands? How to synthesize the metal complexes with different numbers of metal nuclear, or how to regulate the ratio of metals to ligands in metal complexes?
3. How about these complexes be used as catalysts in organic chemistry? The applications can be discussed.
4. Some pyridine-based NO-Ru complexes, which are similar to compound 43 in this review, should be appropriately integrated into this text (Appl Organometal Chem. 2020;34:e5292; Eur. J. Inorg. Chem. 2024, 27, e202300536).
Author Response
«In this review, the structures and properties of different metal complexes based on 2-pyridyl oxime ligands were reported. The studies should be of interest for the readers in coordination chemistry and biochemistry. It can be accepted in molecules after some issues are addressed»
We thank the reviewer for her/his time to study our ms and give us valuable revision points/suggestions. We are also grateful for her/his positive comments and proposal for acceptance.
We are glad because we were able to address all the points he/she raised. For convenience and easy communication, we also give her/his comments before our detailed relevant answers. Please note that the numbering schemes of compounds and references have been changed as a consequence of the revision points raised by Reviewer 3.
- “ The author mentioned that in mononuclear compounds the ligands are either in neutral or deprotonated versions. So, what are the effects of these two different coordination modes on the structures or properties of the corresponding complexes?”
The comment is correct. There is no specific effect of the two different forms of the ligand on the structural features of the complexes. In the mononuclear complexes the deprotonated ligand behaves always with the 1.0110 manner, while the neutral molecule behaves either as 1.0110 or 1.0101 ligand. We have added a relevant text (“There is no specific effect of…can act as bridging ligands”) in part 5.2 of the revised version of the ms.
- “Are there some rules when metals center with different ionic radii react with the ligands? How to synthesize the metal complexes with different numbers of metal nuclear, or how to regulate the ratio of metals to ligands in metal complexes?”
The remark is logical. It is tentatively proposed that large sizes of the metal centers decrease the coordination flexibility of the deprotonated ligand. We discuss this point in detail in section 7 where a text (“There appear to be…higher nuclearity species”) has been added in the first paragraph.
- “How about these complexes be used as catalysts in organic chemistry? The applications can be discussed.”
This question is of vital importance. Only a tetranuclear homovalent Co(III) complex (compound 76 with the new numbering scheme) shows catalytic activity in organic chemistry reactions. We have added a new text (“In the former case…with 100% O2 as oxidant”) in part 5.5 of the revised ms.
- “Some pyridine-based NO-Ru complexes, which are similar to compound 43 in this review,should be appropriately integrated into this text (Appl Organometal Chem. 2020;34:e5292; J. Inorg. Chem. 2024, 27, e202300536).”
This suggestion is valuable. We have added the two new references [41] and [62] and relevant texts in parts 5.2 [“It should be mentioned…for water oxidation (WOCs)”.] and 5.4 (“Complex 50 is structurally similar…as oxidant”) of the revised ms.
Overall, the implementation of the above mentioned revision suggestions have remarkably improved the quality and readability of the ms.
Reviewer 2 Report
Comments and Suggestions for Authors
The authors have put together a very nice review of metal-oxime chemistry. This work will serve as an excellent resource for researchers interested in the field. Although there is a fair amount of self-citation, it seems to be warranted given the authors expertise in the field.
My only major point that should be addressed is stylistic in nature; the ChemDraw figures throughout the manuscript have inconsistent bond angles and lengths. This detracts from the manuscript and, at at times, makes interpretation difficult. I would encourage the authors to use “line-angle” structures rather than explicitly showing CH2 or CH groups (see Figure 4 as an example). The use of ACS 1996 ChemDraw style will help with consistency. This will make structures far easier to interpret.
Although this may seem pedantic, in a review article this is critical.
Beyond this point, the manuscript is suitable for publication in Molecules.
Author Response
“The authors have put together a very nice review of metal-oxime chemistry. This work will serve as an excellent resource for researchers interested in the field. Although there is a fair amount of self-citation, it seems to be warranted given the authors expertise in the field.
My only major point that should be addressed is stylistic in nature; the ChemDraw figures throughout the manuscript have inconsistent bond angles and lengths. This detracts from the manuscript and, at at times, makes interpretation difficult. I would encourage the authors to use “line-angle” structures rather than explicitly showing CH2 or CH groups (see Figure 4 as an example). The use of ACS 1996 ChemDraw style will help with consistency. This will make structures far easier to interpret.
Although this may seem pedantic, in a review article this is critical.
Beyond this point, the manuscript is suitable for publication in Molecules.”
We thank the reviewer for her/his time to study our ms and give us a valuable suggestion. We are also grateful for her/his positive comments and proposal for acceptance. The comment is absolutely correct. We have followed the suggestion and used the “line-angle” structures in all the ChemDraw figures; therefore, CH2 or CH groups do not appear. We used the ACS 1996 Chem Draw style for consistency. Now, the stylistic and aesthetic appearance of the ms have been improved a lot. Please note that the numbering schemes of compounds and references have been changed as a consequence of the revision points raised by Reviewer 3.
Reviewer 3 Report
Comments and Suggestions for Authors
The manuscript by Lada, Konidaris, and Perlepes provides a comprehensive review of the coordination complexes formed by di-2-pyridyl-2-ketoxime with metal cations from across the periodic table. The complexes reviewed are formed by di-2-pyridyl-2-ketoxime, primarily with d-block transition metals, and a few examples of lanthanide elements, such as praseodymium. Mononuclear complexes include nickel, copper, zinc, ruthenium, rhodium, rhenium, cobalt, gold, and iridium. Dinuclear complexes are composed of chromium, manganese, copper, ruthenium, silver, and mercury. Trinuclear complexes consist of chromium, manganese, iron, nickel, copper, ruthenium, and osmium. Tetranuclear complexes involve manganese, iron, cobalt, nickel, and zinc. Pentanuclear complexes are based on nickel, copper, manganese, and zinc. Hexanuclear complexes contain manganese, copper, and zinc, while heptanuclear complexes are nickel-based. Enneanuclear and decanuclear complexes are also based on nickel. Coordination polymers include manganese, copper, and cadmium. Unpublished results focus on cadmium, mercury, praseodymium, and gold. The binding mode has been thoroughly analyzed with appropriate crystal data, and balanced equations have also been provided for the synthetic preparations.
The authors have provided a thorough and comprehensive summary of the literature, including appropriate details and ChemDraw illustrations. While the manuscript is well-written, the presentation of data could be enhanced. I recommend its publication in Molecules pending minor revisions. Below are several suggestions for the authors’ consideration.
-I would suggest rearranging these complexes based on the elements as they appear in the standard periodic table, in alignment with the author's title, "Towards Completion of the 'Periodic Table' of Di-2-pyridyl-2-ketoxime." I have attempted to categorize the complexes and please double check that no elements are overlooked.
Chromium (Dinuclear, Trinuclear)
Manganese (Dinuclear, Trinuclear, Tetranuclear, Pentanuclear, Hexanuclear, Coordination polymer)
Iron (Trinuclear, Tetranuclear)
Cobalt (Mononuclear, Tetranuclear)
Nickel (Mononuclear, Pentanuclear, Decanuclear)
Copper (Mononuclear, Dinuclear, Trinuclear, Pentanuclear, Hexanuclear, Coordination polymer)
Zinc (Mononuclear, Tetranuclear, Pentanuclear, Hexanuclear)
Ruthenium (Mononuclear, Dinuclear, Trinuclear)
Rhodium (Mononuclear)
Silver (Dinuclear)
Cadmium (Mononuclear, Tetranuclear, Coordination polymer)
Rhenium (Mononuclear)
Osmium (Trinuclear)
Iridium (Mononuclear)
Gold (Mononuclear, Pentanuclear)
Mercury (Dinuclear, Tetranuclear)
Praseodymium (Mononuclear)
-Additionally, the sequence of the narrative could be more coherent. When discussing a particular element, it would be clearer to complete the series of all complexes associated with that element before moving on to the next one.
-Additionally, based on my independent literature search using SciFinder, I noticed that these references are not cited in the manuscript. I recommend reviewing them and citing them as appropriate.
DOI: 10.1080/00958972.2019.1590708
DOI: 10.1016/j.poly.2017.09.008
DOI: 10.1016/j.molstruc.2017.10.087
DOI: 10.1016/j.ejmech.2013.12.019
DOI: 10.1016/j.jinorgbio.2016.04.023
DOI: 10.1155/S1565363303000074
Patent: China, CN114874269 A 2022-08-09
Anticancer Research (2000), 20(6B), 4435-4439
-In general, these balanced equations are unrealistic and should be omitted, as the reactions do not yield the sole products discussed in the paper. The authors should exercise caution in the messages conveyed through the inclusion of balanced equations.
Abstract: Please remove all the ‘‘fudge words’’: as such ‘‘aesthetically’’ ‘‘beautiful’’ etc. No need to exaggerate; just maintain a meaningful scientific context.
Abstract: Please list lanthanoids elements too.
Abstract: This sentence ‘‘Our efforts to complete the “periodic table” of di-2-pyridyl ketoxime are continued’’ should be in Conclusions not in abstract.
Page 1, Line 43: The oxime group should be presented as a formula, not as a ChemDraw diagram, in the text, using the notation as R2C=NOH (R = alkyl or aryl).
Line 100-101: “Since the descriptions of the structures and the spectroscopic/physical properties will be short, we assume that the readers of this review have a good background in coordination chemistry.” Please remove this sentence, as this is a review intended for readers specialized in coordination chemistry in this area. There is no need to state something so obvious, which only belabors the context.
Page 3, Line ‘‘some structures will be illustrated in a schematic way (ChemDraw)’’ Why future tense? you could use the present tense: "Some structures are illustrated schematically (ChemDraw)."
Same as here: ‘‘The ligation modes will be also represented in a schematic way (ChemDraw)’’. Please correct.
Page 3, Line 113: ‘‘Sections 2-4 are referred to as the “hors d’oeuvre” of the article.’’ I am unsure what “hors d’oeuvre” means in this context. Since it is non-English and potentially confusing, please remove it.
Page 3, Line 114: ‘‘Sections 5 and 6 contain the “main menu” of this scientific meal.’’ Please remove all these metaphor and analogy. Just keep plain and simple science context.
Page 4: ‘‘The Oxime Group: A “Passepartout” in Chemistry’’ – Please simplify the title by keeping it in plain English and removing non-English words. Such as: ‘‘The Oxime Group: A Versatile Tool in Chemistry’’.
Page 4, Line 149: ‘‘..derivatives of oximes are central “players” in modern agriculture as agents’’. Please remove any informal English.
Page 23, Line 531: ‘‘After many “try and see” synthetic exercises..’’ Please remove any informal English.
Page 27, Figure 29, legend: Please use the formula instead of a ChemDraw image.
Page 48, Line 1185, Please use the formula for the oxime group.
Page 57, Line 1469, ‘‘..due to the ionic [M-ClO4-4H2O]+ ionic species’’. Correct sentence.
Page 76, Line 2015: "‘…ligands participate with five(!) different ligation modes.’ Why is the exclamation symbol used?"
Author Response
“The manuscript by Lada, Konidaris, and Perlepes provides a comprehensive review of the coordination complexes formed by di-2-pyridyl-2-ketoxime with metal cations from across the periodic table. The complexes reviewed are formed by di-2-pyridyl-2-ketoxime, primarily with d-block transition metals, and a few examples of lanthanide elements, such as praseodymium. Mononuclear complexes include nickel, copper, zinc, ruthenium, rhodium, rhenium, cobalt, gold, and iridium. Dinuclear complexes are composed of chromium, manganese, copper, ruthenium, silver, and mercury. Trinuclear complexes consist of chromium, manganese, iron, nickel, copper, ruthenium, and osmium. Tetranuclear complexes involve manganese, iron, cobalt, nickel, and zinc. Pentanuclear complexes are based on nickel, copper, manganese, and zinc. Hexanuclear complexes contain manganese, copper, and zinc, while heptanuclear complexes are nickel-based. Enneanuclear and decanuclear complexes are also based on nickel. Coordination polymers include manganese, copper, and cadmium. Unpublished results focus on cadmium, mercury, praseodymium, and gold. The binding mode has been thoroughly analyzed with appropriate crystal data, and balanced equations have also been provided for the synthetic preparations.
The authors have provided a thorough and comprehensive summary of the literature, including appropriate details and ChemDraw illustrations. While the manuscript is well-written, the presentation of data could be enhanced. I recommend its publication in Molecules pending minor revisions. Below are several suggestions for the authors’ consideration.”
We thank the reviewer for her/his time to study our ms and give us valuable revision points/suggestions. We are also grateful for her/his positive comments and proposal for acceptance after minor revisions.
At the outset, we wish to emphasize that all the points she/he raised are correct and we agree with them. However, we were able to address most (but NOT all) revision points. For convenience and easy communication, we also give her/his comments before our detailed relevant answers. Please note that the numbering schemes of compounds and references have been changed as a consequence of the revision comments by this reviewer.
“-I would suggest rearranging these complexes based on the elements as they appear in the standard periodic table, in alignment with the author's title, "Towards Completion of the 'Periodic Table' of Di-2-pyridyl-2-ketoxime." I have attempted to categorize the complexes and please double check that no elements are overlooked.
Chromium (Dinuclear, Trinuclear)
Manganese (Dinuclear, Trinuclear, Tetranuclear, Pentanuclear, Hexanuclear, Coordination polymer)
Iron (Trinuclear, Tetranuclear)
Cobalt (Mononuclear, Tetranuclear)
Nickel (Mononuclear, Pentanuclear, Decanuclear)
Copper (Mononuclear, Dinuclear, Trinuclear, Pentanuclear, Hexanuclear, Coordination polymer)
Zinc (Mononuclear, Tetranuclear, Pentanuclear, Hexanuclear)
Ruthenium (Mononuclear, Dinuclear, Trinuclear)
Rhodium (Mononuclear)
Silver (Dinuclear)
Cadmium (Mononuclear, Tetranuclear, Coordination polymer)
Rhenium (Mononuclear)
Osmium (Trinuclear)
Iridium (Mononuclear)
Gold (Mononuclear, Pentanuclear)
Mercury (Dinuclear, Tetranuclear)
Praseodymium (Mononuclear)
-Additionally, the sequence of the narrative could be more coherent. When discussing a particular element, it would be clearer to complete the series of all complexes associated with that element before moving on to the next one.”
The suggestion is absolutely logical. The reviewer has done a successful arrangement of the complexes based on the metal and we feel guilty because we do not adopt her/his suggestion. The main reasons are: (a) Since most of the reported compounds are coordination clusters, we prefer to keep the arrangement according to the nuclearity which is related to properties. (b) The arrangement based on the metal is illustrated in Figure 101 in a convenient way which perhaps is more friendly to the reader; and (c) Arrangement based on the metal would practically require writing of a “new” review, a task that is impossible during the narrow timetable (after a two-week extension!) given by the journal. In order to make clear that we respect the reviewer’s opinion, we have added two new sentences (“Another way would be arranging…way of presentation”) in the same section of the revised ms. However, we keep in mind this suggestion for our next review (to be written for “Molecules”) dealing with the “Periodic Table of Phenyl 2-Pyridyl Ketoxime”, the latter being one of our favourite ligands with which we have been working (in the last 25 years or so). Based on the above mentioned, we are asking your and Academic Editor’s indulgence to keep the present organization of the review.
“-Additionally, based on my independent literature search using SciFinder, I noticed that these references are not cited in the manuscript. I recommend reviewing them and citing them as appropriate.
DOI: 10.1080/00958972.2019.1590708
DOI: 10.1016/j.poly.2017.09.008
DOI: 10.1016/j.molstruc.2017.10.087
DOI: 10.1016/j.ejmech.2013.12.019
DOI: 10.1016/j.jinorgbio.2016.04.023
DOI: 10.1155/S1565363303000074
Patent: China, CN114874269 A 2022-08-09
Anticancer Research (2000), 20(6B), 4435-4439”
An excellent help by the reviewer! We have incorporated the suggested references (refs. [30], [32], [36], [38], [45], [89]) with the relevant complexes (7, 9, 10, 15, 17, 26, 105) in the revised version of the ms.
“-In general, these balanced equations are unrealistic and should be omitted, as the reactions do not yield the sole products discussed in the paper. The authors should exercise caution in the messages conveyed through the inclusion of balanced equations.”
This is the second (and final) point for which we have not addressed the suggestion by the reviewer. We had already mentioned the limitation of writing balanced chemical equations in section 1 of the originally submitted ms (“We intend to avoid long preparative…by the low yields of the preparations”). Also, we had used text to describe some processes. In general, we do believe that balanced chemical equations are useful for the reader in order to understand the chemistry involved (e.g. reactants’ theoretical ratio, change of the metal oxidation state in redox processes, necessity of external bases, etc.). Narrative descriptions of the preparations might be somewhat boring. However, in order to show that we respect the reviewer’s view, we have added a new sentence in the revised ms (section 1). [“However (as one of the referees points out), caution is needed in the messages conveyed through the use of balanced equations”]. Also, please note that we used balanced chemical equations in ca. 10 reviews and book chapters [e.g. Chemistry/MDPI, volume 5, pp. 1419-1453 (2023), Inorganics/MDPI, volume 12, article 208 (2024), Inorganics/MDPI, volume 8, article 39 (2020), Int. J. Mol. Sci/MDPI, volume 21, article 555 (2020), Frontiers Chem., volume 6, article 461 (2018), Coord. Chem. Rev., volume 256, 1246-1278 (2012), Inorg. Chim. Acta, volume 539, article 120954 (2022)] and this practice was well received by The Editors and the reviewers. Thus, we are asking your and Academic Editor’s indulgence to keep the ca. 100 balanced chemical equations in the review.
“Abstract: Please remove all the ‘‘fudge words’’: as such ‘‘aesthetically’’ ‘‘beautiful’’ etc. No need to exaggerate; just maintain a meaningful scientific context.”
We have removed all the “fudge words” from Abstract and text.
“Abstract: Please list lanthanoids elements too.”
We have listed lanthanoid elements in Abstract.
“Abstract: This sentence ‘‘Our efforts to complete the “periodic table” of di-2-pyridyl ketoxime are continued’’ should be in Conclusions not in abstract.”
We have moved the suggested sentence from Abstract in section 7 (“Concluding Comments and Perspectives”).
“Page 1, Line 43: The oxime group should be presented as a formula, not as a ChemDraw diagram, in the text, using the notation as R2C=NOH (R = alkyl or aryl).”
We have presented the oxime group as suggested.
“Line 100-101: “Since the descriptions of the structures and the spectroscopic/physical properties will be short, we assume that the readers of this review have a good background in coordination chemistry.” Please remove this sentence, as this is a review intended for readers specialized in coordination chemistry in this area. There is no need to state something so obvious, which only belabors the context.”
The comment is correct. We have removed the sentence.
“Page 3, Line ‘‘some structures will be illustrated in a schematic way (ChemDraw)’’ Why future tense? you could use the present tense: "Some structures are illustrated schematically (ChemDraw)."
We have performed the correction.
“Same as here: ‘‘The ligation modes will be also represented in a schematic way (ChemDraw)’’. Please correct.”
We have performed the correction.
“Page 3, Line 113: ‘‘Sections 2-4 are referred to as the “hors d’oeuvre” of the article.’’ I am unsure what “hors d’oeuvre” means in this context. Since it is non-English and potentially confusing, please remove it.”
We have replaced this French phrase with the word “introductory”.
“Page 3, Line 114: ‘‘Sections 5 and 6 contain the “main menu” of this scientific meal.’’ Please remove all these metaphor and analogy. Just keep plain and simple science context.”
We have modified this metaphor and used plain English.
“Page 4: ‘‘The Oxime Group: A “Passepartout” in Chemistry’’ – Please simplify the title by keeping it in plain English and removing non-English words. Such as: ‘‘The Oxime Group: A Versatile Tool in Chemistry’’.
We have modified the title by following the reviewer’s suggestion. Thus, the new title is “The Oxime Group: A Versatile Tool in Chemistry”.
“Page 4, Line 149: ‘‘..derivatives of oximes are central “players” in modern agriculture as agents’’. Please remove any informal English.”
We have removed this informal English and used the word “important”.
“Page 23, Line 531: ‘‘After many “try and see” synthetic exercises..’’ Please remove any informal English.”
We have removed this informal phrase and replaced by “After many experiments……”.
“Page 27, Figure 29, legend: Please use the formula instead of a ChemDraw image.”
We have replaced the ChemDraw image.
“Page 48, Line 1185, Please use the formula for the oxime group.”
We have used the formula for the oxime group.
“Page 57, Line 1469, ‘‘..due to the ionic [M-ClO4-4H2O]+ ionic species’’. Correct sentence.”
We have corrected the sentence.
“Page 76, Line 2015: "‘…ligands participate with five(!) different ligation modes.’ Why is the exclamation symbol used?"
We agree and have removed the exclamation symbol.
In summary, the implementation of most of the above mentioned revision points/comments/suggestions has remarkably improved the quality and readability of the ms. Again, we express our gratitude to the reviewer for her/his critical and laborious work.